# Learning with convolution and pooling operations in kernel methods

**Theodor Misiakiewicz**
Department of Statistics
Stanford University
Stanford, CA 94305
misiakie@stanford.edu

**Song Mei**
Department of Statistics
University of California Berkeley
Berkeley, CA 94720
songmei@berkeley.edu

## Abstract

Recent empirical work has shown that hierarchical convolutional kernels inspired by convolutional neural networks (CNNs) significantly improve the performance of kernel methods in image classification tasks. A widely accepted explanation for their success is that these architectures encode hypothesis classes that are suitable for natural images. However, understanding the precise interplay between approximation and generalization in convolutional architectures remains a challenge. In this paper, we consider the stylized setting of covariates (image pixels) uniformly distributed on the hypercube, and characterize exactly the RKHS of kernels composed of single layers of convolution, pooling, and downsampling operations. We use this characterization to compute sharp asymptotics of the generalization error for any given function in high-dimension. In particular, we quantify the gain in sample complexity brought by enforcing locality with the convolution operation and approximate translation invariance with average pooling. Notably, these results provide a precise description of how convolution and pooling operations trade off approximation with generalization power in one layer convolutional kernels.

## 1   Introduction

Convolutional neural networks (CNNs) have become essential elements of the deep learning toolbox, achieving state-of-the-art performance in many computer vision tasks [27, 30]. CNNs are constructed by stacking convolution and pooling layers, which were shown to be paramount to their empirical success [31]. A widely accepted hypothesis to explain their favorable properties is that these architectures successfully encode useful properties of natural images: locality and compositionality of the data, stability by local deformations, and translation invariance. While some theoretical progress has been made in studying the approximation and generalization benefits brought by convolution and pooling operations [6, 15, 16], our mathematical understanding of the interaction between network architecture, image distribution, and efficient learning remains limited.

Consider $\boldsymbol{x} \in \mathbb{R}^d$ an input signal, which we can think of as a grayscale pixel representation of an image. For mathematical convenience, we will consider one-dimensional images with cyclic convention $x_{d+i} := x_i$, and denote $\boldsymbol{x}_{(k)} = (x_k, x_{k+1}, \ldots, x_{k+q-1})$ the $k$-th patch of the signal $\boldsymbol{x}$, $k \in [d]$, with patch size $q \leq d$. Most of our results can be extended to two-dimensional images.

We further consider a simple convolutional neural network composed of a single convolution layer followed by local average pooling and downsampling. The network first computes the nonlinear convolution of $N$ filters $\boldsymbol{w}_1, \ldots, \boldsymbol{w}_N \in \mathbb{R}^q$ with the image patches $\boldsymbol{x}_{(k)}$. The outputs of the convolution operation $\sigma(\langle \boldsymbol{w}_i, \boldsymbol{x}_{(k)} \rangle)$ are then averaged locally over segments of length $\omega$ (local average pooling). This pooling operation is followed by downsampling which extracts one out of every $\Delta$ output coordinates (for simplicity, $\Delta$ is assumed to be a divisor of $d$). Finally, the results are

36th Conference on Neural Information Processing Systems (NeurIPS 2022).

combined linearly using coefficients $(a_{ik})_{i \in [N], k \in [d/\Delta]}$:

$$f_{\mathsf{CNN}}(\boldsymbol{x}; \boldsymbol{a}, \boldsymbol{\Theta}) = \sqrt{\frac{\Delta}{N\omega d}} \sum_{i \in [N]} \sum_{k \in [d/\Delta]} a_{ik} \sum_{s \in [\omega]} \sigma\left(\langle \boldsymbol{w}_i, \boldsymbol{x}_{(k\Delta+s)} \rangle\right) . \qquad \text{(CNN-AP-DS)}$$

Note that pooling and downsampling operations are often tied together in the literature. However in this work we will treat these two operations separately.

In the formula above, different values for $q, \omega, \Delta$ lead to different architectures with vastly different behaviors. For example, when $q = \Delta = d$ and $\omega = 1$, we recover a two-layer *fully-connected* neural network $f_{\mathsf{FC}}(\boldsymbol{x}; \boldsymbol{a}, \boldsymbol{\Theta}) = N^{-1/2} \sum_{i \in [N]} a_i \sigma(\langle \boldsymbol{w}_i, \boldsymbol{x} \rangle)$ which has the universal approximation property at large $N$. When $\omega = \Delta = 1$ and $q < d$, the network is *locally connected* $f_{\mathsf{LC}}(\boldsymbol{x}; \boldsymbol{a}, \boldsymbol{\Theta}) = N^{-1/2} \sum_{i \in [N], k \in [d]} a_{ik} \sigma(\langle \boldsymbol{w}_i, \boldsymbol{x}_{(k)} \rangle)$, and not a universal approximator anymore: however, $f_{\mathsf{LC}}$ vastly outperforms $f_{\mathsf{FC}}$ in some cases [33]. For $\omega > 1$, local pooling enables learning functions that are locally invariant by translations more efficiently than without pooling. For $\omega = d$ (*global pooling*), the network only fits functions fully invariant by cyclic translations.

The aim of this paper is to formalize and quantify the *interplay between the target function class and the statistical efficiency brought by these different architectures*. As a concrete first step in this direction, we consider kernel models that are naturally associated to (CNN-AP-DS) through the neural tangent kernel perspective [18, 28]. Kernel methods have the advantage of 1) being tractable—leaving the computational issue of learning CNNs aside; 2) having well-understood approximation and generalization properties, which depends on the eigendecomposition of the kernel and the alignment between the target function and the RKHS [12, 47] (see Appendices B and C for background). While kernel models only describe neural networks in the lazy training regime [1, 13, 19, 20, 49] and miss important properties of deep learning, such as feature learning, architecture choice already plays a crucial role to learn efficiently 'image-like' functions in the fixed-feature regime.

Neural tangent kernels are obtained by linearizing the associated neural networks. Here we consider the tangent kernel associated to the network $f_{\mathsf{CNN}}$ (c.f. Appendix A.2 for a detailed derivation):

$$H^{\mathsf{CK}}_{\omega,\Delta}(\boldsymbol{x}, \boldsymbol{y}) = \frac{\Delta}{d\omega} \sum_{k \in [d/\Delta]} \sum_{s,s' \in [\omega]} h\left(\langle \boldsymbol{x}_{(k\Delta+s)}, \boldsymbol{y}_{(k\Delta+s')} \rangle / q\right), \qquad \text{(CK-AP-DS)}$$

where $h : \mathbb{R} \to \mathbb{R}$ is related to the activation function $\sigma$ in (CNN-AP-DS). As a linearization of CNNs, the kernel (CK-AP-DS) inherits some of the favorable properties of convolution, pooling, and downsampling operations. Indeed, a line of work [2, 32, 35, 36, 45] showed that, though performing slightly worse than CNNs, such (hierarchical) convolutional kernels have empirically outperformed the former state-of-the-art kernels. For instance, these kernels achieved test accuracy around $87\% - 90\%$ on CIFAR-10 (the state-of-the-art CNNs can achieve test accuracy $99\%$), against $79.6\%$ for the best former unsupervised feature-extraction method [14].

In this paper, we will further consider a stylized setting with input signal distribution $\boldsymbol{x} \sim \mathrm{Unif}(\mathscr{Q}^d)$ (uniform distribution over $\mathscr{Q}^d := \{-1, +1\}^d$ the discrete hypercube in $d$ dimensions). This simple choice allows for a complete characterization of the eigendecomposition of $H^{\mathsf{CK}}_{\omega,\Delta}$, thanks to all patches having same marginal distribution $\boldsymbol{x}_{(k)} \sim \mathrm{Unif}(\mathscr{Q}^q)$. We will be particularly interested in four specific choices of $(q, \omega, \Delta)$ in (CK-AP-DS):

$$H^{\mathsf{FC}}(\boldsymbol{x}, \boldsymbol{y}) = h\left(\langle \boldsymbol{x}, \boldsymbol{y} \rangle / d\right), \qquad \text{(FC)}$$

$$H^{\mathsf{CK}}(\boldsymbol{x}, \boldsymbol{y}) = \frac{1}{d} \sum_{k \in [d]} h\left(\langle \boldsymbol{x}_{(k)}, \boldsymbol{y}_{(k)} \rangle / q\right), \qquad \text{(CK)}$$

$$H^{\mathsf{CK}}_{\omega}(\boldsymbol{x}, \boldsymbol{y}) = \frac{1}{d\omega} \sum_{k \in [d]} \sum_{s,s' \in [\omega]} h\left(\langle \boldsymbol{x}_{(k+s)}, \boldsymbol{y}_{(k+s')} \rangle / q\right), \qquad \text{(CK-AP)}$$

$$H^{\mathsf{CK}}_{\mathsf{GP}}(\boldsymbol{x}, \boldsymbol{y}) = \frac{1}{d} \sum_{k,k' \in [d]} h\left(\langle \boldsymbol{x}_{(k)}, \boldsymbol{y}_{(k')} \rangle / q\right). \qquad \text{(CK-GP)}$$

These kernels are respectively the neural tangent kernels of a fully-connected network $f_{\mathsf{FC}}$ (FC), a convolutional network $f_{\mathsf{LC}}$ (CK), a convolutional network followed by local average pooling (CK-AP)

and a convolutional network followed by global pooling (CK-GP). We will further be interested in (CK-GP) with patch size $q = d$, which we denote $H_{\mathsf{GP}}^{\mathsf{FC}}$: this corresponds to a convolutional kernel with full-size patches $q = d$, followed by global pooling.

In this paper, we first characterize the reproducing kernel Hilbert space (RKHS) of these convolutional kernels, and then investigate their generalization properties in the regression setup. More specifically, assume $\{(\boldsymbol{x}_i, y_i)\}_{i \leq n}$ are $n$ i.i.d. samples with $\boldsymbol{x}_i \sim \mathrm{Unif}(\mathscr{Q}^d)$ and $y_i = f_\star(\boldsymbol{x}_i) + \varepsilon_i$. Here $f_\star \in L^2(\mathscr{Q}^d)$ and $(\varepsilon_i)_{i \leq n}$ are independent errors with mean zero and variance bounded by $\sigma_\varepsilon^2$. We will focus on the generalization error of kernel ridge regression (KRR) (see Appendix B.1 for general kernel methods). In particular, given a kernel function $H : \mathscr{Q}^d \times \mathscr{Q}^d \to \mathbb{R}$ and a regularization parameter $\lambda \geq 0$, the KRR estimator is the solution of the tractable convex problem

$$\hat{f}_\lambda = \operatorname*{arg\,min}_{f \in \mathcal{H}} \left\{ \sum_{i \in [n]} \left( y_i - f(\boldsymbol{x}_i) \right)^2 + \lambda \|f\|_{\mathcal{H}}^2 \right\}, \tag{KRR}$$

where $\mathcal{H}$ is the RKHS associated to $H$ with RKHS norm $\| \cdot \|_{\mathcal{H}}$. We denote the test error with square loss by $R(f_\star, \hat{f}_\lambda) = \mathbb{E}_{\boldsymbol{x}}\{(f_\star(\boldsymbol{x}) - \hat{f}_\lambda(\boldsymbol{x}))^2\}$. We will sometimes consider the expected test error $\mathbb{E}_{\boldsymbol{\varepsilon}}\{R(f_\star, \hat{f}_\lambda)\}$, where expectation is taken with respect to noise $\boldsymbol{\varepsilon} = (\varepsilon_i)_{i \leq n}$ in the training data.

The generalization properties of kernels $H^{\mathsf{FC}}$ and $H_{\mathsf{GP}}^{\mathsf{FC}}$ were recently studied in [7, 39]. In particular, they showed that global pooling (kernel $H_{\mathsf{GP}}^{\mathsf{FC}}$) leads to a gain of a factor $d$ in sample complexity when fitting cyclic invariant functions, but still suffers from the curse of dimensionality ($H_{\mathsf{GP}}^{\mathsf{FC}}$ only fits very smooth functions in high-dimension). More precisely, Mei et al. [39] considered the high-dimensional framework of [38] and showed the following: KRR with $H^{\mathsf{FC}}$ requires $n \approx d^\ell$ samples to fit degree-$\ell$ cyclic polynomials, while KRR with $H_{\mathsf{GP}}^{\mathsf{FC}}$ only needs $n \approx d^{\ell-1}$. To enable milder dependence on the dimension $d$, further structural assumptions on the kernel and the target function should be considered (for instance, in this paper, we use the kernel $H^{\mathsf{CK}}$ and consider 'local' functions).

## 1.1 Summary of main results

Our contributions are two-fold. First, we describe the RKHS associated with the convolutional kernel (CK-AP-DS) in the stylized setting $\boldsymbol{x} \sim \mathrm{Unif}(\mathscr{Q}^d)$, which provides a fully explicit picture of the roles of convolution, pooling and downsampling operations in approximating specific classes of functions. Second, we provide sharp asymptotics for the generalization error of KRR in high-dimension, given any target function and one of the kernels described in the introduction[1]. These asymptotics are obtained rigorously using the framework of [38] (see Appendix C for background). For completeness, we also include bounds on the KRR test error in the classical fixed-dimension setting with capacity/source assumptions (see Appendix C for limitations of this classical approach).

We summarize our results below. Define the $q$-local function class $L^2(\mathscr{Q}^d, \mathrm{Loc}_q)$ and the cyclic $q$-local function class $L^2(\mathscr{Q}^d, \mathrm{CycLoc}_q)$ (subspace of $L^2(\mathscr{Q}^d, \mathrm{Loc}_q)$ consisting of cyclic-invariant functions) as follows:

$$L^2(\mathscr{Q}^d, \mathrm{Loc}_q) = \left\{ f \in L^2(\mathscr{Q}^d) : \exists \{g_k\}_{k \in [d]} \subseteq L^2(\mathscr{Q}^q), f(\boldsymbol{x}) = \sum_{k \in [d]} g_k(\boldsymbol{x}_{(k)}) \right\}, \tag{LOC}$$

$$L^2(\mathscr{Q}^d, \mathrm{CycLoc}_q) = \left\{ f \in L^2(\mathscr{Q}^d) : \exists g \in L^2(\mathscr{Q}^q), f(\boldsymbol{x}) = \sum_{k \in [d]} g(\boldsymbol{x}_{(k)}) \right\}. \tag{CYC-LOC}$$

**One-layer convolutional layer.** The RKHS of $H^{\mathsf{CK}}$ is equal to $L^2(\mathscr{Q}^d, \mathrm{Loc}_q)$: kernel methods with $H^{\mathsf{CK}}$ can only fit the projection $\mathrm{P}_{\mathrm{Loc}_q} f_*$ of the target function onto $L^2(\mathscr{Q}^d, \mathrm{Loc}_q)$. For a sample size $n \asymp dq^{\ell-1}$, KRR fits exactly a degree-$\ell$ polynomial approximation to $\mathrm{P}_{\mathrm{Loc}_q} f_*$. In particular, for $q \ll d$, the convolution kernel $H^{\mathsf{CK}}$ is much more sample efficient than the standard inner-product kernel $H^{\mathsf{FC}}$ for fitting functions in $L^2(\mathscr{Q}^d, \mathrm{Loc}_q)$ (sample sizes $dq^{\ell-1} \ll d^\ell$ for fitting a degree-$\ell$ polynomials). *The convolution operation breaks the curse of dimensionality by restricting the RKHS to local functions.*

---

[1]Note that we modify slightly $H_\omega^{\mathsf{CK}}$ to simplify the derivation of the high-dimension asymptotics. However, we believe such a simplification to be unecessary. The fixed-dimension bounds do not require such a simplification.

| To fit a degree $\ell$ polynomial | $H^{\mathsf{FC}}$ | $H^{\mathsf{FC}}_{\mathsf{GP}}$ | $H^{\mathsf{CK}}$ | $H^{\mathsf{CK}}_{\omega}$ | $H^{\mathsf{CK}}_{\mathsf{GP}}$ |
|---|---|---|---|---|---|
| Sample complexity | $d^{\ell}$ | $d^{\ell-1}$ | $dq^{\ell-1}$ | $dq^{\ell-1}/\omega$ | $q^{\ell-1}$ |

Table 1: Sample size $n$ required to fit a $q$-local cyclic-invariant polynomial of degree $\ell$ using kernel ridge regression (KRR) with the 5 different kernels of interest in this paper.

**Average pooling.** The RKHS of $H^{\mathsf{CK}}_{\omega}$ is still constituted of $q$-local functions $f_* \in L^2(\mathscr{Q}^d, \mathrm{Loc}_q)$, but penalizes differently the frequency components $f_{*,j}(\boldsymbol{x})$ by reweighting their eigenspaces by a factor $\kappa_j$, where $f_{*,j}(\boldsymbol{x}) = \sum_{k\in[d]} \rho_j^k f_*(t_k \cdot \boldsymbol{x})$ with $\rho_j = e^{\frac{2i\pi j}{d}}$ and $t_k \cdot \boldsymbol{x} = (x_{k+1}, \ldots, x_d, x_1, \ldots, x_k)$ denotes the $k$-shift. As $\omega$ increases, local pooling penalizes more and more heavily the high-frequency components ($\kappa_j \ll 1$), while making low-frequency components statistically easier to learn ($\kappa_j \gg 1$). For global pooling $\omega = d$, $H^{\mathsf{CK}}_{\mathsf{GP}}$ only learns cyclic local functions $L^2(\mathscr{Q}^d, \mathrm{CycLoc}_q)$ and enjoy a factor $d$ gain in statistical complexity compared to $H^{\mathsf{CK}}$ (sample sizes $q^{\ell-1} \ll dq^{\ell-1}$ to learn a degree-$\ell$ polynomial). *Local pooling biases learning towards functions that are stable by small translations.*

**Downsampling.** When $\Delta \leq \omega$, downsampling after average pooling leaves the low-frequency eigenspaces of $H^{\mathsf{CK}}_{\omega}$ stable. In particular, the downsampling operation does not modify the statistical complexity of learning low-frequency functions in one-layer kernels, while being potentially beneficial in further layers in deep convolutional kernels.

These theoretical results answer the following question: *given a target function and a sample size $n$, what is the impact of the architecture on the test error?* For example, Table 1 shows how the architecture modify the sample size required to achieve small test error when learning a degree-$\ell$ polynomial in $L^2(\mathscr{Q}^d, \mathrm{CycLoc}_q)$.

There are two important model assumptions in this paper, which deserve some discussions:

**One-layer convolutional kernel (CK):** extra layers allow for hierarchical interactions between the patches (see for example [6]). However, we believe that the main insights on the approximation and statistical trade-off are already captured in the one-layer case (see [48] for multi-layer but independent patches). Note that depth might be less important for CKs than for CNNs: the one-layer CK considered in this paper achieves $80.9\%$ accuracy on CIFAR-10 [6] (versus $79.6\%$ in [14]) and 3-layers CK achieves $88.2\%$ accuracy [6] (versus $90\%$ for the best multi-layer CK [45]). See Appendix A.5 for a discussion on how our results could be extended to 2-layers.

**Data uniform on the hypercube:** this choice is motivated by our goal of deriving rigorous fine-grained approximation and generalization errors, which requires to diagonalize the kernel (CK-AP-DS). More general data distributions either require strong assumptions (independent patches [43, 48]), loose minmax bounds on the generalization error (e.g., classical source/capacity assumptions) or non-rigorous statistical physics heuristics [22].

The rest of the paper is organized as follows. We discuss related work in Section 1.2. In Section 2, we present our main results on convolutional kernels and describe precisely the roles of convolution, pooling and downsampling operations. Finally, we present a numerical simulation on synthetic data in Section 3 and conclude in Section 4. Some details and discussions are deferred to Appendix A.

## 1.2 Related work

Convolutional kernels have been considered in [6, 32, 35, 36, 45, 46]. In particular, they showed that these architectures achieve good results in image classification ($90\%$ accuracy on Cifar10) and that pooling and downsampling were necessary for their good performance [32].

The generalization error of kernel ridge regression (KRR) has been well-studied in both the fixed dimension regime [47, Chap. 13], [12] and the high-dimensional regimes [21, 23, 24, 34, 39, 48]. These results show that the generalization error depends on the eigenvalues and eigenfunctions of the kernel, and the alignment of the kernel with the target function.

Recently, a few theoretical work have considered the generalization properties of invariant kernels and convolutional kernels [7, 22, 39, 43, 48]. In particular, a concurrent work [48] considers sharp asymptotics of the KRR test error using the framework of [38] for certain hierarchical convolutional

kernels under the strong assumption of non-overlapping patches (whereas we consider the more natural architecture of overlapping patches). They arrive at a similar trade-off between approximation and generalization power in convolutional kernels, which they call 'eigenspace restructuring principle': given a finite statistical budget (i.e., a sample size $n$), convolutional architectures allocate the 'eigenvalue mass' by weighting differently the eigenspaces. Favero et al. [22] considers a one-layer convolutional kernel with and without global pooling and derive asymptotic rates in sample size $n$ in a student-teacher scenario using statistical physics heuristics and a Gaussian equivalence conjecture. In particular, they show that locality rather than translation-invariance breaks the curse of dimensionality. Our goal in this paper is different: we derive *rigorous quantitative bounds* that give separation in generalization power between different architectures.

See [33, 37] for more theoretical results on the separation between convolutional and fully connected neural networks, and [10, 16] for the inductive bias of pooling operations in convolutional neural networks.

## 2 Main results

We start by introducing some background on functions on the hypercube and eigendecomposition of kernel operators in Section 2.1. We first consider a kernel with a single convolution layer in Section 2.2, and characterize its eigendecomposition and generalization properties. We then show how these results are modified when applying local average pooling and downsampling in Section 2.3.

### 2.1 Functions on the hypercube and eigendecomposition of kernel operators

Recall that we work on the $d$-dimensional hypercube $\mathcal{Q}^d := \{-1, +1\}^d$. Let $L^2(\mathcal{Q}^d) = L^2(\mathcal{Q}^d, \mathrm{Unif})$ be the $2^d$-dimensional vector space of all functions $f : \mathcal{Q}^d \to \mathbb{R}$, with scalar product $\langle f, g \rangle_{L^2} := \mathbb{E}_{\boldsymbol{x} \sim \mathrm{Unif}(\mathcal{Q}^d)}[f(\boldsymbol{x})g(\boldsymbol{x})]$. Let $\| \cdot \|_{L^2}$ be the norm associated with the scaler product. We introduce the set of Fourier functions $\{Y_S^{(d)}(\boldsymbol{x})\}_{S \subseteq [d]}$ which forms an orthonormal basis of $L^2(\mathcal{Q}^d)$. For any subset $S \subseteq [d]$, the Fourier function is defined as $Y_S^{(d)}(\boldsymbol{x}) := \prod_{i \in S} x_i$ with the convention that $Y_\emptyset^{(d)} := 1$ (it is easy to verify that $\langle Y_S^{(d)}, Y_{S'}^{(d)} \rangle_{L^2} = \mathbf{1}_{S=S'}$). We will omit the superscript $(d)$ which will be clear from context and write $Y_S := Y_S^{(d)}$.

Consider a nonnegative definite kernel function $H : \mathcal{Q}^p \times \mathcal{Q}^p \to \mathbb{R}$ ($p = d$ or $q$ in this paper) with associated integral operator $\mathbb{H} : L^2(\mathcal{Q}^p) \to L^2(\mathcal{Q}^p)$ defined as $\mathbb{H}f(\boldsymbol{u}) = \mathbb{E}_{\boldsymbol{v}}\{h(\boldsymbol{u}, \boldsymbol{v})f(\boldsymbol{v})\}$ with $\boldsymbol{v} \sim \mathrm{Unif}(\mathcal{Q}^p)$. By spectral theorem of compact operators, there exists an orthonormal basis $\{\psi_j\}_{j \geq 1}$ of $L^2(\mathcal{Q}^p)$ and nonnegative eigenvalues $(\lambda_j)_{j \geq 1}$ such that $\mathbb{H} = \sum_{j \geq 1} \lambda_j \psi_j \psi_j^*$ (i.e., $H(\boldsymbol{u}, \boldsymbol{v}) = \sum_{j \geq 1} \lambda_j \psi_j(\boldsymbol{u})\psi_j(\boldsymbol{v})$ for any $\boldsymbol{u}, \boldsymbol{v} \in L^2(\mathcal{Q}^p)$).

The most widespread example are *inner-product* kernels defined as $H(\boldsymbol{u}, \boldsymbol{v}) := h(\langle \boldsymbol{u}, \boldsymbol{v} \rangle / p)$ for some function $h : \mathbb{R} \to \mathbb{R}$. Inner-product kernels have the following simple eigendecomposition in $L^2(\mathcal{Q}^p)$ (taking here $\boldsymbol{u}, \boldsymbol{v} \in \mathcal{Q}^p$):

$$h(\langle \boldsymbol{u}, \boldsymbol{v} \rangle / p) = \sum_{\ell=0}^{p} \xi_{p,\ell}(h) \sum_{S \subseteq [p], |S|=\ell} Y_S(\boldsymbol{u}) Y_S(\boldsymbol{v}), \tag{1}$$

where $\xi_{p,\ell}(h)$ is the $\ell$-th Gegenbauer coefficient of $h(\cdot / \sqrt{p})$ in dimension $p$, i.e.,

$$\xi_{p,\ell}(h) = \mathbb{E}_{\boldsymbol{u} \sim \mathrm{Unif}(\mathcal{Q}^p)}\big[h(\langle \boldsymbol{u}, \boldsymbol{e} \rangle / p) Q_\ell^{(p)}(\langle \boldsymbol{u}, \boldsymbol{e} \rangle)\big], \tag{2}$$

for $\boldsymbol{e} \in \mathcal{Q}^p$ arbitrary and $Q_\ell^{(p)}$ the degree-$\ell$ Gegenbauer polynomial on $\mathcal{Q}^p$ (see Appendix D for details). Note that $(\xi_{p,\ell})_{0 \leq \ell \leq q}$ are non-negative by positive semidefiniteness of the kernel. We will write $\xi_{p,\ell} := \xi_{p,\ell}(h)$ and use extensively the decomposition identity (1) in the rest of the paper.

### 2.2 One-layer convolutional kernel

We first consider the convolutional kernel $H^{\mathsf{CK}}$ (CK) given by a one-layer convolution layer with patch size $q$ and inner-product kernel function $h : \mathbb{R} \to \mathbb{R}$:

$$H^{\mathsf{CK}}(\boldsymbol{x}, \boldsymbol{y}) = \frac{1}{d} \sum_{k=1}^{d} h\left(\langle \boldsymbol{x}_{(k)}, \boldsymbol{y}_{(k)} \rangle / q\right), \tag{3}$$

where we recall that $\boldsymbol{x}_{(k)} = (x_k, \ldots, x_{k+q-1}) \in \mathcal{Q}^q$ is the $k$'th patch of the image with size $q$.

Before stating the eigendecomposition of $H^{\mathsf{CK}}$, we introduce some notations. For any subset $S \subseteq [d]$, denote $\gamma(S)$ the diameter of $S$ with cyclic convention, i.e., $\gamma(S) = \max\{\min\{\mathrm{mod}(j - i, d) + 1, \mathrm{mod}(i - j, d) + 1\} : i, j \in S\}$ (e.g., $\gamma(\{2, d\}) = 3$). For any integer $\ell \le q$, consider the set $\mathcal{E}_\ell = \{S \subseteq [d] : |S| = \ell, \gamma(S) \le q\}$ of all subsets of $[d]$ of size $\ell$ with diameter less or equal to $q$. We will assume throughout this paper that $q \le d/2$ to avoid additional overlap between sets.

**Proposition 1** (Eigendecomposition of $H^{\mathsf{CK}}$). *Let $H^{\mathsf{CK}}$ be a convolutional kernel as defined in Eq. (3). Then $H^{\mathsf{CK}}$ admits the following eigendecomposition:*

$$H^{\mathsf{CK}}(\boldsymbol{x}, \boldsymbol{y}) = \xi_{q,0} + \sum_{\ell=1}^q \sum_{S \in \mathcal{E}_\ell} \frac{r(S)\xi_{q,\ell}}{d} \cdot Y_S(\boldsymbol{x}) Y_S(\boldsymbol{y}), \tag{4}$$

*where $r(S) = q + 1 - \gamma(S)$ and $\xi_{q,\ell} \ge 0$ is defined in Eq. (2).*

Notice that $Y_S$ with $\gamma(S) > q$ (monomials with support not contained in a segment of size $q$) are in the null space of $H^{\mathsf{CK}}$. Hence (as long as $\xi_{q,\ell} > 0$ for all $0 \le \ell \le q$), the RKHS associated to $H^{\mathsf{CK}}$ exactly contains all the functions in the $q$-local function class $L^2(\mathcal{Q}^d, \mathrm{Loc}_q)$ (c.f. Eq. (LOC)). In words, $L^2(\mathcal{Q}^d, \mathrm{Loc}_q)$ consists of functions that are localized on patches, with no long-range interactions between different parts of the image. An example of local function with $q = 3$ is given by $f(\boldsymbol{x}) = x_1 x_2 x_3 + x_4 x_6 + x_5$.

On the other hand, the RKHS associated to the fully-connected kernel $H^{\mathsf{FC}}$ (FC) typically contains all the functions in $L^2(\mathcal{Q}^d)$ (under genericity assumptions on $h$). The RKHS with convolution $\dim(L^2(\mathcal{Q}^d, \mathrm{Loc}_q)) = d2^{q-1} + 1$ is significantly smaller than $\dim(L^2(\mathcal{Q}^d)) = 2^d$, which prompts the following question: *what is the statistical advantage of using $H^{\mathsf{CK}}$ over $H^{\mathsf{FC}}$ when learning functions in $L^2(\mathcal{Q}^d, \mathrm{Loc}_q)$?*

We first consider the classical approach to bounding the test error of [3, 12, 47] which relies on the following two standard assumptions:

(A1) *Capacity condition:* we assume $\mathcal{N}(h, \lambda) := \mathrm{Tr}[h/(h + \lambda \mathbf{I})^{-1}] \le C_h \lambda^{-1/\alpha}$ with[2] $\alpha > 1$.

(A2) *Source condition:* $\|h^{-\beta/2} g\|_{L^2} \le B$ with[3] $\beta > \frac{\alpha-1}{\alpha}$ and $B \ge 0$.

The capacity condition (A1) characterizes the size of the RKHS: for increasing $\alpha$, the RKHS contains less and less functions. The source condition (A2) characterizes the regularity of the target function (the 'source') with respect to the kernel: increasing $\beta$ corresponds to smoother and smoother functions. See Appendix B.2 for more discussions.

Based on these two assumptions, we can apply standard bounds on the KRR test error and obtain:

**Theorem 1** (Generalization error of KRR with $H^{\mathsf{CK}}$). *Let $h : \mathbb{R} \to \mathbb{R}$ be an inner-product kernel satisfying (A1). Let $f_\star \in L^2(\mathcal{Q}^d, \mathrm{Loc}_q)$ with $f_\star(\boldsymbol{x}) = \sum_{k \in [d]} g_k(\boldsymbol{x}_{(k)})$ satisfying source condition $\sum_{k \in [d]} \|h^{-\beta/2} g_k\|_{L^2}^2 \le B^2$. Then there exists $C_1, C_2, C_3 > 0$ constants that only depend on (A1) and $B$ (and independent of $d$), such that for $n \ge C_1 \max(\|f_\star\|_{L^\infty}^2, d)$ and $\lambda_\star = \frac{C_2}{d}(d/n)^{\frac{\alpha}{\alpha\beta+1}}$,*

$$\mathbb{E}_{\boldsymbol{\varepsilon}}\{R(f_\star, \hat{f}_{\lambda_\star})\} \le C_3 \left(\frac{d}{n}\right)^{\frac{\alpha\beta}{\alpha\beta+1}}. \tag{5}$$

Note that the exponent $\frac{\beta\alpha}{\beta\alpha+1}$ only depends on the $q$-dimensional kernel $h$. Hence, the generalization bound with respect to $(n/d)$ is independent of the dimension $d$ of the image. Let's compare to KRR with inner-product kernel $H^{\mathsf{FC}}$ (FC): from [12], we have the minmax rate $\mathbb{E}_{\boldsymbol{\varepsilon}}\{R(f_\star, \hat{f}_\lambda)\} \asymp n^{-\frac{\tilde{\alpha}\tilde{\beta}}{\tilde{\alpha}\tilde{\beta}+1}}$ where $h$ is now defined in $d$ dimension and verifies (A1) and (A2) with constants $\tilde{\alpha}, \tilde{\beta}$. Typically, if $f_\star$ is only assumed Lipschitz, then $\tilde{\beta}\tilde{\alpha} = O(1/d)$, which leads to a minmax rate $n^{-O(1/d)}$ for $H^{\mathsf{FC}}$, while for $H^{\mathsf{CK}}$, $\beta\alpha = O(1/q)$, which leads to a minmax rate $n^{-O(1/q)}$. Hence, for $q \ll d$,

---

[2] Here, $h$ is the integral operator and $\mathrm{Tr}[h/(h + \lambda \mathbf{I})^{-1}] = \sum_{j \ge 1} \frac{\lambda_j}{\lambda_j + \lambda}$ with $\{\lambda_j\}_{j \ge 1}$ eigenvalues of $h$.

[3] Again, $h$ is the operator with $h^{-\kappa} g = \sum_{j \ge 1} \lambda_j^{-\kappa} \langle f, \psi_j \rangle \psi_j$, where $\{\psi_j\}_{j \ge 1}$ are the eigenvectors of $h$.

$H^{\mathsf{CK}}$ breaks the curse of dimensionality by restricting the RKHS to 'local' functions. Similarly, [22] derived a decay rates in $n$ that do not depend on $d$ for a one-layer convolutional kernel. The key difference between Theorem 1 and [22] is that we obtain a non-asymptotic bound that is minmax optimal up to a constant multiplicative factor in both $d$ and $n$ (this can be showed for example by adapting the proof in Appendix B.6 in [7]) using a rigorous framework of source and capacity condition.

Theorem 1 and results of this type suffers from several limitations: 1) they are tight only in a minmax sense; 2) they do not provide comparisons for specific subclasses of functions; 3) in order to obtain the minmax rate, the regularization parameter $\lambda$ has to be carefully tuned to balance the bias and variance terms, which is in contrast to modern practice where often the model is trained until interpolation. This led several groups to consider instead the test error of KRR in a high-dimensional limit [11, 24, 38] and derive exact asymptotic predictions correct up to an additive vanishing constant for any $f_\star \in L^2$ (see Appendix C for more details).

Using the general framework in [38], we get the following result for $q, d$ large:

**Theorem 2** (Generalization error of KRR with $H^{\mathsf{CK}}$ in high-dimension (informal)). *Let $f_\star \in L^2(\mathcal{Q}^d, \mathrm{Loc}_q)$ and $h : \mathbb{R} \to \mathbb{R}$ verifying some 'genericity condition'. Then for $n = dq^{\mathsf{s}-1+\nu}$ with $0 < \nu < 1$, and $\lambda = O(1)$ (in particular $\lambda = 0$ works), we have*

$$\hat{f}_\lambda = \mathsf{P}_{\mathcal{E}_{\leq \mathsf{s}, \nu}} f_\star + o_q(1) \,, \tag{6}$$

*where $\mathsf{P}_{\mathcal{E}_{\leq \mathsf{s}, \nu}}$ is the projection on the span of $Y_S$ with either $|S| < \mathsf{s}$ and $S \in \mathcal{E}_{|S|}$ or $|S| = \mathsf{s}$ and $\gamma(S) \leq q(1 - q^{-\nu})$.*

See Appendix C.1 for a rigorous statement. In words, when $dq^{\mathsf{s}-1} \ll n \ll dq^{\mathsf{s}}$, KRR with $H^{\mathsf{CK}}$ only learns a degree-$\mathsf{s}$ polynomial approximation to $f_\star$.

On the other hand, when considering the standard inner-product kernel $H^{\mathsf{FC}}$ (FC) we get:

**Theorem 3** (Generalization error of KRR with $H^{\mathsf{FC}}$ in high-dimension (informal)). *Let $f_\star \in L^2(\mathcal{Q}^d)$ and $\tilde{h} : \mathbb{R} \to \mathbb{R}$ with some 'genericity condition'. Then for $d^{\mathsf{s}} \ll n \ll d^{\mathsf{s}+1}$ and $\lambda = O(1)$,*

$$\hat{f}_\lambda = \mathsf{P}_{\leq \mathsf{s}} f_\star + o_d(1) \,, \tag{7}$$

*where $\mathsf{P}_{\leq \mathsf{s}}$ is the projection on the subspace of degree-$\mathsf{s}$ polynomials.*

This theorem was proved in [24, 38]. Notice that Eq. (7) does not depend on the structure of $f_\star$. Hence, when $f_\star \in L^2(\mathcal{Q}^d, \mathrm{Loc}_q)$, Theorems 2 and 3 shows a clear statistical advantage of $H^{\mathsf{CK}}$ over $H^{\mathsf{FC}}$ when $q \ll d$ (and therefore of one-layer CNNs over fully-connected neural networks in the kernel regime).

## 2.3   Local average pooling and downsampling

In many applications such as object recognition, we expect the target function to depend mildly on the absolute spatial position of an object and to be stable under small shifts of the input. To take this local invariance into account, convolution layers are often followed by a pooling operation. Here we consider local average pooling on a segment of length $\omega$ and obtain the kernel

$$H_\omega^{\mathsf{CK}}(\boldsymbol{x}, \boldsymbol{y}) = \frac{1}{d\omega} \sum_{k \in [d]} \sum_{s, s' \in [\omega]} h\left( \langle \boldsymbol{x}_{(k+s)}, \boldsymbol{y}_{(k+s')} \rangle / q \right) \,. \tag{8}$$

Define $\mathcal{S}_\ell = \{ S \subseteq [q] : |S| = \ell \}$ as the collection of sets of size $\ell$. We further define an equivalence relation $\sim$ on $\mathcal{S}_\ell$: $S \sim S'$ if $S'$ is a translated subset of $S$ in $[q]$ (without cyclic convention). We denote $\mathcal{C}_\ell$ the quotient set of $\mathcal{A}_\ell$ under the equivalence relation $\sim$.

**Proposition 2** (Eigendecomposition of $H_\omega^{\mathsf{CK}}$). *Let $H_\omega^{\mathsf{CK}}$ be a convolutional kernel with local average pooling as defined in Eq. (8). Then $H_\omega^{\mathsf{CK}}$ admits the following eigendecomposition:*

$$H_\omega^{\mathsf{CK}}(\boldsymbol{x}, \boldsymbol{y}) = \omega \xi_{q,0} + \sum_{\ell=1}^{q} \sum_{S \in \mathcal{C}_\ell} \sum_{j \in [d]} \frac{\kappa_j r(S) \xi_{q,\ell}}{d} \cdot \psi_{j,S}(\boldsymbol{x}) \psi_{j,S}(\boldsymbol{y}) \,, \tag{9}$$

*where (denoting $k + S$ the translated set $S$ by $k$ positions with cyclic convention in $[d]$)*

$$\kappa_j = 1 + 2 \sum_{k=1}^{\omega-1} (1 - k/\omega) \cos\left( \frac{2\pi jk}{d} \right) \,, \qquad \psi_{j,S}(\boldsymbol{x}) = \frac{1}{\sqrt{d}} \sum_{k=1}^{d} e^{\frac{2i\pi jk}{d}} Y_{k+S}(\boldsymbol{x}) \,. \tag{10}$$

First notice that, as long as $\gcd(\omega, d) = 1$, the RKHS associated to $H^{\mathsf{CK}}$ contains the same set of functions as the RKHS of $H^{\mathsf{CK}}$, i.e., all local functions $L^2(\mathcal{Q}^d, \mathrm{Loc}_q)$. (There are $\gcd(\omega, d) - 1$ number of zero weights: $\kappa_j = 0$ for all $j \in [d-1]$ such that $d$ is a divisor of $j\omega$. See Appendix A.3 for details.) However $H^{\mathsf{CK}}$ will penalize different frequency components of the functions differently. Denote $f_j(\boldsymbol{x})$ the $j$-th component of the discrete Fourier transform of the function, i.e., $f_j(\boldsymbol{x}) = \frac{1}{\sqrt{d}} \sum_{k \in [d]} \rho_j^k f(t_k \cdot \boldsymbol{x})$ where $\rho_j = e^{2i\pi j/d}$ and $t_k \cdot \boldsymbol{x} = (x_{k+1}, \ldots, x_d, x_1, \ldots, x_k)$ is the cyclic shift by $k$ pixels. Then $H^{\mathsf{CK}}$ reweights the eigenspaces associated with $f_j(\boldsymbol{x})$ by a factor $\kappa_j$, promoting low-frequency components ($\kappa_j > 1$) and penalizing the high-frequencies ($\kappa_j < 1$). In words, *pooling biases the learning towards low-frequency functions*, which are stable by small shifts.

Let us focus on two special choices here: the pooling parameter $\omega = 1$ and $\omega = d$. When $\omega = 1$, $H_\omega^{\mathsf{CK}}$ reduces to $H^{\mathsf{CK}}$ ($\kappa_j = 1$ for all $j \in [d]$) which does not bias towards either low or high frequency components. When $\omega = d$, we denote such kernel $H_{\omega=d}^{\mathsf{CK}}$ by $H_{\mathsf{GP}}^{\mathsf{CK}}$ which corresponds to global average pooling. In this case, we have $\kappa_d = d$ and $\kappa_j = 0$ for $j < d$ which enforces exact invariance under the group of cyclic translations. More precisely, $H_{\mathsf{GP}}^{\mathsf{CK}}$ has RKHS that contains all cyclic q-local functions $f(\boldsymbol{x}) = \sum_{k \in [d]} g(\boldsymbol{x}_{(k)}) \in L^2(\mathcal{Q}^d, \mathrm{CycLoc}_q)$ (c.f. Eq. (CYC-LOC)).

We obtain a bound on the test error of KRR with $H_\omega^{\mathsf{CK}}$ similar to Theorem 1, but with $d$ replaced by an effective dimension $d^{\mathrm{eff}}$.

**Theorem 4** (Generalization of KRR with average pooling (fixed $d, q$)). *Assume that $h : \mathbb{R} \to \mathbb{R}$ has $\xi_{q,0} = 0$ and satisfies (A1). Further assume (A2') that $\|(H_\omega^{\mathsf{CK}}/\omega)^{-\beta/2} f_\star\|_{L^2} \leq B$. Define $d^{\mathrm{eff}} = \sum_{j \in [d]:\kappa_j > 0} (\kappa_j/\omega)^{1/\alpha}$. Then there exists $C_1, C_2, C_3 > 0$ constants independent of $d$, such that for $n \geq C_1 \max(\|f_\star\|_{L^\infty}^2, d_{\mathrm{eff}})$ and setting $\lambda_\star = C_2 (d_{\mathrm{eff}}/n)^{\frac{\alpha}{\alpha\beta+1}}$, we get*

$$\mathbb{E}_{\boldsymbol{\varepsilon}}\{R(f_\star, \hat{f}_{\lambda_\star})\} \leq C_3 \left(\frac{d_{\mathrm{eff}}}{n}\right)^{\frac{\alpha\beta}{\alpha\beta+1}}. \tag{11}$$

By Jensen's inequality, we have $d^{\mathrm{eff}} \leq d/\omega^{1/\alpha}$. In particular, for global pooling, $d^{\mathrm{eff}} = 1$ and the bound (11) does not depend on $d$ at all. Adding average pooling improve by a factor $\omega^{1/\alpha}$ the upper bound on the sample complexity for fitting low-frequency functions. Can we confirm this statistical advantage using the predictions for KRR in high dimension? Consider first the case of global pooling:

**Theorem 5** (Generalization of KRR with $H_{\mathsf{GP}}^{\mathsf{CK}}$ in high-dimension (informal)). *Let $f_\star \in L^2(\mathcal{Q}^d, \mathrm{CycLoc}_q)$ and $h : \mathbb{R} \to \mathbb{R}$ verifying some 'genericity condition'. Then for $n = q^{\mathsf{s}-1+\nu}$ with $0 < \nu < 1$, and $\lambda = O(1)$, we have ($\mathsf{P}_{\mathcal{E}_{\leq \mathsf{s},\nu}}$ is defined as in Theorem 2)*

$$\hat{f}_\lambda = \mathsf{P}_{\mathcal{E}_{\leq \mathsf{s},\nu}} f_\star + o_q(1). \tag{12}$$

Hence, global average pooling results in an improvement by a factor $d$ in statistical efficiency when fitting cyclic local functions, compared to $H^{\mathsf{CK}}$. This improvement was already noticed in [7, 39] but in the case of $q = d$ (fully connected neural networks).

For $\omega < d$, a direct application of the theorems in [38] is more challenging because of the mixing of eigenvalues. In this case, a modification of [38], where eigenvalues are not necessary ordered anymore would apply. However, for simplicity, we present in Appendix C.1 a simplified kernel with non-overlapping local pooling which we believe captures the statistical behavior of local pooling. In this case, we show that Theorem 5 holds with $n = (d/\omega) \cdot q^{\mathsf{s}-1+\nu}$, which interpolates between Theorem 2 ($\omega = 1$) and Theorem 5 ($\omega = d$).

**Downsampling:** Often pooling is associated with a downsampling operation, which subsample one every $\Delta$ output coordinates. In Appendix A.4, we characterize the eigendecomposition of $H_{\omega,\Delta}^{\mathsf{CK}}$ (Proposition 4) and prove for the popular choice $\omega = \Delta$, that downsampling does not modify the cyclic invariant subspace $j = d$ (Proposition 5). More generally, we conjecture and check numerically that downsampling with $\Delta \leq \omega$ leaves the low-frequency eigenspaces approximately unchanged. In particular, the statistical complexity of learning low-frequency functions is not modified by downsampling operation in the one-layer case (while downsampling is potentially beneficial in further layers).

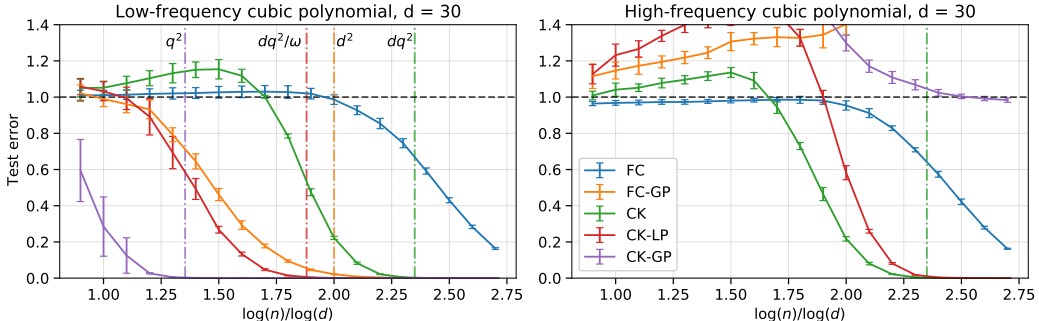

Figure 1: Learning low-frequency (left) and high-frequency (right) cubic polynomials over the hypercube $d = 30$, using KRR with $H^{\mathsf{FC}}$ (FC), $H^{\mathsf{FC}}_{\mathsf{GP}}$ (FC-GP), $H^{\mathsf{CK}}$ (CK), $H^{\mathsf{CK}}_{\omega}$ (CK-LP) and $H^{\mathsf{CK}}_{\mathsf{GP}}$ (CK-GP), and regularization parameter $\lambda = 0^+$. We report the average and the standard deviation of the test error over 5 realizations, against the sample size $n$.

## 3   Numerical simulations

In order to check our theoretical predictions, we perform a simple numerical experiment on simulated data. We take $\boldsymbol{x} \sim \mathrm{Unif}(\mathscr{Q}^d)$ with $d = 30$, and consider two target functions:

$$f_{\mathsf{LF},3}(\boldsymbol{x}) = \frac{1}{\sqrt{d}} \sum_{i \in [d]} x_i x_{i+1} x_{i+2} \,, \qquad f_{\mathsf{HF},3}(\boldsymbol{x}) = \frac{1}{\sqrt{d}} \sum_{i \in [d]} (-1)^i \cdot x_i x_{i+1} x_{i+2} \,. \tag{13}$$

Here $f_{\mathsf{LF},3}$ is a cyclic-invariant local polynomial ($f_{\mathsf{LF},3}$ is 'low-frequency'). The function $f_{\mathsf{HF},3}$ is a high-frequency local polynomial, and is orthogonal to the space of cyclic invariant functions. On these target functions, we compare the test error of kernel ridge regression with 5 different kernels: a standard inner-product kernel $H^{\mathsf{FC}}(\boldsymbol{x}, \boldsymbol{y}) = h(\langle \boldsymbol{x}, \boldsymbol{y} \rangle / d)$; a cyclic invariant kernel $H^{\mathsf{FC}}_{\mathsf{GP}}(\boldsymbol{x}, \boldsymbol{y})$ (convolutional kernel with global pooling and full-size patches $q = d$); a convolutional kernel $H^{\mathsf{CK}}$ with patch size $q = 10$; a convolutional kernel with local pooling $H^{\mathsf{CK}}_{\omega}$ with $q = 10$ and $\omega = 5$; and a convolutional kernel with global pooling $H^{\mathsf{CK}}_{\mathsf{GP}}$ with $q = 10$. In all these kernels, we choose a common $h(t) = \sum_{i \in [5]} 0.2 * t^i$ which is a degree 5-polynomial.

In Figure 1, we report the test errors of fitting $f_{\mathsf{LF},3}$ (left) and $f_{\mathsf{HF},3}$ (right) using kernel ridge regression with these 5 kernels. We choose a small regularization parameter $\lambda = 10^{-6}$, and the noise level $\sigma_{\varepsilon} = 0$. The curves are averaged over 5 independent instances and the error bar stands for the standard deviation of these instances. The results match well our theoretical predictions. For the function $f_{\mathsf{LF},3}$, the sample sizes required to achieve vanishing test errors are ordered as $H^{\mathsf{CK}}_{\mathsf{GP}} < H^{\mathsf{CK}}_{\omega} < H^{\mathsf{CK}} < H^{\mathsf{FC}}_{\mathsf{GP}} < H^{\mathsf{FC}}$ and are around the predicted thresholds $q^2 < dq^2/\omega < d^2 < dq^2 < d^3$ respectively. Next we look at the test error of fitting the high frequency local function $f_{\mathsf{HF},3}$. The test errors of $H^{\mathsf{CK}}$ and $H^{\mathsf{FC}}$ are the same for $f_{\mathsf{HF},3}$ and $f_{\mathsf{LF},3}$: this is because these kernels do not have bias towards either high-frequency or low-frequency functions. The kernel $H^{\mathsf{CK}}_{\omega}$ perform worse on $f_{\mathsf{HF},3}$ than on $f_{\mathsf{LF},3}$: this is because the eigenspaces of $H^{\mathsf{CK}}_{\omega}$ are biased towards low-frequency polynomials. The kernels $H^{\mathsf{CK}}_{\mathsf{GP}}$ and $H^{\mathsf{FC}}_{\mathsf{GP}}$ do not fit $f_{\mathsf{HF},3}$ at all (test error greater than or equal to 1): this is because the RKHS of these two kernels only contain cyclic polynomials, but $f_{\mathsf{HF},3}$ is orthogonal to the space of cyclic polynomials.

## 4   Discussion and Future Work

In this paper, we characterized in a stylized setting how convolution, average pooling and downsampling operations modify the RKHS, by restricting it to $q$-local functions and then biasing the RKHS towards low-frequency components. We quantified precisely the gain in statistical efficiency of KRR using these operations. Beyond illustrating the 'RKHS engineering' of image-like function classes, these results can further provide intuition and a rigorous foundation for convolution and pooling operations in kernels and CNNs. A natural extension would be to study the multilayer convolutional kernels in details and consider other pooling operations such as max-pooling. Another important question is how anisotropy of the data impacts the results of this paper: in particular, it was shown that pre-processing (whitening of the patches) greatly improves the performance of convolutional kernels [6, 46]. A more challenging question is to study how training and feature learning can further improve the performance of CNNs outside the kernel regime.

## Acknowledgement

S.M. is supported by NSF grant DMS-2210827. T.M. is supported by NSF through award DMS-2031883 and the Simons Foundation through Award 814639 for the Collaboration on the Theoretical Foundations of Deep Learning. T.M. also acknowledge the NSF grant CCF-2006489 and the ONR grant N00014-18-1-2729.

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
