# Contents

# A  Details from the main text

## A.1  Notations

For a positive integer, we denote by $[n]$ the set $\{1, 2, \ldots, n\}$. For vectors $\boldsymbol{u}, \boldsymbol{v} \in \mathbb{R}^d$, we denote $\langle \boldsymbol{u}, \boldsymbol{v} \rangle = u_1 v_1 + \ldots + u_d v_d$ their scalar product, and $\|\boldsymbol{u}\|_2 = \langle \boldsymbol{u}, \boldsymbol{u} \rangle^{1/2}$ the $\ell_2$ norm. Given a matrix $\boldsymbol{A} \in \mathbb{R}^{n \times m}$, we denote $\|\boldsymbol{A}\|_{\text{op}} = \max_{\|\boldsymbol{u}\|_2 = 1} \|\boldsymbol{A}\boldsymbol{u}\|_2$ its operator norm and by $\|\boldsymbol{A}\|_F = \left( \sum_{i,j} A_{ij}^2 \right)^{1/2}$ its Frobenius norm. If $\boldsymbol{A} \in \mathbb{R}^{n \times n}$ is a square matrix, the trace of $\boldsymbol{A}$ is denoted by $\text{Tr}(\boldsymbol{A}) = \sum_{i \in [n]} A_{ii}$.

We use $O_d(\,\cdot\,)$ (resp. $o_d(\,\cdot\,)$) for the standard big-O (resp. little-o) relations, where the subscript $d$ emphasizes the asymptotic variable. Furthermore, we write $f = \Omega_d(g)$ if $g(d) = O_d(f(d))$, and $f = \omega_d(g)$ if $g(d) = o_d(f(d))$. Finally, $f = \Theta_d(g)$ if we have both $f = O_d(g)$ and $f = \Omega_d(g)$.

We use $O_{d,\mathbb{P}}(\,\cdot\,)$ (resp. $o_{d,\mathbb{P}}(\,\cdot\,)$) the big-O (resp. little-o) in probability relations. Namely, for $h_1(d)$ and $h_2(d)$ two sequences of random variables, $h_1(d) = O_{d,\mathbb{P}}(h_2(d))$ if for any $\varepsilon > 0$, there exists $C_\varepsilon > 0$ and $d_\varepsilon \in \mathbb{Z}_{>0}$, such that

$$\mathbb{P}(|h_1(d)/h_2(d)| > C_\varepsilon) \leq \varepsilon, \qquad \forall d \geq d_\varepsilon,$$

and respectively: $h_1(d) = o_{d,\mathbb{P}}(h_2(d))$, if $h_1(d)/h_2(d)$ converges to 0 in probability. Similarly, we will denote $h_1(d) = \Omega_{d,\mathbb{P}}(h_2(d))$ if $h_2(d) = O_{d,\mathbb{P}}(h_1(d))$, and $h_1(d) = \omega_{d,\mathbb{P}}(h_2(d))$ if $h_2(d) = o_{d,\mathbb{P}}(h_1(d))$. Finally, $h_1(d) = \Theta_{d,\mathbb{P}}(h_2(d))$ if we have both $h_1(d) = O_{d,\mathbb{P}}(h_2(d))$ and $h_1(d) = \Omega_{d,\mathbb{P}}(h_2(d))$.

## A.2  Convolutional neural tangent kernel

In this section, we justify the expression of the convolutional neural tangent kernel $H_{\boldsymbol{w},\Delta}^{\text{CK}}$ (CK-AP-DS), obtained as the tangent kernel of a neural network composed of a one convolution layer followed by local average pooling and downsampling (CNN-AP-DS).

**Proposition 3.** *Let $\sigma \in \mathcal{C}^1(\mathbb{R})$ be an activation function. Consider the following one-layer convolutional neural network with $\omega$-local average pooling and $\Delta$-downsampling:*

$$f_N^{\text{CNN}}(\boldsymbol{x}; \boldsymbol{\Theta}) = \sum_{i \in [N]} \sum_{k \in [d/\Delta]} a_{ik} \sum_{s \in [\omega]} \sigma\big( \langle \boldsymbol{w}_i, \boldsymbol{x}_{(k\Delta+s)} \rangle \big). \tag{14}$$

*Let $a_{ik}^0 \sim_{\text{i.i.d.}} \mathsf{N}(0,1)$ and $\sqrt{q}\boldsymbol{w}_i^0 \sim_{\text{i.i.d.}} \text{Unif}(\mathcal{Q}^q)$ independently, and $\boldsymbol{\Theta}^0 = \{(a_{ik}^0)_{i \in [N], k \in [d/\Delta]}, (\boldsymbol{w}_i^0)_{i \in [N]}\}$. Then there exists $h : [-1, 1] \to \mathbb{R}$, such that for any $\boldsymbol{x}, \boldsymbol{y} \in \mathcal{Q}^d$, we have almost surely*

$$\lim_{N \to \infty} \big\langle \nabla_{\boldsymbol{\Theta}} f_N^{\text{CNN}}(\boldsymbol{x}; \boldsymbol{\Theta}^0), \nabla_{\boldsymbol{\Theta}} f_N^{\text{CNN}}(\boldsymbol{y}; \boldsymbol{\Theta}^0) \big\rangle / N = \sum_{k \in [d/\Delta]} \sum_{s, s' \in [\omega]} h\big( \langle \boldsymbol{x}_{(k\Delta+s)}, \boldsymbol{y}_{(k\Delta+s')} \rangle / q \big). \tag{15}$$

*Proof of Proposition 3.* For $\boldsymbol{u}, \boldsymbol{v} \in \mathcal{Q}^q$, define

$$h^{(1)}(\langle \boldsymbol{u}, \boldsymbol{v} \rangle / q) = \mathbb{E}_{\boldsymbol{w} \sim \text{Unif}(\mathcal{Q}^q)} \big[ \sigma(\langle \boldsymbol{u}, \boldsymbol{w} \rangle / \sqrt{q}) \sigma(\langle \boldsymbol{v}, \boldsymbol{w} \rangle / \sqrt{q}) \big],$$

$$h^{(2)}(\langle \boldsymbol{u}, \boldsymbol{v} \rangle / q) = \mathbb{E}_{\boldsymbol{w} \sim \text{Unif}(\mathcal{Q}^q)} \big[ \sigma'(\langle \boldsymbol{u}, \boldsymbol{w} \rangle / \sqrt{q}) \sigma'(\langle \boldsymbol{v}, \boldsymbol{w} \rangle / \sqrt{q}) \langle \boldsymbol{u}, \boldsymbol{v} \rangle \big] / q.$$

The functions $h^{(1)}, h^{(2)}$ are well defined (the RHS only depend on the inner product $\langle \boldsymbol{u}, \boldsymbol{v} \rangle$) and can be extended to functions $h^{(1)}, h^{(2)} : [-1, 1] \to \mathbb{R}$.

Computing the derivative of the convolutional neural network with respect to $\boldsymbol{a} = (a_{ik}^0)_{i \in [N], k \in [d/\Delta]}$, we have

$$\frac{1}{N} \big\langle \nabla_{\boldsymbol{a}} f_N^{\text{CNN}}(\boldsymbol{x}; \boldsymbol{\Theta}^0), \nabla_{\boldsymbol{a}} f_N^{\text{CNN}}(\boldsymbol{y}; \boldsymbol{\Theta}^0) \big\rangle$$

$$= \sum_{k \in [d/\Delta]} \sum_{s, s' \in [\omega]} \frac{1}{N} \sum_{i \in [N]} \sigma\big( \langle \boldsymbol{w}_i^0, \boldsymbol{x}_{(k\Delta+s)} \rangle \big) \sigma\big( \langle \boldsymbol{w}_i^0, \boldsymbol{x}_{(k\Delta+s')} \rangle \big).$$

Hence by law of large number, we have almost surely

$$\lim_{N \to \infty} \frac{1}{N} \big\langle \nabla_{\boldsymbol{a}} f_N^{\text{CNN}}(\boldsymbol{x}; \boldsymbol{\Theta}^0), \nabla_{\boldsymbol{a}} f_N^{\text{CNN}}(\boldsymbol{y}; \boldsymbol{\Theta}^0) \big\rangle = \sum_{k \in [d/\Delta]} \sum_{s, s' \in [\omega]} h^{(1)}\big( \langle \boldsymbol{x}_{(k\Delta+s)}, \boldsymbol{y}_{(k\Delta+s')} \rangle / q \big).$$

Similarly, computing the derivative with respect to $\sqrt{q}\boldsymbol{W} = (\sqrt{q}\boldsymbol{w}_i^0)_{i\in[N]}$ gives

$$\frac{1}{N}\langle \nabla_{\boldsymbol{W}} f_N^{\mathsf{CNN}}(\boldsymbol{x};\boldsymbol{\Theta}^0), \nabla_{\boldsymbol{W}} f_N^{\mathsf{CNN}}(\boldsymbol{y};\boldsymbol{\Theta}^0)\rangle$$

$$= \sum_{k,k'\in[d/\Delta]} \sum_{s,s'\in[\omega]} \frac{1}{N}\sum_{i\in[N]} a_{ik}a_{ik'}\sigma'\big(\langle \boldsymbol{w}_i^0, \boldsymbol{x}_{(k\Delta+s)}\rangle\big)\sigma'\big(\langle \boldsymbol{w}_i^0, \boldsymbol{x}_{(k'\Delta+s')}\rangle\big)\frac{\langle \boldsymbol{x}_{(k\Delta+s)}, \boldsymbol{x}_{(k'\Delta+s')}\rangle}{q}\,.$$

By law of large number, using that $a_{ik}$ and $a_{ik'}$ are independent of mean zero and variance 1, we get almost surely

$$\lim_{N\to\infty} \frac{1}{N}\langle \nabla_{\boldsymbol{W}} f_N^{\mathsf{CNN}}(\boldsymbol{x};\boldsymbol{\Theta}^0), \nabla_{\boldsymbol{W}} f_N^{\mathsf{CNN}}(\boldsymbol{y};\boldsymbol{\Theta}^0)\rangle = \sum_{k\in[d/\Delta]} \sum_{s,s'\in[\omega]} h^{(2)}\big(\langle \boldsymbol{x}_{(k\Delta+s)}, \boldsymbol{y}_{(k\Delta+s')}\rangle/q\big)\,.$$

Taking $h = h^{(1)} + h^{(2)}$ concludes the proof. $\qquad\square$

### A.3 Local average pooling operation

Consider a function $f \in L^2(\mathscr{Q}^d)$: we can decompose it as

$$f(\boldsymbol{x}) = \frac{1}{\sqrt{d}}\sum_{j\in[d]} f_j(\boldsymbol{x})\,, \tag{16}$$

$$f_j(\boldsymbol{x}) = \frac{1}{\sqrt{d}}\sum_{k\in[d]} \rho_j^k f(t_k \cdot \boldsymbol{x})\,, \tag{17}$$

where $\rho_j = e^{\frac{2i\pi j}{d}}$ and $t_k \cdot \boldsymbol{x} = (x_{k+1}, \ldots, x_d, x_1, \ldots, x_k)$ is the cyclic shift of $\boldsymbol{x}$ by $k$ pixels. We can think about $f_j(\boldsymbol{x})$ as the $j$-th component of the discrete Fourier transform of the function $f(\boldsymbol{x})$ seen as a $d$-dimensional vector $\{f(t_k \cdot \boldsymbol{x})\}_{k\in[d]}$ for any $\boldsymbol{x} \in \mathscr{Q}^d$.

Notice furthermore that if $f$ is a local function, i.e., $f$ can be decomposed as a sum of functions on patches $f(\boldsymbol{x}) = \sum_{k\in[d]} g_k(\boldsymbol{x}_{(k)})$, then we can write

$$f_j(\boldsymbol{x}) = \frac{1}{\sqrt{d}}\sum_{k\in[d]} \rho_j^k f(t_k \cdot \boldsymbol{x}) = \frac{1}{\sqrt{d}}\sum_{k,u\in[d]} \rho_j^k g_u(\boldsymbol{x}_{(u+k)}) = \frac{1}{\sqrt{d}}\sum_{k\in[d]} \rho_j^k \tilde{g}_j(\boldsymbol{x}_{(k)})\,,$$

where we denoted ($\boldsymbol{v} \in \mathscr{Q}^q$)

$$\tilde{g}_j(\boldsymbol{v}) = \sum_{u\in[d]} \rho_j^{-u} g_u(\boldsymbol{v})\,.$$

In particular, decomposing $\tilde{g}_j$ in the Fourier basis, we get (denoting $c_S = \langle \tilde{g}_j, Y_S\rangle_{L^2}$),

$$f_j(\boldsymbol{x}) = \frac{1}{\sqrt{d}}\sum_{k\in[d]} \rho_j^k \tilde{g}_j(\boldsymbol{x}_{(k)}) = \sum_{S\subseteq[q]} c_S \cdot \frac{1}{\sqrt{d}}\sum_{k\in[d]} \rho_j^k Y_{k+S}(\boldsymbol{x})\,,$$

which shows that the $j$-th frequency component $f_j$ is in the span of $\{Y_{j,S}\}_{S\subseteq[q]}$. In particular, applying average pooling operation in the kernel will reweight this eigenspace by a factor $\kappa_j$.

Let us further comment on the values of $\kappa_j$. First, we have

$$\kappa_j = \sum_{k=-\omega}^{\omega} (1 - k/\omega)\rho_j^k\,.$$

In particular, the maximal eigenvalue is attained at $j = d$ with $\kappa_d = \omega$, which corresponds to the subspace of cyclic invariant functions. Furthermore, $\kappa_j = 0$ if and only if $d$ is a divisor of $j\omega$ for $j \le d-1$, i.e., $j$ is a multilple of $\gcd(\omega, d)$. There are $\gcd(\omega, d) - 1$ such zero eigenvalues.

In convolutional kernels, a weighted average is often preferred to local average pooling [6, 35, 36]: in that case we consider $\tau : \mathbb{R} \to \mathbb{R}$ and obtain the kernel

$$H_\tau^{\mathsf{CK}}(\boldsymbol{x}, \boldsymbol{y}) = \frac{1}{d}\sum_{k,s,s'\in[d]} \tau(d(s))\tau(d(s'))h\big(\langle \boldsymbol{x}_{(k+s)}, \boldsymbol{y}_{(k+s')}\rangle/q\big)\,,$$

where $d(s) = \min(s, d-s)$ (the distance between $k+s$ and $k$ on $[d]$ with cyclic convention). Note that $H_\tau^{\mathsf{CK}}$ has the same eigendecomposition as $H_\omega^{\mathsf{CK}}$ but with different weights $\kappa_j$.

A popular choice for $\tau$ is the Gaussian filter $\tau(x) = \frac{1}{\sqrt{2\pi}\sigma} e^{-\frac{x^2}{2\sigma^2}}$. In Figure 2, we compare the eigenvalues $\kappa_j$ for local average pooling and Gaussian filter with different value of $\omega$ and $\sigma^2$. Note that the eigenvalue decay controls how much high-frequencies are penalized: faster decay induces heavier penalty on the high-frequency components.

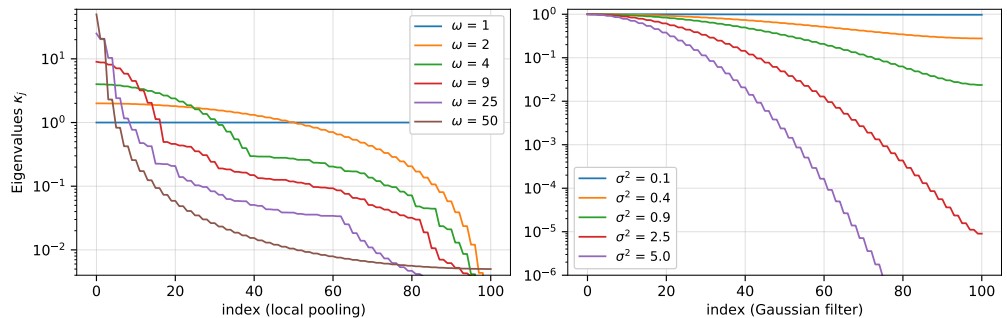

Figure 2: Decay of the weights $\kappa_j$ for different lenfths $\omega$ for local average pooling (on the left) and bandwidths $\sigma^2$ for pooling with Gaussian filter (on the right), for $d = 101$.

### A.4 Downsampling operation

As mentioned in the main text, a downsampling operation is often added after pooling. The kernel is given by

$$H_{\omega,\Delta}^{\mathsf{CK}}(\boldsymbol{x}, \boldsymbol{y}) = \frac{\Delta}{d\omega} \sum_{k \in [d/\Delta]} \sum_{s,s' \in [\omega]} h\left(\langle \boldsymbol{x}_{(k\Delta+s)}, \boldsymbol{y}_{(k\Delta+s')}\rangle / q\right). \tag{18}$$

Let us introduce the family $\{\boldsymbol{M}^r\}_{r \in [q]}$ of block-circulant matrices defined by

$$M_{ij}^r = \frac{\Delta}{\omega(q+1-r)} \left| \left\{(k, s, s', t) \in \mathcal{I}_{\omega,\Delta,r} : k\Delta + s + t \equiv i[d], k\Delta + s' + t \equiv j[d]\right\}\right|, \tag{19}$$

where we introduced the set of indices

$$\mathcal{I}_{\omega,\Delta,r} = \left\{(k, s, s', t) : k \in [d/\Delta], s, s' \in [\omega], 0 \leq t \leq q - r\right\}. \tag{20}$$

We can now state the eigendecomposition of $H_{\omega,\Delta}^{\mathsf{CK}}$ in terms of the eigenvalues and eigenvectors of the matrices $\{\boldsymbol{M}^r\}_{r \in [q]}$.

**Proposition 4** (Eigendecomposition of $H_{\omega,\Delta}^{\mathsf{CK}}$). *Let $H_{\omega,\Delta}^{\mathsf{CK}}$ be a convolutional kernel with local average pooling and downsampling, as defined in Eq. (18). Then $H_{\omega,\Delta}^{\mathsf{CK}}$ admits the following eigendecomposition:*

$$H_\omega^{\mathsf{CK}}(\boldsymbol{x}, \boldsymbol{y}) = \omega\xi_{q,0} + \sum_{\ell=1}^q \sum_{S \in \mathcal{C}_\ell} \sum_{j \in [d]} \frac{\xi_{q,\ell} r(S) \kappa_j^S}{d} \cdot \psi_{j,S}(\boldsymbol{x}) \psi_{j,S}(\boldsymbol{y}), \tag{21}$$

*where $\psi_{j,S}^\Delta(\boldsymbol{x}) = \sum_{k=1}^d v_{j,k}^S Y_{k+S}(\boldsymbol{x})$ with $\{\kappa_j^S, \boldsymbol{v}_j^S\}_{j \in [d]}$ eigenvalues and eigenvectors of $\boldsymbol{M}^{\gamma(S)}$.*

Let us make a few comments on these matrices $\boldsymbol{M}^{\gamma(S)}$. First because they only depend on $S$ through the diameter $\gamma(S)$, the eigenvalues and eigenvectors $\{\kappa_j^S, \boldsymbol{v}_j^S\}_{j \in [d]}$ only depend on $\gamma(S)$. Second, we see that $M_{(i+\Delta)(j+\Delta)}^{\gamma(S)} = M_{ij}^{\gamma(S)}$ and $M_{ij}^{\gamma(S)} = 0$ if $d(i, j) \geq \omega$, where $d(i, j) = \min(|i - j|, d - |i - j|)$ (i.e., the distance between $i$ and $j$ on the torus $[d]$). In words $\boldsymbol{M}^{\gamma(S)}$ is a

symmetric block-circulant matrix with non-zero elements on a band of size $\omega - 1$ on the left and right of the diagonal, and on the upper-right and lower-left corners. Furthermore, notice that

$$\mathrm{Tr}(\boldsymbol{M}^{\gamma(S)}) = \frac{\Delta}{d\omega r(S)}\left|\left\{(k,s,t): k \in [d/\Delta], s \in [\omega], 0 \le t \le q - \gamma(S)\right\}\right| = 1,$$

which is independent of $\omega, \Delta, \gamma(S)$ and justify the chosen normalization. In particular, this implies that (for $\xi_{q,0} = 0$)

$$\mathrm{Tr}(\mathbb{H}_{\omega,\Delta}^{\mathsf{CK}}) := \mathbb{E}_{\boldsymbol{x}}\{H_{\omega,\Delta}^{\mathsf{CK}}(\boldsymbol{x},\boldsymbol{x})\} = \sum_{\ell \in [q]} \xi_{q,\ell} \sum_{S \in \mathcal{C}_\ell} r(S) = \sum_{\ell \in [q]} \xi_{q,\ell} B(\mathcal{Q}^q; \ell) = h(1), \qquad (22)$$

is also independent of the parameters $(q, \omega, \Delta)$.

**Example 1.** Take $\Delta = 3$, $\omega = 5$, $q = 11$, then

$$\boldsymbol{M}^1 = \frac{3}{50}\left(\begin{array}{ccc|ccc|ccc|c}
18 & 15 & 11 & 7 & 4 & 0 & & & & \\
15 & 19 & 15 & 11 & 8 & 4 & 0 & & & \cdots \\
11 & 15 & 18 & 14 & 11 & 7 & 3 & 0 & & \\
\hline
7 & 11 & 14 & 18 & 15 & 11 & 7 & 3 & 0 & \\
4 & 8 & 11 & 15 & 19 & 15 & 11 & 8 & 4 & \cdots \\
0 & 4 & 7 & 11 & 15 & 18 & 14 & 11 & 7 & \\
\hline
 & 0 & 3 & & & & & & & \\
 & & 0 & & \vdots & & & \ddots & & \cdots \\
\end{array}\right),$$

and

$$\boldsymbol{M}^4 = \frac{3}{35}\left(\begin{array}{ccc|ccc|ccc|c}
13 & 11 & 8 & 5 & 3 & 0 & & & & \\
11 & 14 & 11 & 8 & 6 & 3 & 0 & & & \\
8 & 11 & 13 & 10 & 8 & 5 & 2 & 0 & & \\
\hline
5 & 8 & 10 & & & & & & & \\
3 & 6 & 8 & & \ddots & & & \cdots & & \\
0 & 3 & 5 & & & & & & & \\
\hline
 & \vdots & & & \vdots & & & \ddots & & \\
\end{array}\right).$$

**Remark 1.** Symmetric block-circulant matrices can be easily diagonalized as follows. Consider $\boldsymbol{M} = \mathsf{Circulant}(\boldsymbol{B}_1, \boldsymbol{B}_2, \ldots, \boldsymbol{B}_m)$ where $\boldsymbol{B}_k \in \mathbb{R}^{\Delta \times \Delta}$, $\boldsymbol{B}_1^{\mathsf{T}} = \boldsymbol{B}_1$ and $\boldsymbol{B}_{2+k} = \boldsymbol{B}_{m-k}^{\mathsf{T}}$ for $k = 0, \ldots, m-2$. Denote $\rho_j = e^{2i\pi j/m}$ and $\gamma_j(\boldsymbol{v}) = [\boldsymbol{v}, \rho_j \boldsymbol{v}, \cdots, \rho_j^{m-1}\boldsymbol{v}]/\sqrt{m} \in \mathbb{R}^{m\Delta}$ for any $\boldsymbol{v} \in \mathbb{R}^{\Delta}$. Introduce for $j = 0, \ldots, m-1$, the matrix $\boldsymbol{H}_j \in \mathbb{R}^{\Delta \times \Delta}$ given by

$$\boldsymbol{H}_j = \boldsymbol{B}_1 + \rho_j \boldsymbol{B}_2 + \ldots + \rho_j^{m-1}\boldsymbol{B}_m. \qquad (23)$$

The matrix $\boldsymbol{H}_j$ is Hermitian and we denote $(\lambda_{j,s})_{s \in [\Delta]}$ and $(\boldsymbol{v}_{j,s})_{s \in [\Delta]}$ its eigenvalues and eigenvectors. Then the eigenvalues and eigenvectors of $\boldsymbol{M}$ are given by $\{\lambda_{j,s}\}_{j \in [m], s \in [\Delta]}$ and $\{\gamma_j(\boldsymbol{v}_{j,s})\}_{j \in [m], s \in [\Delta]}$.

In particular, if $\Delta = 1$ and $\boldsymbol{M} = \mathsf{Circulant}(b_1, b_2, \ldots, b_m)$ is a circulant matrix, then the eigenvalues are simply given by

$$\lambda_j = b_1 + \rho_j b_2 + \ldots + \rho_j^{m-1} b_m,$$

and eigenvectors $\boldsymbol{v}_j = [1, \rho_j, \cdots, \rho_j^{m-1}]/\sqrt{m}$.

Here we will focus on the impact of downsampling for single-layer convolutional kernels. We expect the downsampling operation to have a much more important role for the next layers: for example, increasing the scale of interactions or reducing the dimensionality of the pixel space.

We will argue below that adding a downsampling operation after local pooling leaves the low-frequency components approximately unchanged (while potentially modifying the high-frequency eigenspaces). We consider $\Delta \le \omega$: for $\Delta > \omega$, some basis functions $Y_S$ with $S \in \mathcal{E}_\ell$ are in the null space of $H_{\omega,\Delta}^{\mathsf{CK}}$, which impact all frequencies.

To emphasize the dependency on $\omega, \Delta$, denote $M_{\omega,\Delta}^r$ the matrix (19). We will study the change in the matrix $M_{\omega,1}^r$ when adding downsampling $\Delta$, and consider

$$M_{\omega,\Delta}^r = M_{\omega,1}^r + A_{\omega,\Delta}^r, \tag{24}$$

where we denote $A_{\omega,\Delta}^r = M_{\omega,\Delta}^r - M_{\omega,1}^r$. Notice that $A_{\omega,\Delta}^r$ is a symmetric block-circulant matrix. Therefore, from Remark 1, the eigenvectors of $A_{\omega,\Delta}^r$ are given by $\{\gamma_j(v_{j,s})\}_{j\in[m],s\in[\Delta]}$ where $d = m\Delta$ and $\gamma_j(v_{j,s}) = [v_{j,s}, \rho_{m,j}v_{j,s}, \ldots, \rho_{m,j}^{m-1}v_{j,s}]$ with $\rho_{m,j} = e^{\frac{2i\pi j}{m}}$ and $(v_{j,s})_{s\in[\Delta]}$ eigenvectors of $H_j$ (23). The eigenvectors of $M_{\omega,1}^r$ are given by $u_t = [1, \rho_{d,t}, \cdots, \rho_{d,t}^{d-1}]/\sqrt{d}$ with $\rho_{d,t} = e^{\frac{2i\pi t}{d}}$. Notice that

$$\langle u_t^*, \gamma_j(v_{j,s})\rangle = \frac{1}{\sqrt{dm}} \sum_{k\in[m]} \sum_{u\in[\Delta]} \rho_{m,j}^{k-1} \rho_{d,t}^{-(k-1)\Delta-(t-1)}(v_{j,s})_u$$

$$= \frac{1}{\sqrt{dm}}\left(\sum_{u\in[\Delta]} \rho_{d,t}^{-(u-1)}(v_{j,s})_u\right) \cdot \sum_{k\in[m]}\left(\rho_{m,j}\rho_{d,t}^{-\Delta}\right)^{k-1},$$

which is 0 except when $t \equiv j[m]$. Hence, we see that $A_{\omega,\Delta}^r$ in Eq. (24) only modify the eigenspaces of $M_{\omega,1}^r$ as follows: the eigendirections $\{\gamma_j(v_{j,s})\}_{j\in[m],s\in[\Delta]}$ coming from $H_j$ (23) only modify the eigenspaces of $M_{\omega,1}^r$ spanned by $\{u_{am+j}\}_{a=0,\ldots,\Delta-1}$.

For simplicity, we will focus on the popular choice $\Delta = \omega$. Furthermore, we will only look at the impact of the eigenvalues $H_0$ on the eigenspace spanned by $\{u_{am}\}_{a=0,\ldots,\Delta-1}$, which contain the cyclic invariant direction. We show below that $H_0 = 0$ and therefore $A_{\omega,\omega}^r$ does not modify the cyclic invariant eigenspace of $M_{\omega,1}^r$:

**Proposition 5.** *Consider $d = m\omega$ and the symmetric block-circulant matrix $A_{\omega,\omega}^r = M_{\omega,\omega}^r - M_{\omega,1}^r$. Denote $A_{\omega,\omega}^r = \mathsf{Circulant}(B_1, B_2, \ldots, B_m)$ and*

$$H_0 = B_1 + \ldots + B_m.$$

*We have the following properties:*

*(a) If $q + 1 - r \equiv 0[\omega]$, then $A_{\omega,\omega}^r = 0$, and downsampling does not modify the matrix $M_{\omega,\omega}^r = M_{\omega,1}^r$.*

*(b) We have $H_0 = 0$ and downsampling does not modify the cyclic invariant eigenspace $A_{\omega,\omega}^r \mathbf{1} = 0$.*

*Proof of Proposition 5.* Let us first start by proving point (a). Consider $q + 1 - r = p\omega$. Fix $i \in \{0, \ldots, \Delta - 1\}$ and $\kappa \in \{0, \ldots, \omega - 1\}$. Let us compute the entry $(i, i + \kappa)$ of the matrix $M_{\omega,\omega}^r$: this amounts to counting the number of quadruples $(k, s, s', t)$ with $k \in [d/\omega]$, $s, s' \in [\omega]$ and $0 \le t \le p\omega - 1$, satisfying $(k\omega + s + t, k\omega + s' + t) \equiv (i, i + \kappa)[d]$. Notice that we must have $s' = s + \kappa$ and therefore $s \in \{0, \ldots, \omega - 1 - \kappa\}$. Notice that for each interval $u\omega \le t < (u+1)\omega$ with $u \in \{0, \ldots, p-1\}$, there are exactly $\omega - \kappa$ ways of choosing $s$ and then $t$ and $k$ to satisfy the equality. We deduce that

$$(M_{\omega,\omega}^r)_{i(i+\kappa)} = \frac{\omega}{\omega(q+1-r)}p(\omega - \kappa) = 1 - \frac{\kappa}{\omega} = (M_{\omega,1}^r)_{i(i+\kappa)}.$$

By symmetry of $M_{\omega,\omega}^r$, this concludes the proof of point (a).

Consider now point (b). First notice, because $M_{\omega,\omega}^r$ has zero entries for $\min(|i-j|, d-|i-j|) \ge \omega$, the only non-zero blocks are $B_1, B_2$ and $B_m$. Furthermore, when computing $H_0$, the diagonal entries only have one contribution from the diagonal elements of $B_1$. The off-diagonal elements of $H_0$ have two contribution: one from $B_1$ and one from $B_2$ (if below the diagonal) or $B_m$ (if above the diagonal), i.e.,

$$(H_0)_{ii} = (B_1)_{ii} \qquad (H_0)_{i(i+\kappa)} = (B_1)_{i(i+\kappa)} + (B_m)_{i(i+\kappa)}.$$

Let us compute first the diagonal elements: we have easily, by a similar argument as above $(M_{\omega,\omega}^r)_{ii} = 1 = (M_{\omega,1}^r)_{ii}$, and therefore $H_0$ has zero zero diagonal entries. For off-diagonal

elements, first notice that $(\boldsymbol{M}_{\omega,\omega}^r)_{i(i+\kappa-\omega)} = (\boldsymbol{M}_{\omega,\omega}^r)_{i(i+\omega-\kappa)}$. Then for $q + 1 - r = p\omega + v$, we can consider each subsegment $u\omega \le t < (u+1)\omega$ separately, and by a simple counting argument, get $(\boldsymbol{M}_{\omega,\omega}^r)_{i(i+\omega-\kappa)} + (\boldsymbol{M}_{\omega,\omega}^r)_{i(i+\kappa)} = 1 - \frac{\kappa}{\omega}$. We deduce that $(\boldsymbol{H}_0)_{i(i+\kappa)} = 0$, which by symmetry implies $\boldsymbol{H}_0 = \boldsymbol{0}$ and concludes the proof. $\qquad\square$

From the above result, we conjecture that more generally, for $\Delta \le \omega$, the low-frequency eigenspaces of $H_\omega^{\mathsf{CK}}$ remain approximately unchanged when applying a downsampling operation. We verify this conjecture numerically in several examples. In Figure 3, we plot the eigenvalues $\kappa_j$ with and without downsampling. On the left, we compare $\kappa_j$ for fixed $\omega = 25$ and increasing $\Delta$. We notice that the eigenvalues do not change much for $\Delta \le \omega$, and for $\Delta > \omega$, some $\kappa_j$ become null, as discussed above. On the right, we plot $\kappa_j$ for $\Delta = 1$ (continuous line) and $\Delta = \omega$ (dashed lines) for several $\omega$. As conjectured, the top eigenvalues (low-frequency) are left approximately unchanged. In Figure 4, we plot a heatmap of the eigenvectors ordered vertically from highest associated eigenvalue (bottom) to lowest (top) for a fixed $\omega = 25$ and increasing downsampling $\Delta \in \{1, 25, 40\}$. First indeed check that the top eigenvectors correspond to low-frequency functions and the bottom eigenvectors correspond to high-frequency functions. Second, most eigenvectors are not much modified between $\Delta = 1$ and $\Delta = \omega = 25$. For the case, $\Delta > \omega$, the top eigenvectors corresponds still low-frequency functions.

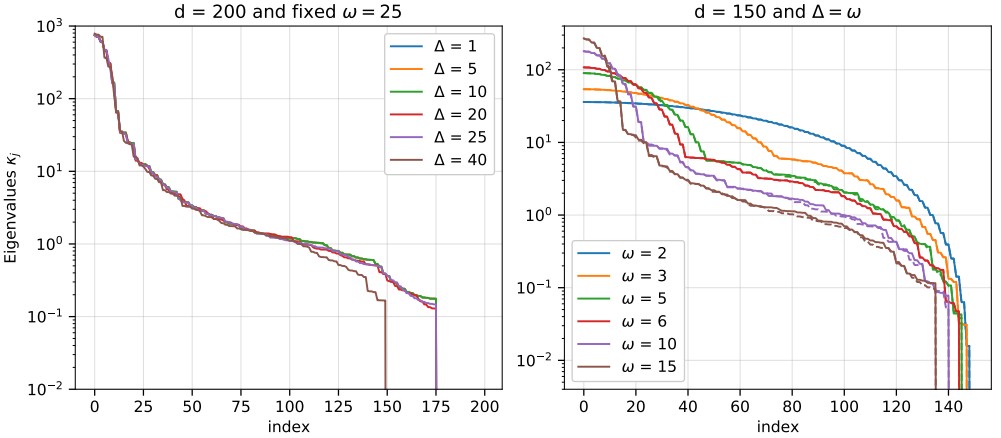

Figure 3: Impact of downsampling on the eigenvalues $\kappa_j$. On the left, we fix $\omega = 25$ ($d = 200$, $q = 30$, $r = 1$) and increase $\delta$ from 1 (no downsampling) to 40. On the right, we compare $\Delta = 1$ (continuous line) and $\Delta = \omega$ (dashed lines), with $d = 150, q = 20, r = 1$.

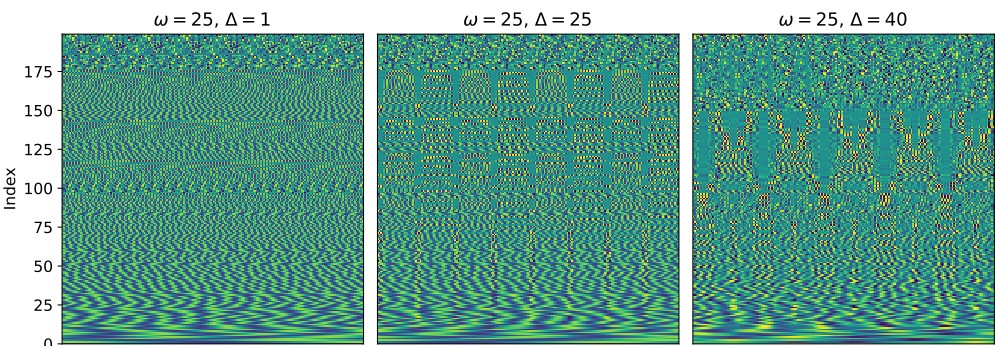

Figure 4: Heatmap of the eigenvectors $\{\boldsymbol{v}_j\}_{j \in [d]}$ ordered from highest associated eigenvalue (bottom) to lowest (top), for $d = 200, q = 30, r = 1, \omega = 25$, and $\Delta = 1$ (left), $\Delta = \omega = 25$ (middle) and $\Delta = 40$ (right).

From these observations, we expect $H^{\mathsf{CK}}_{\omega,\Delta}$ to have the same statistical properties as $H^{\mathsf{CK}}_{\omega}$ when learning low-frequency functions. In Figure 5, we plot the test error of kernel ridge regression for fitting cyclic $q$-local polynomials (see Section A.7) on the hypercube of dimension $d = 30$. We report the test error of one realization, against the sample size $n$, and choose regularization $\lambda = 10^{-6}$ and noise $\sigma_\varepsilon = 0$. We compare kernels with and without downsampling. On the left, we consider $q = 10$ and $\omega = \Delta = 5$, and compare the test error with $H^{\mathsf{CK}}_{\omega}$ (continous line) and with $H^{\mathsf{CK}}_{\omega,\Delta}$ (dashed line) when learning degree 2, 3 and 4 polynomials. On the right, we fix the target function to be the cubic local cyclic polynomial and consider the test error of learning with $H^{\mathsf{CK}}_{\omega,\Delta}$ for $q = 10$, $\omega = 10$, and $\Delta \in \{1, 3, 6, 10\}$. As expected, we observe in both simulations that the test error is almost identical between the kernels with and without downsampling, when learning cyclic invariant functions.

In Section C.1, we further check that downsampling with $\Delta > \omega$ does not improve the high-dimensional predictions for the test error of KRR.

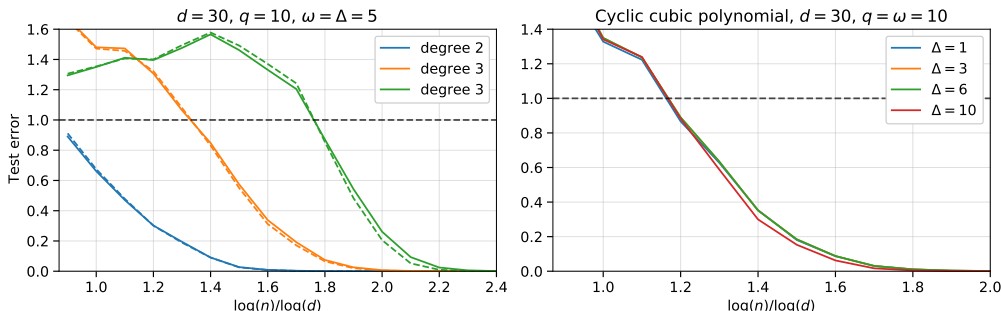

Figure 5: Test error of kernel ridge regression with and without downsampling. We report the test error of one realization, against the sample size $n$. On the left, we consider a unique architecture $q = 10$ and $\omega = \Delta = 5$, and compare $H^{\mathsf{CK}}_{\omega}$ (continuous line) versus $H^{\mathsf{CK}}_{\omega,\Delta}$ (dashed line) when learning cyclic $q$-local polynomials of degree 2, 3 and 4. On the right, we consider a unique cyclic $q$-local polynomial of degree 3 for fixed $q = 10$, $\omega = 10$ and $\Delta \in \{1, 3, 6, 10\}$.

## A.5 Multilayer convolutional kernels

For completeness, we briefly discuss here some intuitions of multilayer convolutional kernels. The benefit of depth in convolutional kernels has been investigated in [6, 16, 40, 43]. In particular, [6] observed that the top layer operation of a two-layers convolutional kernel can be replaced by a low-degree polynomial without a performance change.

As an example, we will consider a two layers convolutional kernel with patch and local average pooling sizes $(q_1, \omega_1)$ on the first layer and $(q_2, \omega_2)$ on the second layer. We consider a general inner-product kernel for the first layer:

$$h_1\big(\langle \boldsymbol{u}, \boldsymbol{v} \rangle / q_1\big) = \langle \psi(\boldsymbol{u}), \psi(\boldsymbol{v}) \rangle, \tag{25}$$

where the feature map is given explicitly $\psi(\boldsymbol{u}) = \{\xi_{q_1,|S|} Y_S(\boldsymbol{u})\}_{S \subseteq [q_1]} \in \mathbb{R}^{2^{q_1}}$. Following the work [6], we consider a degree-2 polynomial kernel on the second layer, i.e., $h_2(\langle \phi, \phi' \rangle) = \langle \phi, \phi' \rangle^2$.

Let us decompose this two-layers convolutional kernel in the Fourier basis. Let $\Psi(\boldsymbol{x}) = \{\Psi_k(\boldsymbol{x})\}_{k \in [d]}$ be the output of the first layer, with

$$\Psi_k(\boldsymbol{x}) = \sum_{s \in [\omega_1]} \psi(\boldsymbol{x}_{(k+s)}) = \left\{ \xi_{q_1,|S|} \sum_{s \in [\omega_1]} Y_{k+s+S}(\boldsymbol{x}) \right\}_{S \subseteq [q_1]} \in \mathbb{R}^{2^{q_1}}. \tag{26}$$

Then denoting $\Psi_{(k)}(\boldsymbol{x}) = (\Psi_{k+1}(\boldsymbol{x}), \ldots, \Psi_{k+q_2}(\boldsymbol{x}))$, the two-layers convolutional kernel is given by

$$
\begin{aligned}
&H^{\mathsf{2CK}}_{\omega_1,\omega_2}(\boldsymbol{x},\boldsymbol{y}) \\
&= \sum_{k\in[d]} \sum_{s,s'\in[\omega_2]} \langle \Psi_{(k+s)}(\boldsymbol{x}), \Psi_{(k+s')}(\boldsymbol{x}) \rangle^2 \\
&= \sum_{k\in[d]} \sum_{s,s'\in[\omega_2]} \sum_{u,u'\in[q_2]} \sum_{t,t',r,r'\in[\omega_1]} \\
&\qquad \langle \psi(\boldsymbol{x}_{(k+s+u+t)}) \otimes \psi(\boldsymbol{x}_{(k+s+u'+r)}), \psi(\boldsymbol{y}_{(k+s'+u+t')}) \otimes \psi(\boldsymbol{y}_{(k+s'+u'+r')}) \rangle .
\end{aligned}
\tag{27}
$$

We believe that techniques contained in this paper can be used to study kernels of the type (**??**) by a careful combinatorial argument and a 2-dimensional Fourier transform on the second layer (see [6]). We leave this problem to future work. Here we only comment on the structure of $H^{\mathsf{2CK}}_{\omega_1,\omega_2}$:

1. Including a second convolutional layer allows interactions between patches. The associated RKHS, which we will denote $\mathcal{H}^{\mathsf{2CK}}$, contains all the homogeneous polynomials $Y_S$ with $S = S_1 \cup S_2$ with $S_1, S_2$ contained on segments of size $q_1$, with the two segments separated by at most $q_2 + \omega_2 - 2$. In words, the RKHS contains interaction between patches $\boldsymbol{x}_{(k)}$ and $\boldsymbol{x}_{(k')}$ that are within some distance.

2. The eigenvalue associated to a degree-$k$ homogeneous polynomials is still of order $q^{-k}$ in high-dimension. To learn functions restricted to $L^2(\mathscr{Q}^2, \mathrm{Loc}_q)$, it is statistically more efficient to use $H^{\mathsf{CK}}$ (smaller degeneracy of eigenvalues). However $H^{\mathsf{2CK}}$ will fit a richer class of functions with two-patch interactions, while still not being plagued by dimensionality: $\dim(\mathcal{H}^{\mathsf{2CK}}) \leq q_2 d 2^{2q_1}$. Hence we still expect $H^{\mathsf{2CK}}$ to be much more statistically efficient than a standard inner-product kernel.

3. Local pooling on the two layers plays different roles: pooling on the first layer encourages the interactions to not depend strongly on the relative positions of the patches, while pooling on the second layer penalizes functions that depend on the global position of these interactions.

For more layers and higher degree kernels, one obtain hierarchical interactions of higher-order, with multi-scale absolute and relative local invariances brought by pooling layers.

## A.6 Proofs diagonalization of convolutional kernels

In this section, we prove the diagonalization of the kernels $H^{\mathsf{CK}}$, $H^{\mathsf{CK}}_\omega$ and $H^{\mathsf{CK}}_{\omega,\Delta}$ introduced in Propositions 1, 2 and 4 respectively.

Recall that we can associate to a kernel function $H : \mathcal{X} \times \mathcal{X} \to \mathbb{R}$ defined on a probability space $(\mathcal{X}, \tau)$ (assume $x \mapsto H(\boldsymbol{x}, \boldsymbol{x})$ square integrable), the integral operator $\mathbb{H} : L^2(\mathcal{X}, \tau) \to L^2(\mathcal{X}, \tau)$

$$
\mathbb{H}f(\boldsymbol{x}) = \int_{\mathcal{X}} H(\boldsymbol{x}, \boldsymbol{x}') f(\boldsymbol{x}') \tau(\mathrm{d}\boldsymbol{x}') .
\tag{28}
$$

By the spectral theorem of compact operators, there exists an orthonormal basis $(\psi_j)_{j\geq 1}$ of $L^2(\mathcal{X}, \tau)$ and eigenvalues $(\lambda_j)_{j\geq 1}$, with nonincreasing values $\lambda_1 \geq \lambda_2 \geq \cdots \geq 0$ and $\sum_{j\geq 1} \lambda_j < \infty$, such that

$$
\mathbb{H} = \sum_{j=1}^{\infty} \lambda_j \psi_j \psi_j^*, \qquad H(\boldsymbol{x}, \boldsymbol{x}') = \sum_{j=1}^{\infty} \lambda_j \psi_j(\boldsymbol{x}) \psi_j(\boldsymbol{x}') .
$$

We first prove the diagonalization of $H^{\mathsf{CK}}_{\omega,\Delta}$ in Proposition 4. The case of $H^{\mathsf{CK}}_\omega$ and $H^{\mathsf{CK}}$ then follows by setting $\Delta = 1$, and $\Delta = \omega = 1$ respectively.

*Proof of Proposition 4.* Consider the inner-product kernel function $h : \mathbb{R} \to \mathbb{R}$ defined on the hypercube $\mathscr{Q}^q$. By rotational symmetry (see Section 2.1 and Appendix D), $h$ admits the following diagonalization: for any $\boldsymbol{u}, \boldsymbol{v} \in \mathscr{Q}^q$,

$$
h\left(\langle \boldsymbol{u}, \boldsymbol{v} \rangle / q\right) = \sum_{\ell=0}^{q} \xi_{q,\ell} \sum_{S\subseteq[q], |S|=\ell} Y_S(\boldsymbol{u}) Y_S(\boldsymbol{v}) ,
\tag{29}
$$

where $(Y_S)_{S \subseteq [q]}$ is the Fourier basis on $\mathcal{Q}^q$, and $\xi_{d,\ell}(h)$ is the $\ell$-th Gegenbauer coefficient of $h$ in dimension $q$ (see Sections 2.1 or D for background).

Recall that we defined $\mathcal{S}_\ell = \{S \subseteq [q] : |S| = \ell\}$, the equivalence relation $S \sim S'$ if $S'$ is a translated subset of $S$ in $[q]$ (without cyclic convention), and $\mathcal{C}_\ell$ the quotient set of $\mathcal{A}_\ell$ by $\sim$. For each equivalence class $\overline{S} \in \mathcal{C}_\ell$, consider $S$ the unique subset in $\overline{S}$ that contains 1. Then the equivalence class $\overline{S}$ contains the subsets $u + S = \{u + k : k \in S\} \subseteq [q]$ with $u = 0, \dots, q - \gamma(S)$. By a slight abuse of notations, we will identify $\overline{S}$ and this subset $S$. Below we will denote $u + S$ the translated subset with cyclic convention on $[d]$ (e.g., $2 + \{1, 3, d-1\} = \{3, 5, 1\}$).

Using Eq. (29) and that $Y_S(\boldsymbol{x}_{(k)}) = Y_{k+S}(\boldsymbol{x})$, we have the following decomposition of $H^{\mathsf{CK}}_{\omega,\Delta}$ in the Fourier basis

$$
\begin{aligned}
&H^{\mathsf{CK}}_{\omega,\Delta}(\boldsymbol{x}, \boldsymbol{y}) \\
&= \frac{\Delta}{\omega} \sum_{k \in [d/\Delta]} \sum_{s,s' \in [\omega]} h\left( \langle \boldsymbol{x}_{(k\Delta+s)}, \boldsymbol{y}_{(k\Delta+s')} \rangle / q \right) \\
&= d\omega \xi_{q,0} + \sum_{\ell=1}^{q} \xi_{q,\ell} \sum_{S \in \mathcal{C}_\ell} \left\{ \frac{\Delta}{\omega} \sum_{(k,s,s',t) \in \mathcal{I}_{\omega,\Delta,\gamma(S)}} Y_{k\Delta+s+t+S}(\boldsymbol{x}) Y_{k\Delta+s'+t+S}(\boldsymbol{y}) \right\},
\end{aligned}
\tag{30}
$$

where we recall the definition of the set of indices

$$
\mathcal{I}_{\omega,\Delta,\gamma(S)} = \left\{ (k, s, s', t) : k \in [d/\Delta], s, s' \in [\omega], 0 \leq t \leq q - \gamma(S) \right\}.
\tag{31}
$$

Note that the diagonalization of the kernel $H$ can be obtained by computing the matrix $\boldsymbol{M} = (M_{SS'})_{S,S' \subseteq [d]} \in \mathbb{R}^{2^d \times 2^d}$ with $M = \mathbb{E}_{\boldsymbol{x},\boldsymbol{y}}[Y_S(\boldsymbol{x}) H(\boldsymbol{x}, \boldsymbol{y}) Y_{S'}(\boldsymbol{y})]$: if $\lambda_j$ and $\boldsymbol{v}_j \in \mathbb{R}^{2^d}$ are the eigenvalues and eigenvectors of $\boldsymbol{M}$, then $\lambda_j$ and $\psi_j(\boldsymbol{x}) = \sum_{S \subseteq [d]} v_{j,S} Y_S(\boldsymbol{x})$ are the eigenvalues and eigenvectors of $H$.

From Eq. (30), we see 1) the basis functions $Y_S$ with $\gamma(S) > q$ (subset $S$ not contained in a segment of size $q$) are in the null space of $H^{\mathsf{CK}}_{\omega,\Delta}$, 2) for $S, S' \subseteq [d]$ with $S$ and $S'$ not translations of each other, then $\mathbb{E}_{\boldsymbol{x},\boldsymbol{y}}[Y_S(\boldsymbol{x}) H^{\mathsf{CK}}_{\omega,\Delta}(\boldsymbol{x}, \boldsymbol{y}) Y_{S'}(\boldsymbol{y})] = 0$, and $Y_S$ and $Y_{S'}$ are contained in orthogonal eigenspaces. We deduce that it is sufficient to diagonalize $H^{\mathsf{CK}}_{\omega,\Delta}$ on each of the (orthogonal) subspaces $V_S := \mathsf{span}\{Y_{k+S} : k \in [d]\}$ for $0 \leq \ell \leq q$ and $S \in \mathcal{C}_\ell$.

For each $S \in \mathcal{C}_\ell$, define $\boldsymbol{M}^{\gamma(S)} \in \mathbb{R}^{d \times d}$ the matrix with entries $M_{ij}^{\gamma(S)} = \frac{1}{r(S)} \mathbb{E}_{\boldsymbol{x},\boldsymbol{y}}[Y_{i+S}(\boldsymbol{x}) H^{\mathsf{CK}}_{\omega,\Delta}(\boldsymbol{x}, \boldsymbol{y}) Y_{j+S}(\boldsymbol{y})]$. From Eq. (30), we get

$$
M_{ij}^{\gamma(S)} = \frac{\Delta}{\omega r(S)} \left| \left\{ (k, s, s', t) \in \mathcal{I}_{\omega,\Delta,\gamma(S)} : k\Delta + s + t \equiv i[d], k\Delta + s' + t \equiv j[d] \right\} \right|,
\tag{32}
$$

which concludes the proof of Proposition 4. $\qquad \square$

We can now prove Propositions 1 and 2 by taking $\omega = \Delta = 1$ and $\Delta = 1$ respectively.

*Proof of Proposition 1.* Set $\Delta = \omega = 1$ in Proposition 4. We get

$$
\begin{aligned}
\boldsymbol{M}_{ij}^{\gamma(S)} &= \frac{1}{r(S)} \left| \left\{ (k, t) : k \in [d], 0 \leq t \leq q - \gamma(S), k + 1 + t \equiv i[d], k + 1 + t \equiv j[d] \right\} \right| \\
&= \delta_{ij}.
\end{aligned}
$$

In this case, $\boldsymbol{M}^{\gamma(S)}$ is simply equal to identity, which concludes the proof. $\qquad \square$

*Proof of Proposition 2.* Set $\Delta = 1$ in Proposition 4. We get

$$
\begin{aligned}
\boldsymbol{M}_{ij}^{\gamma(S)} &= \frac{1}{\omega r(S)} \left| \left\{ (k, s, s', t) \in \mathcal{I}_{\omega,\Delta,\gamma(S)} : k + s + t \equiv i[d], k + s' + t \equiv j[d] \right\} \right| \\
&= \left( 1 - \frac{d(i,j)}{\omega} \right)_+,
\end{aligned}
$$

where $d(i, j)$ is the distance between $i$ and $j$ on the torus $[d]$ (i.e., if $i > j$, $d(i, j) = \min(i - j, d + j - i)$). Hence, $\boldsymbol{M}^{\gamma(S)}$ is a circulant matrix independent of $\gamma(S)$, which has well known explicit formula for eigenvalues and eigenvectors (see for example Remark 1). $\qquad\square$

## A.7  Additional numerical simulations

Here, we consider a numerical experiment similar to Figure 1. We consider $\boldsymbol{x} \sim \mathrm{Unif}(\mathcal{Q}^d)$ with $d = 30$ and consider three cyclic invariant target functions:

$$f_2(\boldsymbol{x}) = \frac{1}{\sqrt{d}} \sum_{i \in [d]} x_i x_{i+1}\,, \qquad f_3(\boldsymbol{x}) = \frac{1}{\sqrt{d}} \sum_{i \in [d]} x_i x_{i+1} x_{i+2}\,,$$

$$f_4(\boldsymbol{x}) = \frac{1}{\sqrt{d}} \sum_{i \in [d]} x_i x_{i+1} x_{i+2} x_{i+3}\,.$$

We consider a higher order polynomial kernel $h(x) = \sum_{k \in [7]} 0.2 \cdot x^k$ than in Figure 1, which should lead to higher self-induced regularization. We consider the same kernels as before, with $q = 10$ and $\omega = 5$.

In Figure 6, we report the test errors of fitting $f_2$ (top), $f_3$ (middle) and $f_4$ (bottom) using kernel ridge regression with the 5 kernels of interests in the main text. We choose a small regularization parameter $\lambda = 10^{-6}$, and the noise level $\sigma_\varepsilon = 0$. The curves are averaged over 5 independent instances and the error bar stands for the standard deviation of these instances. The results again match with our overall theoretical predictions. We report the predicted thresholds for the three functions:

1. For $f_2$ target: $q < d < dq/\omega < dq < d^2$ for $H_{\mathsf{GP}}^{\mathsf{CK}} < H_{\mathsf{GP}}^{\mathsf{FC}} < H_\omega^{\mathsf{CK}} < H^{\mathsf{CK}} < H^{\mathsf{FC}}$.
2. For $f_3$ target: $q^2 < dq^2/\omega < d^2 < dq^2 < d^3$ for $H_{\mathsf{GP}}^{\mathsf{CK}} < H_\omega^{\mathsf{CK}} < H^{\mathsf{CK}} < H_{\mathsf{GP}}^{\mathsf{FC}} < H^{\mathsf{FC}}$.
3. For $f_4$ target: $q^3 < dq^3/\omega < d^3 < dq^3 < d^4$ for $H_{\mathsf{GP}}^{\mathsf{CK}} < H_\omega^{\mathsf{CK}} < H^{\mathsf{CK}} < H_{\mathsf{GP}}^{\mathsf{FC}} < H^{\mathsf{FC}}$.

We see that the kernels, especially for $f_4$, perform much better than their theoretical high-dimension predictions: this can be explained by the low-dimensionality of the experiment where $q = 10$.

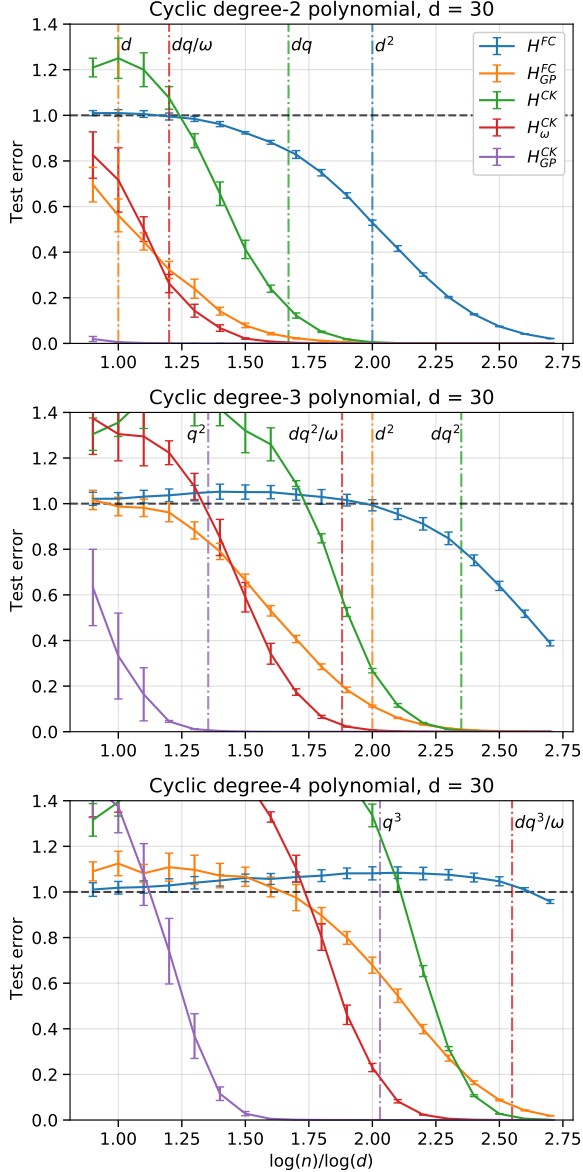

Figure 6: Learning cyclic polynomials of degree 2 (top), 3 (middle) and 4 (bottom) over the hypercube $d = 30$, using KRR with $H^{\mathsf{FC}}$ (FC), $H_{\mathsf{GP}}^{\mathsf{FC}}$ (FC-GP), $H^{\mathsf{CK}}$ (CK), $H_{\omega}^{\mathsf{CK}}$ (CK-LP) and $H_{\mathsf{GP}}^{\mathsf{CK}}$ (CK-GP), regularization parameter $\lambda = 0^+$ and $h(x) = \sum_{k \in [7]} 0.2 \cdot x^k$. We report the average and the standard deviation of the test error over $5$ realizations, against the sample size $n$.

# B  Generalization error of kernel methods in fixed dimension

## B.1  Bound on kernel methods using Rademacher complexities

We first consider the case of a Lipschitz bounded loss and uniform convergence, and make a few simple remarks on the connection between generalization error and eigendecomposition in kernel methods.

Consider i.i.d data $(\boldsymbol{x}_i, y_i) \in \mathcal{X} \times \mathbb{R}$ with $(\boldsymbol{x}, y) \sim P$ and a loss function $\ell : \mathbb{R} \times \mathbb{R} \to \mathbb{R}$ that we take 1-Lipschitz w.r.t second argument and bounded by 1. The goal is to minimize the expected loss $L(\hat{f}) = \mathbb{E}_{\boldsymbol{y}, \boldsymbol{x}}\{\ell(y, \hat{f}(\boldsymbol{x}))\}$. Take a RKHS $\mathcal{H}$ with kernel function $H : \mathcal{X} \times \mathcal{X} \to \mathbb{R}$ and consider following constrained empirical risk minimizer:

$$\hat{f}_B = \underset{\|f\|_{\mathcal{H}} \leq B}{\arg\min} \left\{ \sum_{i=1}^{n} \ell(y_i, f(\boldsymbol{x}_i)) \right\} . \tag{33}$$

The generalization error of $\hat{f}_B$ has the following standard bound on the Rademacher complexity of the kernel class $\{f : \|f\|_{\mathcal{H}} \leq B\}$ [9, 44]: with probability $1 - \delta$,

$$L(\hat{f}_B) - \min_{\|f\|_{\mathcal{H}} \leq B} L(f) \leq \frac{8B}{\sqrt{n}} \sqrt{\mathbb{E}_{\boldsymbol{x}}\{H(\boldsymbol{x}, \boldsymbol{x})\}} + \sqrt{\frac{2 \log \frac{2}{\delta}}{n}} . \tag{34}$$

Note that instead of a constraint on the norm in Eq. (33), one might find more convenient to use a penalty. In that case, there exists an equivalent to the bound (34) [3, 47], but we focus here on the constrained formulation for simplicity.

From the bound (34), we see that the generalization error depends crucially on the choice of $B$. For simplicity, let us forget about the approximation error and take $\|f_\star\|_{\mathcal{H}} \leq B$ where $f_\star = \mathbb{E}\{y|\boldsymbol{x}\}$. Recall that for a kernel $H$ with eigenvalues $\{\lambda_j\}_{j \geq 1}$ and eigenvectors $\{\psi_j\}_{j \geq 1}$, we have

$$\|f\|_{\mathcal{H}}^2 = \sum_{j \geq 1} \lambda_j^{-1} \langle \psi_j, f \rangle_{L^2(P)}^2 .$$

Consider $H_{\omega, \Delta}^{\mathsf{CK}}$ as in Eq. (8) and assume $\xi_{q,0} = 0$. From the normalization choice of the kernel (see Eq. (22)), we have

$$\mathbb{E}_{\boldsymbol{x}}\{H_{\omega, \Delta}^{\mathsf{CK}}(\boldsymbol{x}, \boldsymbol{x})\} = h(1).$$

Consider now for simplicity $\Delta = 1$. From the eigendecomposition in Proposition 2, the RKHS norm of $f \in L^2(\mathscr{Q}^d, \mathrm{Loc}_q)$ is given by

$$\|f\|_{\mathcal{H}}^2 = \sum_{\ell \in [q]} \sum_{j \in [d]} \sum_{S \in \mathcal{C}_\ell} \frac{\langle \psi_{j,S}, f \rangle_{L^2}^2}{\xi_{q,\ell} r(S) \kappa_j / d} .$$

Consider the case where $f \in L^2(\mathscr{Q}^d, \mathrm{Loc}_q)$ has a unique non-zero component in its discrete Fourier transform, i.e., $f(\boldsymbol{x}) = \frac{1}{\sqrt{d}} \sum_{k \in [d]} \rho_j^k g(\boldsymbol{x}_{(k)})$ with $\mathbb{E}\{g(\boldsymbol{x})\} = 0$ and $\rho_j = e^{2i\pi j/d}$ (see Section A.3). Note that, denoting $c_S = \langle Y_S, g \rangle_{L^2(\mathscr{Q}^q)}$:

$$f(\boldsymbol{x}) = \sum_{\ell=1}^{q} \sum_{S \in \mathcal{C}_\ell} \left( \sum_{u=0}^{r(S)-1} \rho_j^{-u} c_{u+S} \right) \psi_{j,S} .$$

Hence,

$$\|f\|_{\mathcal{H}}^2 = \sum_{\ell=1}^{q} \sum_{S \in \mathcal{C}_\ell} \frac{\langle \psi_{j,S}, f \rangle_{L^2}^2}{\xi_{q,\ell} r(S) \kappa_j / d} \leq d \sum_{\ell=1}^{q} \sum_{S \in \mathcal{C}_\ell} \sum_{u=0}^{r(S)-1} \frac{c_{u+S}^2}{\xi_{q,\ell} r(S)} \leq \frac{d \|g\|_h^2}{\kappa_j} ,$$

where $\|g\|_h^2$ is the RKHS norm associated to the inner-product kernel $h : \mathbb{R} \to \mathbb{R}$ in $\mathscr{Q}^q$, i.e., $\|g\|_h^2 = \sum_{S \subseteq [q]} \frac{c_S}{\xi_{q,|S|}}$. From the bound (34), we deduce the first generalization bound using a convolutional kernel: with probability at least $1 - \delta$,

$$L(\hat{f}_B) - \min_{\|f\|_{\mathcal{H}} \leq B} L(f) \leq 8 \left( \frac{d \|g\|_h^2 h(1)}{n \kappa_j} \right)^{1/2} + \sqrt{\frac{2 \log \frac{2}{\delta}}{n}} .$$

We make the following two remarks on this bound:

1. It depends on $\|g\|_h$, which is a RKHS norm on $\mathcal{Q}^q$ instead of $\mathcal{Q}^d$, which has potentially much lower dimension and contain less smooth function for balls of same radius.

2. There is a factor $\kappa_j$ gain in sample complexity when learning functions that have $j$-th frequency with $\kappa_j > 1$. In particular, for $j = d$ (cyclic invariant functions), $\kappa_j = \omega$, and we need $\omega$ less samples to get the same (upper) bound on the generalization error. On the contrary, when $\kappa_j < 1$, i.e., high-frequency oscillatory functions, the generalization bound becomes worse.

## B.2 Generalization error of KRR in the classical regime

We consider here the regression setting which allows for finer results. Several works have considered bounding the generalization error of kernel ridge regression (KRR) [12, 29], [47, Theorem 13.17]. In this section, we consider the following fully-explicit upper bound from [3].

Consider i.i.d data $(\boldsymbol{x}_i, y_i) \in \mathcal{X} \times \mathbb{R}$ with $\boldsymbol{x}_i \sim P$, and $y_i = f_\star(\boldsymbol{x}_i) + \varepsilon_i$. Assume the noise $\mathbb{E}[\varepsilon_i | \boldsymbol{x}_i] = 0$ and $\mathbb{E}[\varepsilon_i^2 | \boldsymbol{x}_i] \leq \sigma_\varepsilon^2$, and denote $\boldsymbol{\varepsilon} = (\varepsilon_1, \ldots, \varepsilon_n)$.

Let $\mathcal{H}$ be a RKHS with reproducing kernel $H : \mathcal{X} \times \mathcal{X} \to \mathbb{R}$. The KRR solution with regularization parameter $\lambda \geq 0$ is given by

$$\hat{f}_\lambda = \arg\min_{f \in \mathcal{H}} \left\{ \sum_{i=1}^n (y_i - f(\boldsymbol{x}_i))^2 + \lambda \|f\|_{\mathcal{H}}^2 \right\},$$

which has the following analytical formula:

$$\hat{f}_\lambda(\boldsymbol{x}) = \boldsymbol{h}(\boldsymbol{x})(\boldsymbol{H} + \lambda \mathbf{I}_n)^{-1} \boldsymbol{y},$$

where $\boldsymbol{H} = (H(\boldsymbol{x}_i, \boldsymbol{x}_j))_{ij \in [n]}$ is the empirical kernel matrix, $\boldsymbol{h}(\boldsymbol{x}) = [H(\boldsymbol{x}, \boldsymbol{x}_1), \ldots, H(\boldsymbol{x}, \boldsymbol{x}_n)]$ and $\boldsymbol{y} = (y_1, \ldots, y_n)$. The risk is taken to be the test error with squared error loss

$$R(f_\star, \hat{f}_\lambda) = \mathbb{E}_{\boldsymbol{x}} \left\{ \left( f_\star(\boldsymbol{x}) - \hat{f}_\lambda(\boldsymbol{x}) \right)^2 \right\}. \tag{35}$$

Below, we give an upper bound on the expected risk over the noise $\boldsymbol{\varepsilon}$ in the training data, i.e., $\mathbb{E}_{\boldsymbol{\varepsilon}}\{R(f_\star, \hat{f}_\lambda)\}$ (it is also possible to give high probability bounds by concentration arguments, but we restrict ourselves to bounding the expected risk).

**Theorem 6.** *[3, Theorem 7.2] Assume $H(\boldsymbol{x}, \boldsymbol{x}) \leq R^2$ almost surely and let the regularization parameter $\lambda \leq R^2$. If $n \geq \frac{5R^2}{\lambda} \left( 1 + \log \frac{R^2}{\lambda} \right)$, then*

$$\mathbb{E}_{\boldsymbol{\varepsilon}}\{R(f_\star, \hat{f}_\lambda)\} \leq 16 \frac{\sigma_\varepsilon^2}{n} \mathcal{N}(H, \lambda) + 16 \inf_{f \in \mathcal{H}} \left\{ \|f - f_\star\|_{L^2}^2 + \lambda \|f\|_{\mathcal{H}}^2 \right\} + \frac{24}{n^2} \|f_\star\|_{L^\infty}^2, \tag{36}$$

*where $\mathcal{N}(H, \lambda) = \mathrm{Tr}[(\mathbb{H} + \lambda \mathbf{I})^{-1} \mathbb{H}]$.*

Let us comment on the upper-bound in Eq. (36). The first term corresponds to an upper bound on the variance: $\mathcal{N}(H, \lambda)$ is sometimes called the *degrees of freedom* or the *effective dimension* of the kernel $H$. The second term bounds the bias term and corresponds to an approximation error. In particular, for any $r > 0$,

$$\inf_{f \in \mathcal{H}} \left\{ \|f - f_\star\|_{L^2}^2 + \lambda \|f\|_{\mathcal{H}}^2 \right\} \leq \lambda^r \|\mathbb{H}^{-r/2} f_\star\|_{L^2}^2, \tag{37}$$

where we recall that $\mathbb{H}$ is the integral operator associated to $H$ (see Eq. (28)). The third term can be removed by a more intricate analysis.

From the above discussion, it is natural to consider the following two assumptions on $H$ and $f_\star$, that are standard in the kernel literature:

(B1) *Capacity condition:* $\mathcal{N}(H, \lambda) \leq C_H \lambda^{-1/\alpha}$ with $\alpha > 1$.

(B2) *Source condition:* there exists $\beta > 0$ such that $\|\mathbb{H}^{-\beta/2} f_\star\|_{L^2}^2 =: B_{f_\star}^2 < \infty$.

Intuitively, the capacity condition (B1) characterizes the size of the RKHS: for increasing $\alpha$, the RKHS contains less and less functions. It is verified when the eigenvalues $\lambda_j$'s of $H$ decay at the rate $j^{-\alpha}$. For example, taking the Matern kernel of order $s > d/2$, whose RKHS is the Sobolev space of order $s$ (i.e., functions with bounded $s$-order derivatives), we have $\alpha = 2s/d$ (e.g., see [26]). The source condition (B2) characterizes the regularity of the target function (the 'source') with respect to the kernel: $\beta = 1$ is equivalent to $f_\star \in \mathcal{H}$, while $\beta > 1$ corresponds to $f_\star$ more smooth (and $\beta < 1$ less smooth $f_\star$).

Assuming (B1) and (B2) in Theorem 6, we get the bound

$$
\begin{aligned}
\mathbb{E}_\varepsilon\{R(f_\star, \hat{f}_\lambda)\} &\leq 16 C_H \frac{\sigma_\varepsilon^2}{n} \lambda^{-1/\alpha} + 16 B_{f_\star}^2 \lambda^\beta + \frac{24}{n^2} \|f_\star\|_{L^\infty}^2 \\
&= 32 \sigma_\varepsilon^2 B_{f_\star}^{\frac{2}{\alpha\beta+1}} \left(\frac{C_H}{n}\right)^{\frac{\alpha\beta}{\alpha\beta+1}} + \frac{24}{n^2} \|f_\star\|_{L^\infty}^2 ,
\end{aligned}
\tag{38}
$$

where in the second line, we balanced the two terms by taking $\lambda_* := \left(\frac{C_H \sigma_\varepsilon^2}{B_{f_\star}^2 n}\right)^{\frac{\alpha}{\alpha\beta+1}}$. Note that in order to use Theorem 6, we need further to constrain $n \geq \frac{5R^2}{\lambda}\left(1 + \log \frac{R^2}{\lambda}\right)$. For simplicity, we will choose $r > \frac{\alpha-1}{\alpha}$, so that this condition is verified for $n$ sufficiently large.

**Remark 2.** The rate in $n$ in Eq. (38) is minmax optimal over all functions that verify assumptions (A1) and (A2) [12]. However, for large $d$, the RKHS is composed of very smooth functions (e.g., Sobolev spaces of order $s$ are RKHS if and only if $s > d/2$, i.e., if the order of the bounded derivatives grows with the dimension $d$) and $\beta$ will be small, such that $\beta\alpha \approx \kappa/d$ for functions with bounded derivatives up to order $\kappa$. In that case, the risk decreases at the rate $n^{-O(\frac{\kappa}{d})}$: KRR suffers from the curse of dimensionality when $\kappa$ does not scale with $d$. As a consequence, the bound (38) is vacuous when $n$ does not scale exponentially in $d$, which led several groups to derive finer bounds on KRR in the high dimensional regime (see Section C).

Let us now apply Theorem 6 and Eq. (38) to our convolutional kernels to show Theorems 1 and 4.

*Proof of Theorem 1.* First notice that $H^{\mathsf{CK}}(\boldsymbol{x}, \boldsymbol{x}) = h(1) =: R^2$ and we can therefore apply Theorem 6. The effective dimension of $H^{\mathsf{CK}}$ is bounded by

$$
\begin{aligned}
\mathcal{N}(\mathbb{H}^{\mathsf{CK}}, \lambda) &= \frac{\xi_{q,0}}{\xi_{q,0} + \lambda} + \sum_{\ell=1}^q \sum_{S \in \mathcal{E}_\ell} \frac{\xi_{q,\ell} r(S)/d}{\xi_{q,\ell} r(S)/d + \lambda} \\
&\leq \frac{d\xi_{q,0}}{\xi_{q,0} + d \cdot \lambda} + \sum_{\ell=0}^q \frac{\xi_{q,\ell}}{\xi_{q,\ell} + d \cdot \lambda} \sum_{S \in \mathcal{E}_\ell} r(S) \\
&= d \sum_{\ell=0}^q B(\mathscr{Q}^q, \ell) \frac{\xi_{q,\ell}}{\xi_{q,\ell} + d \cdot \lambda} = d\mathcal{N}(h, d \cdot \lambda) ,
\end{aligned}
$$

where we used that $r(S) \geq 1$ in the second line and $\mathcal{N}(h, \lambda)$ is the effective dimension of the inner-product kernel $h$ on $\mathscr{Q}^q$. We deduce from (A1) that $\mathcal{N}(H^{\mathsf{CK}}, \lambda) \leq C_h d^{1-1/\alpha} \lambda^{-1/\alpha}$. Furthermore, from (A2) and the assumption that $\mathbb{E}\{g_k(\boldsymbol{x})\} = 0$, we have

$$
\begin{aligned}
\|(H^{\mathsf{CK}})^{-\beta/2} f_\star\|_{L^2}^2 &= d^\beta \sum_{\ell=1}^q \xi_{q,\ell}^{-\beta} \sum_{S \in \mathcal{C}_\ell} \sum_{k \in [d]} r(S)^{-\beta} \left(\sum_{u=0}^{r(S)-1} \langle g_{k-u}, Y_{u+S}\rangle_{L^2}\right)^2 \\
&\leq d^\beta \sum_{\ell=1}^q \xi_{q,\ell}^{-\beta} \sum_{S \in \mathcal{C}_\ell} \sum_{k \in [d]} r(S)^{1-\beta} \sum_{u=0}^{r(S)-1} \langle g_{k-u}, Y_{u+S}\rangle_{L^2}^2 \\
&\leq d^\beta q^{1-\beta} \sum_{k=1}^d \|h^{-\beta/2} g_k\|_{L^2}^2 \leq d^\beta q B^2 .
\end{aligned}
$$

Injecting the two above bounds in Eq. (38), we deduce that there exists constants $C_1, C_2, C_3$ that only depends on the constants in (A1) and (A2), and $h(1), \sigma_\varepsilon^2$ (but independent of $d$), such that taking

$n \geq C_1 \max(\|f_\star\|_{L^\infty}^2, d)$ and $\lambda_* = \frac{C_2}{d}(d/n)^{\frac{\alpha}{\alpha\beta+1}}$, we get

$$\mathbb{E}_{\boldsymbol{\varepsilon}}\{R(f_\star, \hat{f}_{\lambda_*})\} \leq C_3 \left(\frac{d}{n}\right)^{\frac{\alpha\beta}{\alpha\beta+1}} .$$

$\square$

*Proof of Theorem 4.* The proof is similar to the proof of Theorem 1. Notice that $H_\omega^{\mathsf{CK}}(\boldsymbol{x}, \boldsymbol{x}) \leq h(1)$, and that the effective dimension of $H_\omega^{\mathsf{CK}}$ is bounded by

$$\mathcal{N}(H_\omega^{\mathsf{CK}}, \lambda) = \sum_{j=1}^d \sum_{\ell=1}^q \sum_{S \in \mathcal{C}_\ell} \frac{\xi_{q,\ell} r(S)\kappa_j/d}{\xi_{q,\ell} r(S)\kappa_j/d + \lambda}$$

$$\leq \sum_{j=1}^d \sum_{\ell=1}^q \sum_{S \in \mathcal{C}_\ell} r(S) \frac{\xi_{q,\ell}}{\xi_{q,\ell} + d\lambda/\kappa_j} = \sum_{j=1}^d \mathcal{N}(h, d\lambda/\kappa_j) \leq C_h d^{-1/\alpha} \lambda^{-1/\alpha} \sum_{j=1}^d \kappa_j^{1/\alpha} ,$$

where we used condition (A1). Denoting $d_{\text{eff}} = \sum_{j=1}^d (\kappa_j/\omega)^{1/\alpha}$, the rest of the proof follows from the proof of Theorem 1 with $d$ replaced by $d_{\text{eff}}\omega^{1/\alpha}$ and $B^2$ replaced by $\omega^\beta B^2$. $\square$

**Remark 3.** Note that the requirement $\|(\mathbb{H}_\omega^{\mathsf{CK}}/\omega)^{-\beta/2} f_\star\|_{L^2} \leq B$ is to make the result comparable to the other theorems when we consider target functions with low-frequencies. For a cyclic invariant function, we get exactly $\|(\mathbb{H}_\omega^{\mathsf{CK}}/\omega)^{-\beta/2} f_\star\|_{L^2} = \|(\mathbb{H}^{\mathsf{CK}})^{-\beta/2} f_\star\|_{L^2}$.

## C  Generalization error of KRR in high dimension

In Section B.2, we considered upper bounds on the test error of KRR using the standard *capacity* and *source conditions*. However, these results suffer from several limitations:

1. They only provide an upper bound on the test error. While the decay rate with respect to $n$ is minmax optimal (see [12]), this is not strong enough to show, for example, a statistical advantage of using local average pooling, which appears as a prefactor $d_{\text{eff}}$, and which would require a lower bound matching the upper bound within a constant factor.

2. As mentioned in Remark 2, the bound is of order $n^{-1/O(d)}$, except when the target function has smoothness order increasing with $d$. This bound is non-vacuous only if $n = \exp(O(d))$ which is impractical in modern image datasets where typically $d \geq 100$. This motivates a new type of question: given $n \asymp d^\alpha$, what is the prediction error achieved by KRR for a given function?

3. In order to achieve the bound Eq. (38), one need to carefully balance the bias and the variance terms by setting the regularization parameter. This is in contrast with modern practice which usually train until interpolation (which corresponds to setting $\lambda \to 0$).

Given the above limitations, several recent works have instead considered a high-dimensional setting where the number of samples scales with $d$, and derived asymptotic test errors, exact up to a vanishing additive error [23, 24, 38]. In addition to these works, several papers have derived general estimates for the test error using non-rigorous methods [11, 17, 29] that are believe to be correct in the high dimensional limit and which show great agreement with numerical experiments. The picture that emerges in this regime is much more precise than in the classical regime: KRR approximately acts as a *shrinkage operator* on the target function (not assumed to be in a particular space anymore), with shrinkage parameter that scales as a self-induced regularization parameter over the number of samples.

More precisely, [38] shows the following: considers a kernel $H_d : \mathbb{R}^d \times \mathbb{R}^d \to \mathbb{R}$ with eigenvalues $(\lambda_{d,j})_{j \geq 1}$ in nonincreasing order and $n \equiv n(d)$ the number of samples. Let $m \equiv m(d)$ be an integer such that $m \leq n^{1-\delta}$ and

$$\lambda_{d,m+1} \cdot n^{1+\delta} \leq \sum_{j=m+1}^\infty \lambda_{d,j} ,$$

for some $\delta > 0$. Then, assuming some additional conditions insuring that the kernel $H_d$ is 'spread-out' and well behaved, the KRR solution

$$\hat{f}_\lambda = \underset{f \in \mathcal{H}_d}{\arg\min} \left\{ \frac{1}{n} \sum_{i=1}^n \left( y_i - f(\boldsymbol{x}_i) \right)^2 + \frac{\lambda}{n} \|f\|_{\mathcal{H}_d}^2 \right\} , \tag{39}$$

is equal up to a vanishing additive $L^2$-error (as $d \to \infty$) to the following effective ridge regression estimator

$$\hat{f}_{\lambda_{\mathrm{eff}}}^{\mathrm{eff}} = \underset{f \in \mathcal{H}_d}{\arg\min} \left\{ \|f_\star - f\|_{L^2}^2 + \frac{\lambda_{\mathrm{eff}}}{n} \|f\|_{\mathcal{H}_d}^2 \right\} , \tag{40}$$

where $\lambda_{\mathrm{eff}} = \lambda + \sum_{j=m+1}^\infty \lambda_{d,j}$. The effective estimator (40) amounts to replacing the empirical risk in Eq. (39) by its population counterpart $\|f_\star - f\|_{L^2}^2 = \mathbb{E}_{\boldsymbol{x}}\{(f_\star(\boldsymbol{x}) - f(\boldsymbol{x}))^2\}$. In words, in high dimension, KRR with a finite number of samples is the same as KRR with infinite number of samples but with a larger ridge regularization.

The solution of Eq. (40) admits an explicit solution in terms of a *shrinkage operator* in the basis $(\psi_{d,j})_{j \geq 1}$ of eigenfunctions of $H_d$:

$$f_\star(\boldsymbol{x}) = \sum_{j=1}^\infty c_j \psi_{d,j}(\boldsymbol{x}) \quad \mapsto \quad \hat{f}_{\lambda_{\mathrm{eff}}}^{\mathrm{eff}} = \sum_{j=1}^\infty \frac{\lambda_{d,j}}{\lambda_{d,j} + \frac{\lambda_{\mathrm{eff}}}{n}} \cdot c_j \cdot \psi_{d,j}(\boldsymbol{x}) . \tag{41}$$

Hence, KRR will fit better the target function along eigendirections associated to larger eigenvalues of $H$. If $\lambda_{d,j} \gg \lambda_{\mathrm{eff}}/n$, KRR fits perfectly $f_\star$ along the eigendirection $\psi_{d,j}$, while if $\lambda_{d,j} \ll \lambda_{\mathrm{eff}}/n$, KRR does not fit this eigendirection at all. This phenomena has been referred as the *spectral bias* and *task-kernel alignment* of kernel ridge regression in several works.

Finally, notice from Eq. (41) that the minimum test error is achieved for the regularization parameter $\lambda = 0$, which corresponds to the KRR estimator fitting perfectly the training data. In other words, the *interpolating solution is optimal for kernel ridge regression in high dimension*.

### C.1   Generalization error of convolutional kernels in high dimension

Consider a sequence of integers $\{d(q)\}_{q \geq 1}$ which corresponds to a sequence of image spaces $\boldsymbol{x} \in \mathscr{Q}^d$ of increasing dimension, and assume $d(q)/2 \geq q \geq d(q)^\delta$ for some constant $\delta > 0$. For ease of notations, we will keep the dependency on $q$ implicit, i.e., $d := d(q)$. Let $\{h_q\}_{q \geq 1}$ be a sequence of inner-product kernels $h_q : \mathbb{R} \to \mathbb{R}$.

**Test error with one-layer convolutional kernel:**   we first consider a vanilla one-layer convolutional kernel $H^{\mathsf{CK}}$ as defined in Eq. (3). We will assume that the kernels $\{h_q\}_{q \geq 1}$ verify the following 'genericity' condition.

**Assumption 1** (Generecity assumption on $\{h_q\}_{q \geq 1}$ at level $\mathsf{s} \in \mathbb{N}$). *For $\{h_q\}_{q \geq 1}$ a sequence of inner-product kernels $h_q : \mathbb{R} \to \mathbb{R}$, we assume the following conditions to hold. There exists $\mathsf{s}' \geq 1/\delta + 2\mathsf{s} + 3$ where $\delta > 0$ verifies $q \geq d^\delta$ and a constant $C$ such that $h_q(1) \leq C$, and*

$$\min_{k \leq \mathsf{s}-1} q^{\mathsf{s}-1-k} \xi_{q,k} B(q,k) = \Omega_d(1), \tag{42}$$

$$\min_{k \in \{\mathsf{s}, \mathsf{s}+1, \mathsf{s}'\}} \xi_{q,k} B(q,k) = \Omega_d(1), \tag{43}$$

$$\max_{k=0,\ldots,\mathsf{s}'} q^{\mathsf{s}'-k+1} \xi_{q,q-k} B(q,q-k) = O_d(1). \tag{44}$$

Assumption 1 will be verified by standard kernels, e.g., the Gaussian kernel. We discuss this assumption in Section C.2 and present sufficient conditions on the activation function $\sigma$ for its associated CNTK to verify Assumption 1.

Recall that we denoted $L^2(\mathscr{Q}^d, \mathrm{Loc}_q)$ the space of local functions, i.e., that can be decomposed as $f(\boldsymbol{x}) = \sum_{k \in [d]} f_k(\boldsymbol{x}_{(k)})$. Denote $h_{q,>\ell}$ the inner-product kernel $h_q$ with its $(\ell+1)$-first Gegenbauer coefficients set to 0, i.e.,

$$h_{q,>\ell}(\langle \boldsymbol{u}, \boldsymbol{v} \rangle / q) = \sum_{k=\ell+1}^q \xi_{q,k} B(\mathscr{Q}^q; k) Q_k^{(q)}(\langle \boldsymbol{u}, \boldsymbol{v} \rangle) , \tag{45}$$

for any $\boldsymbol{u}, \boldsymbol{v} \in \mathcal{Q}^q$. The following result is a consequence of the general theorem on the generalization error of KRR in [38].

**Theorem 7** (Test error of CK in high dimension). *Let $\{f_d \in L^2(\mathcal{Q}^d, \mathrm{Loc}_q)\}_{q \geq 1}$ be a sequence of local functions. Let $(\boldsymbol{x}_i)_{i \in [n(d)]} \sim_{\text{i.i.d.}} \mathrm{Unif}(\mathcal{Q}^d)$ and $y_i = f_d(\boldsymbol{x}_i) + \varepsilon_i$ with $\varepsilon_i \sim_{\text{i.i.d.}} \mathsf{N}(0, \sigma_\varepsilon^2)$. Assume $d \cdot q^{\mathsf{s}-1+\delta} \leq n \leq d \cdot q^{\mathsf{s}-\delta}$ for some $\delta > 0$ and let $\{h_q\}_{q \geq 1}$ be a sequence of activation functions satisfying Assumption 1 at level $\mathsf{s}$. Consider $\{H^{\mathsf{CK},d}\}_{q \geq 1}$ the sequence of convolutional kernels associated to $\{h_q\}_{q \geq 1}$ as defined in Eq. (3). Then the following holds for the solution $\hat{f}_\lambda$ of KRR with kernels $\{H^{\mathsf{CK},d}\}_{q \geq 1}$.*

*For any regularization parameter $\lambda \geq 0$, define the effective regularization $\lambda_{\text{eff}} := \lambda + h_{q,>\mathsf{s}}(1)$. Then for any $\eta > 0$, we have*

$$\left\| \hat{f}_\lambda - \hat{f}^{\text{eff}}_{\lambda_{\text{eff}}} \right\|_{L^2}^2 = o_{d,\mathbb{P}}(1) \cdot (\|f_d\|_{L^{2+\eta}}^2 + \sigma_\varepsilon^2). \tag{46}$$

The proof of Theorem 7 is deferred to Section C.4.

Let us expound on the predictions of Theorem 7. First, recall that $\hat{f}^{\text{eff}}_{\lambda_{\text{eff}}}$ is given explicitly in Eq. (41) by a shrinkage operator with parameter $\lambda_{\text{eff}}$. From Assumption 1 and taking $\lambda = 0$, the shrinkage operator is of order 1

$$\lambda_{\text{eff}} = h_{q,>\mathsf{s}}(1) = \sum_{\ell=\mathsf{s}+1}^q \xi_{q,\ell} B(\mathcal{Q}^q; \ell) = \Theta_q(1).$$

From the eigendecomposition of $H^{\mathsf{CK}}$ introduced in Proposition 1, KRR fits perfectly $f_\star$ along the eigendirection $Y_S$ with $|S| = \ell$ if $n \cdot \xi_{d,\ell} r(S)/d \gg \lambda_{\text{eff}}$, while it does not fit this eigendirection at all if $n \cdot \xi_{d,\ell} r(S)/d \leq \lambda_{\text{eff}}$. Consider $n = d \cdot q^{\mathsf{s}-1+\alpha}$:

- KRR fits the eigendirections corresponding to the homogeneous polynomials of degree $\mathsf{s} - 1$ and less, and of degree $\mathsf{s}$ for subsets $S$ such that $\gamma(S) \ll q - q^{1-\alpha}$.
- KRR does not fit at all the eigendirections correpsonding to homogeneous polynomials of degree $\mathsf{s} + 1$ and larger, and degree $\mathsf{s}$ for subsets $S$ such that $\gamma(S) \gg q - q^{1-\alpha}$.

In words, for $d \cdot q^{\mathsf{s}-1} \ll n \ll d \cdot q^{\mathsf{s}}$, KRR fits at least a degree-$(\mathsf{s} - 1)$ polynomial approximation to $f_\star$ and at most a degree-$\mathsf{s}$ polynomial approximation. As $n$ increases from $d \cdot q^{\mathsf{s}-1}$ to $d \cdot q^{\mathsf{s}}$, KRR first fits degree-$\mathsf{s}$ homogeneous polynomials that have smaller diameter $\gamma(S)$ (i.e., 'more localized').

**Test error of CK with global average pooling:** we consider the kernel $H^{\mathsf{CK}}_{\mathsf{GP}}$ given by a convolutional layer followed by global average pooling:

$$H^{\mathsf{CK}}_{\mathsf{GP}}(\boldsymbol{x}, \boldsymbol{y}) = \frac{1}{d} \sum_{k,k' \in [d]} h\left( \langle \boldsymbol{x}_{(k)}, \boldsymbol{y}_{(k')} \rangle / q \right), \tag{47}$$

In addition to the genericity condition, we will assume that the kernels $\{h_q\}_{q \geq 1}$ verify the following differentiability condition.

**Assumption 2** (Differentiability assumption on $\{h_q\}_{q \geq 1}$ at level $\mathsf{s} \in \mathbb{N}$). *For $\{h_q\}_{q \geq 1}$ a sequence of inner-product kernels $h_q : \mathbb{R} \to \mathbb{R}$, we assume the following conditions to hold. There exists $v \geq \max(2/\delta, \mathsf{s})$ where $\delta > 0$ verifies $q \geq d^\delta$ such that $h_q$ is $(v+1)$-differentiable and for $k \leq v$,*

$$\sup_{\gamma \in [-1,1]} \left| h_{q,>v}^{(v+1)}(\gamma) \right| \leq O_q(1),$$

$$\left| h_{q,>v}^{(k)}(0) \right| \leq O_q(q^{-(v+1-k)/2}),$$

*where we denoted $h_{q,>v}$ the truncated inner-product kernel $h_q$ as in Eq. (45).*

Assumption 2 is used to extend the following theorem to non-polynomial kernel $h_q$ (in particular, it is trivially verified for polynomial kernels by taking $v$ larger than the degree of $h_q$). This assumption is difficult to check in practice, however we provide some examples where it holds in Appendix C.2.

Recall that we denoted $L^2(\mathcal{Q}^d, \mathrm{CycLoc}_q)$ the space of functions that are given by the convolution of a function $g : \mathbb{R}^q \to \mathbb{R}$ with the image $\boldsymbol{x} \in \mathcal{Q}^d$, i.e., $f(\boldsymbol{x}) = \sum_{k \in [d]} g(\boldsymbol{x}_{(k)})$.

**Theorem 8** (Test error of CK with GP in high dimension). *Let $\{f_d \in L^2(\mathcal{Q}^d, \mathrm{CycLoc}_q)\}_{q \geq 1}$ be a sequence of convolutional functions. Assume $q^{\mathsf{s}-1+\delta} \leq n \leq q^{\mathsf{s}-\delta}$ for some $\delta > 0$ and let $\{h_q\}_{q \geq 1}$ be a sequence of activation functions satisfying Assumptions 1 and 2 at level $\mathsf{s}$. Consider $\{H_{\mathsf{GP}}^{\mathsf{CK},d}\}_{q \geq 1}$ the sequence of convolutional kernels with global pooling associated to $\{h_q\}_{q \geq 1}$ as defined in Eq. (47). Then the solution $\hat{f}_\lambda$ of KRR with kernels $\{H_{\mathsf{GP}}^{\mathsf{CK},d}\}_{q \geq 1}$ verifies Eq. (46) with $\lambda_{\mathrm{eff}} := \lambda + h_{q,>\mathsf{s}}(1)$.*

The proof of Theorem 8 is deferred to Section C.5.

The predictions of Theorem 8 are similar to the ones of Theorem 7 but with a factor $d$ gain in statistical efficiency: this is due to the eigenvalues of $H_{\mathsf{GP}}^{\mathsf{CK}}$ being a factor $d$ larger than for $H^{\mathsf{CK}}$. Therefore, with global average pooling, for $q^{\mathsf{s}-1} \ll n \ll q^{\mathsf{s}}$, KRR fits at least a degree-$(\mathsf{s}-1)$ invariant polynomial approximation to $f_\star$ and at most a degree-$\mathsf{s}$ invariant polynomial approximation. As $n$ increases from $q^{\mathsf{s}-1}$ to $q^{\mathsf{s}}$, KRR first degree-$\mathsf{s}$ invariant homogeneous polynomials with increasing diameter $\gamma(S)$.

**Test error of CK with local average pooling:** In the case of local average pooling with $\omega < d$, the eigenvalues are harder to control. Indeed, we have mixing of the eigenvalues between polynomials of different degree: there exists $j, j' \in [d]$ such that $\xi_{q,\ell}\kappa_j \ll \xi_{q,\ell+1}\kappa_{j'}$. The eigenvalues are not ordered in increasing degree of their associated eigenfunctions anymore. While this case is potentially tractable with a more careful analysis, we instead introduce a simplified kernel which we believe qualitatively captures the statistical behavior of local average pooling.

Assume $q \leq \omega/2$ and $\omega$ is a divisor of $d$. Denote $\boldsymbol{x}^{(k\omega)} = (x_{k\omega+1}, \ldots, x_{k\omega+\omega})$ the $k$-th segment of length $\omega$ in $[d]$ and $\boldsymbol{x}_{(i)}^{(k\omega)} = (x_{k\omega+i}, \ldots, x_{k\omega+q+i})$ the patch of size $q$ with cyclic convention in $\{k\omega+1, \ldots, k\omega+\omega\}$. Consider the following convolutional kernel with 'non-overlapping' average pooling:

$$H_\omega^{\mathsf{CK},\mathsf{NO}}(\boldsymbol{x}, \boldsymbol{y}) = \frac{1}{\omega} \sum_{k \in [d/\omega]} \sum_{i,j \in [\omega]} h_q\big(\langle \boldsymbol{x}_{(i)}^{(k\omega)}, \boldsymbol{y}_{(j)}^{(k\omega)} \rangle / q \big), \tag{48}$$

In words, $H_\omega^{\mathsf{CK},\mathsf{NO}}$ is the combination of $d/\omega$ non-overlapping convolutional kernels with global average pooling on images of size $\omega$:

$$
\begin{aligned}
H_\omega^{\mathsf{CK},\mathsf{NO}} &= \sum_{k \in [d/\omega]} H_{\mathsf{GP}}^{\mathsf{CK}}\big(\boldsymbol{x}^{(k\omega)}, \boldsymbol{y}^{(k\omega)}\big) \\
&= \sum_{\ell=0}^{q} \xi_{q,\ell} \sum_{k \in [d/\omega]} \sum_{S \in \mathcal{C}_\ell} \psi_{k,S}(\boldsymbol{x}) \psi_{k,S}(\boldsymbol{y}),
\end{aligned}
\tag{49}
$$

where $\psi_{k,S}(\boldsymbol{x}) = \frac{1}{\sqrt{\omega}} \sum_{i \in [\omega]} Y_{i+S}(\boldsymbol{x}^{(k\omega)})$ where $i + S$ is the translated set with cyclic convention in $[\omega]$.

Denote $L^2(\mathcal{Q}^d, \mathrm{LocCycLoc}_q)$ the RKHS associated to $H_\omega^{\mathsf{CK},\mathsf{NO}}$, which contains functions that are locally convolutions on segments of size $\omega$. For this simplified model, the proof of Theorem 8 can be easily adapted and we obtain the following result:

**Corollary 1** (Test error of CK with NO pooling in high dimension). *Let $\{f_d \in L^2(\mathcal{Q}^d, \mathrm{LocCycLoc}_q)\}_{q \geq 1}$ be a sequence of local convolutional functions. Assume $(d/\omega) \cdot q^{\mathsf{s}-1+\delta} \leq n \leq (d/\omega) \cdot q^{\mathsf{s}-\delta}$ for some $\delta > 0$ and let $\{h_q\}_{q \geq 1}$ be a sequence of activation functions satisfying Assumptions 1 and 2 at level $\mathsf{s}$. Consider $\{H_\omega^{\mathsf{CK},\mathsf{NO},d}\}_{q \geq 1}$ the sequence of convolutional kernels with non-overlapping pooling associated to $\{h_q\}_{q \geq 1}$ as defined in Eq. (48). Then the solution $\hat{f}_\lambda$ of KRR with kernels $\{H_\omega^{\mathsf{CK},\mathsf{NO},d}\}_{q \geq 1}$ verifies Eq. (46) with $\lambda_{\mathrm{eff}} := \lambda + \frac{d}{\omega} h_{q,>\mathsf{s}}(1)$.*

Corollary 1 shows that $H_\omega^{\mathsf{CK},\mathsf{NO}}$ enjoys a factor $\omega$ gain in statistical efficiency compared to $H^{\mathsf{CK}}$, due to a factor $\omega$ smaller effective ridge regularization. Therefore, with (non-overlapping) local average pooling, for $(d/\omega) \cdot q^{\mathsf{s}-1} \ll n \ll (d/\omega) \cdot q^{\mathsf{s}}$, KRR fits degree-$(\mathsf{s}-1)$ locally invariant polynomials and none of the polynomials of degree-$(\mathsf{s}+1)$ and larger. Heuristically, we see that this yields the same statistical efficiency than $H^{\mathsf{CK}}$ for $\omega = 1$ and $H_{\mathsf{GP}}^{\mathsf{CK}}$ for $\omega = d$, and interpolates between the two cases for $1 < \omega < d$.

**Test error of convolutional kernels with downsampling:** We consider adding a downsampling operation to the previous kernels. Let $\Delta$ be a constant and a divisor of $d$ and $\omega$ and consider the following 'downsampled' kernels:

$$H_\Delta^{\mathsf{CK}}(\boldsymbol{x}, \boldsymbol{y}) = \Delta \sum_{k \in [d/\Delta]} h\big(\langle \boldsymbol{x}_{(k\Delta)}, \boldsymbol{y}_{(k\Delta)} \rangle / q\big), \tag{50}$$

$$H_{\mathsf{GP},\Delta}^{\mathsf{CK}}(\boldsymbol{x}, \boldsymbol{y}) = \frac{\Delta}{d} \sum_{k,k' \in [d/\Delta]} h\big(\langle \boldsymbol{x}_{(k\Delta)}, \boldsymbol{y}_{(k'\Delta)} \rangle / q\big), \tag{51}$$

$$H_{\omega,\Delta}^{\mathsf{CK,NO}}(\boldsymbol{x}, \boldsymbol{y}) = \sum_{k \in [d/\omega]} H_{\mathsf{GP},\Delta}^{\mathsf{CK}}\big(\boldsymbol{x}^{(k\omega)}, \boldsymbol{y}^{(k\omega)}\big). \tag{52}$$

We can easily adapt the proofs of Theorems 7 and 8, and Corollary 1 to these kernels. In particular, their conclusions do not change (for any constant $\Delta$) and downsampling do not provide a statistical advantage.

### C.2  Checking the assumptions

In this section, we discuss Assumptions 1 and 2 and present sufficient conditions for them to be verified.

**Genericity assumption:** Recall that the inner-product kernel $h_q : \mathbb{R} \to \mathbb{R}$ has the following eigendecomposition on $\mathscr{Q}^q$ as

$$h_q\big(\langle \boldsymbol{u}, \boldsymbol{v} \rangle / q\big) = \sum_{\ell=0}^{q} \xi_{q,\ell} \sum_{S \subseteq [q], |S|=\ell} Y_S(\boldsymbol{u}) Y_S(\boldsymbol{v}).$$

The genericity assumption amounts to: 1) A universality condition in Eqs. (42) and (43): if $P_k h(\langle \mathbf{1}, \cdot \rangle / q) = 0$, then $h$ does not learn degree-$k$ homogeneous polynomials; 2) A constant order scaling of the self-induced regularization $h_{q,>\mathsf{s}}(1)$, from $h_q(1) \leq C$ and Eq. (43) with $\mathsf{s}'$, i.e., $h_{q,>\mathsf{s}}(1) \leq h_q(1) = O_q(1)$ and $h_{q,>\mathsf{s}}(1) \geq \xi_{q,\mathsf{s}'} B(q, \mathsf{s}') = \Omega_q(1)$; 3) The last eigenvalues decay sufficiently fast in Eq. (44) in order to avoid pathological cases.

For generic kernels, we have typically $\xi_{q,\ell} \asymp q^{-\ell}$ (for fix $\ell$). For example, if $h$ is smooth, $\xi_{q,\ell} = q^{-\ell}(h^{(k)}(0) + o_q(1))$ and it is sufficient to have $h^{(k)}(0) > 0$. See Appendix D.2 in [38] for a proof of Eq. (44) when $h$ is sufficiently smooth.

Below, we present instead sufficient conditions on the activation $\sigma$ such that the induced neural tangent kernel verifies the 'genericity' assumption. More precisely, we display sufficient conditions on the sequence $\{\sigma_q\}_{q \geq 1}$ of activation functions $\sigma_q : \mathbb{R} \to \mathbb{R}$, such that the induced neural tangent kernels $\{h_q\}_{q \geq 1}$ verifies Assumption 1, where $h_q$ was derived in Section A.2 and is given by ($\boldsymbol{u}, \boldsymbol{v} \in \mathscr{Q}^q$)

$$h_q(\langle \boldsymbol{u}, \boldsymbol{v} \rangle / q) := h_q^{(1)}(\langle \boldsymbol{u}, \boldsymbol{v} \rangle / q) + h_q^{(2)}(\langle \boldsymbol{u}, \boldsymbol{v} \rangle / q), \tag{53}$$

where

$$h_q^{(1)}(\langle \boldsymbol{u}, \boldsymbol{v} \rangle / q) = \mathbb{E}_{\boldsymbol{w} \sim \mathrm{Unif}(\mathscr{Q}^q)}\big[\sigma_q(\langle \boldsymbol{u}, \boldsymbol{w} \rangle / \sqrt{q}) \sigma_q(\langle \boldsymbol{v}, \boldsymbol{w} \rangle / \sqrt{q})\big], \tag{54}$$

$$h_q^{(2)}(\langle \boldsymbol{u}, \boldsymbol{v} \rangle / np.sqrt(q)) = \mathbb{E}_{\boldsymbol{w} \sim \mathrm{Unif}(\mathscr{Q}^q)}\big[\sigma_q'(\langle \boldsymbol{u}, \boldsymbol{w} \rangle / \sqrt{q}) \sigma_q'(\langle \boldsymbol{v}, \boldsymbol{w} \rangle / \sqrt{q}) \langle \boldsymbol{u}, \boldsymbol{v} \rangle\big] / q. \tag{55}$$

**Assumption 3** (Assumptions on $\{\sigma_q\}_{q \geq 1}$ at level $\mathsf{s} \in \mathbb{N}$)**.** *For $\{\sigma_q\}_{q \geq 1}$ a sequence of functions $\sigma_q : \mathbb{R} \to \mathbb{R}$, we assume the following conditions to hold. There exists $\mathsf{s}' \geq 1/\delta + 2\mathsf{s} + 3$ where $\delta > 0$ verifies $q \geq d^\delta$, such that*

*(a) The function $\sigma_q$ is differentiable and there exists $c_0 > 0$ and $c_1 < 1$ independent of $q$, such that $|\sigma_q(x)|, |\sigma_q'(x)| \leq c_0 \exp(c_1 x^2 / 2)$.*

*(b) We have*

$$\min_{k \leq \mathsf{s}-1} q^{\mathsf{s}-1-k} \|P_k \sigma_q(\langle \boldsymbol{e}, \cdot \rangle / \sqrt{q})\|_{L^2(\mathscr{Q}^q)} = \Omega_q(1), \tag{56}$$

$$\min_{k \in \{\mathsf{s}, \mathsf{s}+1, \mathsf{s}'\}} \|P_k \sigma_q(\langle \boldsymbol{e}, \cdot \rangle / \sqrt{q})\|_{L^2(\mathscr{Q}^q)} = \Omega_q(1), \tag{57}$$

*where $\boldsymbol{e} \in \mathscr{Q}^q$ is arbitrary.*

*(c) We have for a fixed $\delta > 0$*

$$\max_{k=0,\ldots,\mathsf{s}'} q^{\mathsf{s}'-k+1}\|\mathsf{P}_k\sigma_q(\langle\boldsymbol{e},\cdot\rangle/\sqrt{q})\|_{L^2(\mathscr{Q}^q)} = O_q(1)\,, \tag{58}$$

$$\max_{k=0,\ldots,\mathsf{s}'} q^{\mathsf{s}'-k+1}\|\mathsf{P}_k\sigma_q'(\langle\boldsymbol{e},\cdot\rangle/\sqrt{q})\|_{L^2(\mathscr{Q}^q)} = O_q(1)\,. \tag{59}$$

**Proposition 6.** *Consider a sequence $\{\sigma_q\}_{q\geq 1}$ of activation functions $\sigma_q : \mathbb{R} \to \mathbb{R}$ that satisfies Assumption 3. Let $\{h_q\}_{q\geq 1}$ be the sequence of neural tangent kernels associated to $\{\sigma_q\}_{q\geq 1}$ as defined in Eq. (53). Then the sequence $\{h_q\}_{q\geq 1}$ satisfies the 'genericity' Assumption 1.*

**Differentiability assumption:** As mentioned in the previous section, this condition is required in our proof technique to extend Theorem 8 to non-polynomial kernel functions. While we believe that weaker conditions should be sufficient, we leave checking them to future work. Note that Assumption 2 was proved for $\boldsymbol{x} \sim \mathrm{Unif}(\mathbb{S}^{d-1}(\sqrt{d}))$ and $h_q(\langle\boldsymbol{x},\boldsymbol{y}\rangle/q) = \mathbb{E}_{\boldsymbol{w}}\{\sigma(\langle\boldsymbol{x},\boldsymbol{w}\rangle)\sigma(\langle\boldsymbol{y},\boldsymbol{w}\rangle)\}$ for $\boldsymbol{w} \sim \mathrm{Unif}(\mathbb{S}^{d-1}(1))$, given that $\sigma$ satisfies some differentiability conditions, in [39].

## C.3 Proof of Proposition 6

*Proof of Proposition 6.* **Step 1. Effective activation function.**

Let us decompose both functions $\sigma_q$ and $\sigma_q'$ in the Gegenbauer polynomial on the hypercube basis:

$$\sigma_q(\langle\boldsymbol{u},\boldsymbol{v}\rangle/\sqrt{q}) = \sum_{\ell=0}^{q}\chi_{q,\ell}B(\mathscr{Q}^q;\ell)Q_\ell^{(q)}(\langle\boldsymbol{u},\boldsymbol{v}\rangle)\,, \tag{60}$$

$$\sigma_q'(\langle\boldsymbol{u},\boldsymbol{v}\rangle/\sqrt{q}) = \sum_{\ell=0}^{q}\kappa_{q,\ell}B(\mathscr{Q}^q;\ell)Q_\ell^{(q)}(\langle\boldsymbol{u},\boldsymbol{v}\rangle)\,, \tag{61}$$

where we recall $B(\mathscr{Q}^q;\ell) = \binom{d}{\ell}$ and (for $\boldsymbol{e} \in \mathscr{Q}^q$ arbitrary)

$$\chi_{q,\ell}(\sigma_q) = \mathbb{E}_{\boldsymbol{u}\sim\mathrm{Unif}(\mathscr{Q}^q)}\big[\sigma_q(\langle\boldsymbol{u},\boldsymbol{e}\rangle/\sqrt{q})Q_\ell^{(q)}(\langle\boldsymbol{u},\boldsymbol{e}\rangle)\big],$$

$$\kappa_{q,\ell}(\sigma_q') = \mathbb{E}_{\boldsymbol{u}\sim\mathrm{Unif}(\mathscr{Q}^q)}\big[\sigma_q'(\langle\boldsymbol{u},\boldsymbol{e}\rangle/\sqrt{q})Q_\ell^{(q)}(\langle\boldsymbol{u},\boldsymbol{e}\rangle)\big].$$

From the definition of $h_q^{(1)}$ in Eq. (54) and the eigendecomposition (60), we have

$$h_q^{(1)}(\langle\boldsymbol{u},\boldsymbol{v}\rangle/q) = \sum_{\ell=0}^{q}\chi_{q,\ell}^2 B(\mathscr{Q}^q;\ell)Q_\ell^{(q)}(\langle\boldsymbol{u},\boldsymbol{v}\rangle).$$

Similarly, from the definition of $h_q^{(2)}$ in Eq. (55), the eigendecomposition (61) and using Lemma 1 stated below, we get

$$h_q^{(2)}(\langle\boldsymbol{u},\boldsymbol{v}\rangle/q) = \sum_{\ell=0}^{q}\kappa_{q,\ell}^2 B(\mathscr{Q}^q;\ell)Q_\ell^{(q)}(\langle\boldsymbol{u},\boldsymbol{v}\rangle)\langle\boldsymbol{u},\boldsymbol{v}\rangle/q = \sum_{\ell=0}^{q}\zeta_{q,\ell}^2 B(\mathscr{Q}^q;\ell)Q_\ell^{(q)}(\langle\boldsymbol{u},\boldsymbol{v}\rangle),$$

where

$$\zeta_{q,\ell}^2 = \frac{\ell}{q}\kappa_{q,\ell-1}^2 + \frac{q-\ell}{q}\kappa_{q,\ell+1}^2. \tag{62}$$

We can therefore define $\pi_{q,\ell} = \sqrt{\chi_{q,\ell}^2 + \zeta_{q,\ell}^2}$ and $\sigma_{\mathrm{eff},q}(\langle\cdot,\cdot\rangle/\sqrt{q}) : \mathscr{Q}^q \times \mathscr{Q}^q \to \mathbb{R}$ by

$$\sigma_{\mathrm{eff},q}(\langle\boldsymbol{u},\boldsymbol{v}\rangle/\sqrt{q}) = \sum_{\ell=0}^{q}\pi_{q,\ell}B(\mathscr{Q}^q;\ell)Q_\ell^{(q)}(\langle\boldsymbol{u},\boldsymbol{v}\rangle),$$

such that the NT kernel (53) can be written as the kernel of the effective activation $\sigma_{\mathrm{eff},q}$:

$$h_q(\langle\boldsymbol{u},\boldsymbol{v}\rangle/q) = \mathbb{E}_{\boldsymbol{\theta}\sim\mathrm{Unif}(\mathscr{Q}^d)}\Big[\sigma_{\mathrm{eff},q}(\langle\boldsymbol{u},\boldsymbol{\theta}\rangle/\sqrt{q})\sigma_{\mathrm{eff},q}(\langle\boldsymbol{y}_{(k)},\boldsymbol{\theta}\rangle/\sqrt{q})\Big]$$
$$= \sum_{\ell=0}^{q}\pi_{q,\ell}^2 B(\mathscr{Q}^q;\ell)Q_\ell^{(q)}(\langle\boldsymbol{u},\boldsymbol{v}\rangle)\,. \tag{63}$$

We will show that $h_q$ with Gegenbauer coefficients $\xi_{q,\ell} := \pi_{q,\ell}^2$ verifies Assumption 1.

**Step 2. Decay of the eigenvalues.**

Recall that the sequence $\{\sigma_q\}_{q \geq 1}$ satisfies Assumption 3 at level $\mathsf{s}$. From Assumption 3.$(a)$ (for example by adapting the proof of Lemma C.1 in [24] to the hypercube), there exists $C > 0$ such that

$$h_q(1) = \|\sigma_{\mathrm{eff},q}\|_{L^2(\mathcal{Q}^q)}^2 = h_q^{(1)}(1) + h_q^{(2)}(1) = \|\sigma_q\|_{L^2(\mathcal{Q}^q)}^2 + \|\sigma_q'\|_{L^2(\mathcal{Q}^q)}^2 \leq C,$$

and we deduce that $\chi_{q,\ell}^2, \kappa_{q,\ell}^2, \pi_{q,\ell}^2 = O_q(B(\mathcal{Q}^q; \ell)^{-1})$. Using that $B(\mathcal{Q}^q; \ell) = \binom{q}{\ell}$, we deduce that for any fixed $\ell$, $\chi_{q,\ell}^2, \kappa_{q,\ell}^2, \pi_{q,\ell}^2 = O_q(q^{-\ell})$. Furthermore, from Assumption 3.$(c)$, we have for $k = 0, \ldots, \mathsf{s}' + 1$,

$$\chi_{q,q-k}^2 = B(\mathcal{Q}^q; q-k)^{-1} \|\mathsf{P}_{q-k}\sigma_q\|_{L^2(\mathcal{Q}^q)}^2 = O_q(q^{-\mathsf{s}'-1}),$$

$$\kappa_{q,q-k}^2 = B(\mathcal{Q}^q; q-k)^{-1} \|\mathsf{P}_{q-k}\sigma_q'\|_{L^2(\mathcal{Q}^q)}^2 = O_q(q^{-\mathsf{s}'-1}),$$

By Eq. (62) and the definition of $\pi_{q,\ell}^2$, we have $\pi_{q,q-k}^2 = O_d(q^{-\mathsf{s}'-1})$ for any $k \leq \mathsf{s}'$, which verifies Eq. (44) in Assumption 1.

Furthermore, by Assumption 3.$(b)$, using that $\chi_{q,k}^2 = B(\mathcal{Q}^q; k)^{-1} \|\mathsf{P}_k\sigma_q\|_{L^2(\mathcal{Q}^q)}^2$ and $\xi_{q,k}^2 \geq \chi_{q,k}^2$, we get

$$\min_{k \leq \mathsf{s}-1} \xi_{q,k}^2 = \Omega_q(q^{-\mathsf{s}+1}),$$

and

$$\xi_{q,\mathsf{s}}^2 = \Omega_q(q^{-\mathsf{s}}), \qquad \xi_{q,\mathsf{s}+1}^2 = \Omega_q(q^{-\mathsf{s}-1}), \qquad \xi_{q,\ell'}^2 = \Omega_q(q^{-\ell'}).$$

In particular, this implies that $\|\sigma_{\mathrm{eff},d,>\mathsf{s}}\|_{L^2(\mathcal{Q}^q)}^2 \geq \|\mathsf{P}_{\mathsf{s}'}\sigma_q\|_{L^2(\mathcal{Q}^q)}^2 = \Omega_q(1)$. $\qquad\square$

**Lemma 1.** *Let $\ell$ be an integer such that $0 \leq \ell \leq q$. Consider the following Gegenbauer polynomial defined on the $q$-dimensional hypercube (see Section D): for $\boldsymbol{x}, \boldsymbol{y} \in \mathcal{Q}^q$,*

$$Q_\ell^{(q)}(\langle \boldsymbol{x}, \boldsymbol{y} \rangle) = \frac{1}{B(\mathcal{Q}^q; \ell)} \sum_{S \subset [q], |S|=\ell} Y_S(\boldsymbol{x}) Y_S(\boldsymbol{y}),$$

*where we recall the definition of the homogeneous polynomial $Y_S(\boldsymbol{x}) = \boldsymbol{x}^S = \prod_{i \in S} x_i$. We have*

$$Q_\ell^{(q)}(\langle \boldsymbol{x}, \boldsymbol{y} \rangle) \langle \boldsymbol{x}, \boldsymbol{y} \rangle / q = \frac{\ell}{q} Q_{\ell-1}^{(q)}(\langle \boldsymbol{x}, \boldsymbol{y} \rangle) + \frac{q-\ell}{q} Q_{\ell+1}^{(q)}(\langle \boldsymbol{x}, \boldsymbol{y} \rangle),$$

*with the convention $Q_{-1}^{(q)} = Q_{q+1}^{(q)} = 0$.*

*Proof of Lemma 1.* Consider $1 \leq \ell \leq q - 1$. We have

$$Q_\ell^{(q)}(\langle \boldsymbol{x}, \boldsymbol{y} \rangle) \langle \boldsymbol{x}, \boldsymbol{y} \rangle / q = \frac{1}{q B(\mathcal{Q}^q; \ell)} \sum_{S \subset [q], |S|=\ell} \sum_{i \in [q]} Y_S(\boldsymbol{x}) x_i \cdot Y_S(\boldsymbol{y}) y_i.$$

We have $Y_S(\boldsymbol{x}) x_i = Y_{S \cup \{i\}}(\boldsymbol{x})$ if $i \notin S$, and $Y_S(\boldsymbol{x}) x_i = Y_{S \setminus \{i\}}(\boldsymbol{x})$ if $i \in S$. Hence, the above sum contains sets of size $\ell - 1$ and $\ell + 1$. For each set $S \subset [q]$ with $|S| = \ell - 1$, there $q + 1 - \ell$ sets $|\tilde{S}| = \ell$, such that by removing one element we can obtain $S$. For each set $S \subset [q]$ with $|S| = \ell + 1$, there $\ell + 1$ sets $|\tilde{S}| = \ell$, such that by adding one element we can obtain $S$.

We deduce that

$$Q_\ell^{(q)}(\langle \boldsymbol{x}, \boldsymbol{y} \rangle) \langle \boldsymbol{x}, \boldsymbol{y} \rangle / q$$
$$= \frac{q+1-\ell}{q B(\mathcal{Q}^q; \ell)} \sum_{S \subset [q], |S|=\ell-1} Y_S(\boldsymbol{x}) Y_S(\boldsymbol{y}) + \frac{\ell+1}{q B(\mathcal{Q}^q; \ell)} \sum_{S \subset [q], |S|=\ell+1} Y_S(\boldsymbol{x}) Y_S(\boldsymbol{y}).$$

Using $B(\mathcal{Q}^q; \ell) = \binom{q}{\ell}$, we obtain

$$Q_\ell^{(q)}(\langle \boldsymbol{x}, \boldsymbol{y} \rangle) \langle \boldsymbol{x}, \boldsymbol{y} \rangle / q = \frac{\ell}{q} Q_{\ell-1}^{(q)}(\langle \boldsymbol{x}, \boldsymbol{y} \rangle) + \frac{q-\ell}{q} Q_{\ell+1}^{(q)}(\langle \boldsymbol{x}, \boldsymbol{y} \rangle).$$

The cases $\ell = 0$ and $\ell = q$ are straightforward. $\qquad\square$

## C.4 Proof of Theorem 7

Let $\{d(q)\}_{q\geq 1}$ be a sequence of integers with $2q \leq d(q) \leq q^{1/\delta}$ for some $\delta > 0$. We will denote $d = d(q)$ for simplicity. Consider $\boldsymbol{x} \sim \text{Unif}(\mathscr{Q}^d)$, $dq^{\mathsf{s}-1+\delta} \leq n \leq dq^{\mathsf{s}-\delta}$ for some $\delta > 0$ and a sequence of inner-product kernels $\{h_q\}_{q\geq 1}$ that satisfies Assumption 1 at level $\mathsf{s}$. We consider the vanilla one-layer convolutional kernel

$$H^{\mathsf{CK},d}(\boldsymbol{x}, \boldsymbol{y}) = \frac{1}{d} \sum_{k=1}^{d} h_q(\langle \boldsymbol{x}_{(k)}, \boldsymbol{y}_{(k)} \rangle / q).$$

Theorem 7 is a consequence of Theorem 4 in [38] where we take $\mathcal{X}_d = \mathscr{Q}^d$, $\nu_d = \text{Unif}(\mathcal{X}_d)$ and $\mathcal{D}_d = L^2(\mathscr{Q}^d, \text{Loc}_q) \subset L^2(\mathscr{Q}^d)$. The proof amounts to checking that $\{H^{\mathsf{CK},d}\}_{q\geq 1}$ verifies the kernel concentration properties and eigenvalue condition (see Section 3.2 in [38]). We borrow some of the notations introduced in [38] and we refer the reader to their Section 2.1.

*Proof of Theorem 7.* **Step 1. Diagonalization of the kernel and choosing** $\mathsf{m} = \mathsf{m}(q)$**.**

From Proposition 1, we have the following diagonalization of $H^{\mathsf{CK},d}$:

$$H_d(\boldsymbol{x}, \boldsymbol{y}) := H^{\mathsf{CK},d}(\boldsymbol{x}, \boldsymbol{y}) = \frac{1}{d} \sum_{\ell=0}^{q} \sum_{S \in \mathcal{E}_\ell} \xi_{q,\ell} r(S) \cdot Y_S(\boldsymbol{x}) Y_S(\boldsymbol{y}),$$

where $r(\emptyset) = d$ and $r(S) = q + 1 - \gamma(S)$ for $S \subset [q] \setminus \{\emptyset\}$, and we recall $\mathcal{E}_\ell = \{S \subseteq [d] : |S| = \ell, \gamma(S) \leq q\}$. Using that $B(\mathscr{Q}^q; \ell) = \Theta_q(q^\ell)$, $\xi_{q,\ell} B(\mathscr{Q}^q; \ell) \leq h_q(1)$ and Assumption 1, we have

$$\min_{\ell \leq \mathsf{s}-1} \xi_{q,\ell} = \Omega_q(q^{-\mathsf{s}+1}), \qquad\qquad \xi_{q,\mathsf{s}} = \Theta_q(q^{-\mathsf{s}}),$$

$$\xi_{q,\mathsf{s}+1} = \Theta_q(q^{-\mathsf{s}-1}), \qquad \sup_{\ell \geq \mathsf{s}+2} \xi_{q,\ell} = O_q(q^{-\mathsf{s}-2}). \tag{64}$$

Further define $\mathcal{E}_{\ell,h} = \{S \in \mathcal{E}_\ell : \gamma(S) = h\}$ for $h = \ell, \dots, q$. It is easy to check that $|\mathcal{E}_{\ell,h}| = d\binom{h-2}{\ell-2}$ and

$$|\mathcal{E}_\ell| = \sum_{h=\ell}^{q} |\mathcal{E}_{\ell,h}| = d\sum_{h=\ell}^{q} \binom{h-2}{\ell-2} = d\binom{q-1}{\ell-1},$$

and therefore $|\mathcal{E}_\ell| = \Theta_q(d \cdot q^{\ell-1})$.

Denote $\{\lambda_{q,j}\}_{j\geq 1}$ the eigenvalues $\{\xi_{q,\ell} r(S)/d\}_{\ell=0,\dots,q; S \in \mathcal{E}_\ell}$ in nonincreasing order, and $\{\psi_{q,j}\}_{j\geq 1}$ the reordered eigenfunctions. Set $\mathsf{m}$ to be the number of eigenvalues such that $\lambda_{q,j} > q\xi_{q,\mathsf{s}+1}/d$ (recall $q\xi_{q,\mathsf{s}+1} = \Theta_d(q^{-\mathsf{s}})$). Denote $\alpha = q\xi_{q,\mathsf{s}+1}/\xi_{q,\mathsf{s}}$. From the bounds (64) on $\xi_{q,\mathsf{s}+1}$ and $\xi_{q,\mathsf{s}}$, we have $\alpha = \Theta_q(1)$. Denote $\tilde{\alpha} = q + 1 - \alpha$ and $\mathcal{E}_{\mathsf{s},\geq\tilde{\alpha}} = \{S \in \mathcal{E}_\mathsf{s} : \gamma(S) \geq \tilde{\alpha}\}$ and $\mathcal{E}_{\mathsf{s},<\tilde{\alpha}} = \mathcal{E}_\mathsf{s} \setminus \mathcal{E}_{\mathsf{s},\geq\tilde{\alpha}}$. Using Eq. (64) and that $1 \leq r(S) \leq q$, we have $\{\lambda_{d,j}\}_{j\in[\mathsf{m}]}$ that contains exactly the eigenvalues associated to homogeneous polynomials of degree less or equal to $\mathsf{s}-1$ and of degree $\mathsf{s}$ with $S \in \mathcal{E}_{\mathsf{s},<\tilde{\alpha}}$ (which corresponds to the sets $S$ such that $r(S) > \alpha$, i.e., $\xi_{q,\mathsf{s}} r(S) > q\xi_{q,\mathsf{s}+1}$). In particular, if $\alpha < 1$, then $\{\lambda_{d,j}\}_{j\in[\mathsf{m}]}$ contains exactly the eigenvalues associated to all homogeneous polynomials of degree less or equal to $\mathsf{s}$.

Note that we have

$$\mathsf{m} \leq \sum_{\ell=0}^{\mathsf{s}} |\mathcal{E}_\ell| = O_q(dq^{\mathsf{s}-1}) = O_q(q^{-\delta}n). \tag{65}$$

**Step 2. Diagonal elements of the truncated kernel.**

Define the truncated kernel $H_{d,>\mathsf{m}}$ to be

$$H_{d,>\mathsf{m}}(\boldsymbol{x}, \boldsymbol{y}) = \sum_{j\geq \mathsf{m}+1} \lambda_{q,j} \psi_{q,j}(\boldsymbol{x}) \psi_{d,j}(\boldsymbol{y})$$

$$= \frac{\xi_{q,\mathsf{s}}}{d} \sum_{S \in \mathcal{E}_{\mathsf{s},\geq\tilde{\alpha}}} r(S) \cdot Y_S(\boldsymbol{x}) Y_S(\boldsymbol{y}) + \frac{1}{d} \sum_{\ell=\mathsf{s}+1}^{q} \xi_{q,\ell} \sum_{S \in \mathcal{E}_\ell} r(S) \cdot Y_S(\boldsymbol{x}) Y_S(\boldsymbol{y}).$$

The diagonal elements of the truncated kernel are given by: for any $\boldsymbol{x} \in \mathscr{Q}^d$,

$$H_{d,>\mathsf{m}}(\boldsymbol{x}, \boldsymbol{x}) = \frac{\xi_{q,\mathsf{s}}}{d} \sum_{S \in \mathcal{E}_{\mathsf{s},\geq\tilde{\alpha}}} r(S) + \frac{1}{d} \sum_{\ell=\mathsf{s}+1}^{q} \xi_{q,\ell} \sum_{S \in \mathcal{E}_\ell} r(S) = \mathrm{Tr}(\mathbb{H}_{d,>\mathsf{m}}). \tag{66}$$

Notice that

$$\sum_{S \in \mathcal{E}_\ell} r(S) = \sum_{h=\ell}^{q}(q+1-h)|\mathcal{E}_{\ell,h}| = d \sum_{h=\ell}^{q}(q+1-h)\binom{h-2}{\ell-2} = d\binom{q}{\ell} = dB(\mathscr{Q}^q; \ell),$$

$$\sum_{S \in \mathcal{E}_{\mathsf{s},\geq\tilde{\alpha}}} r(S) \leq \alpha \sum_{h=q+1-\alpha}^{q} |\mathcal{E}_{\mathsf{s},h}| \leq d\alpha^2 \binom{q-2}{\mathsf{s}-2} = O_d(dq^{\mathsf{s}-2}).$$

Hence using that $\xi_{q,\mathsf{s}} = O_d(q^{-\mathsf{s}})$, we have

$$\mathrm{Tr}(\mathbb{H}_{d,>\mathsf{m}}) = \frac{\xi_{q,\mathsf{s}}}{d} \sum_{S \in \mathcal{E}_{\mathsf{s},\geq\tilde{\alpha}}} r(S) + \sum_{\ell=\mathsf{s}+1}^{q} \xi_{q,\ell} B(\mathscr{Q}^q; \ell) = h_{q,>\mathsf{s}}(1) + o_{q,\mathbb{P}}(1),$$

where $h_{q,>\mathsf{s}}$ is the inner-product kernel with the $(\mathsf{s}+1)$-first Gegenbauer coefficients set to zero, i.e., $h_{q,>\mathsf{s}}(\langle \boldsymbol{u}, \boldsymbol{v} \rangle / q) = \sum_{\ell=\mathsf{s}+1}^{q} \xi_{q,\ell} B(\mathscr{Q}^q; \ell) Q_\ell^{(q)}(\langle \boldsymbol{u}, \boldsymbol{v} \rangle)$, for any $\boldsymbol{u}, \boldsymbol{v} \in \mathscr{Q}^q$. From Assumption 1 at level $\mathsf{s}$, we have $\Omega_q(1) = \xi_{q,\ell'} B(\mathscr{Q}^q; \ell') \leq h_{q,>\mathsf{s}}(1) \leq h_q(1) = O_q(1)$. Hence, $\mathrm{Tr}(\mathbb{H}_{d,>\mathsf{m}}) = \Theta_d(1)$.

Similarly,

$$\mathbb{E}_{\boldsymbol{x}'}[H_{d,>\mathsf{m}}(\boldsymbol{x}, \boldsymbol{x}')^2] = \frac{\xi_{q,\mathsf{s}}^2}{d} \sum_{S \in \mathcal{E}_{\mathsf{s},\geq\tilde{\alpha}}} r(S)^2 + \frac{1}{d} \sum_{\ell=\mathsf{s}+1}^{q} \xi_{q,\ell}^2 \sum_{S \in \mathcal{E}_\ell} r(S)^2 = \mathrm{Tr}(\mathbb{H}_{d,>\mathsf{m}}^2). \tag{67}$$

**Step 3. Choosing the sequence $u = u(d)$.**

Let $\mathsf{s}'$ be chosen as in Assumption 1, i.e., such that $\xi_{q,\mathsf{s}'} B(\mathscr{Q}^q; \mathsf{s}') = \Omega_q(1)$. We have

$$\xi_{q,\mathsf{s}'} = \Theta_q(q^{-\mathsf{s}'}), \qquad \sup_{\ell \geq \mathsf{s}'+1} \xi_{q,\ell} = O_q(q^{-\mathsf{s}'-1}). \tag{68}$$

Set $u = u(d)$ to be the number of eigenvalues such that $\lambda_{q,j} > q\xi_{q,\mathsf{s}'}/d = \Theta_q(q^{-\mathsf{s}'+1}/d)$. From Eqs. (64) and (68), and recalling that $1 \leq r(S) \leq q$, we deduce that $\{\lambda_{d,j}\}_{j \in [u]}$ must contain all the eigenvalues associated to homogeneous polynomials of degree less or equal to $\ell$ and does not contain any of the eigenvalues associated to homogeneous polynomials of degree larger or equal to $\mathsf{s}'$.

We have

$$\mathrm{Tr}(\mathbb{H}_{d,>u}) = \sum_{j>u} \lambda_{q,j} \leq \mathrm{Tr}(\mathbb{H}_{d,>m}) = O_q(1),$$

$$\mathrm{Tr}(\mathbb{H}_{d,>u}) \geq \frac{\xi_{q,\mathsf{s}'}}{d} \sum_{S \in \mathcal{E}_{\mathsf{s}'}} r(S) = \xi_{q,\mathsf{s}'} B(\mathscr{Q}^q; \mathsf{s}') = \Omega_q(1).$$

Similarly, we have

$$\mathrm{Tr}(\mathbb{H}_{d,>u}^2) = \sum_{j>u} \lambda_{q,j}^2 \leq \mathrm{Tr}(\mathbb{H}_{d,>u}) \cdot \sup_{j>m} \lambda_{d,j} = qd^{-1}\xi_{q,\mathsf{s}'} \mathrm{Tr}(\mathbb{H}_{d,>m}) = O_q(d^{-1}q^{-\mathsf{s}'+1}),$$

$$\mathrm{Tr}(\mathbb{H}_{d,>u}^2) \geq \frac{\xi_{q,\mathsf{s}'}^2}{d^2} \sum_{S \in \mathcal{E}_{\mathsf{s}'}} r(S)^2 \geq d^{-1}\xi_{q,\mathsf{s}'}^2 B(\mathscr{Q}^q; \mathsf{s}') = \Omega_q(d^{-1}q^{-\mathsf{s}'}).$$

Finally,

$$\mathrm{Tr}(\mathbb{H}_{d,>u}^4) = \sum_{j>u} \lambda_{d,j}^4 \leq d^{-3}q^3\xi_{q,\mathsf{s}'}^3 \mathrm{Tr}(\mathbb{H}_{d,>m}) = O_q(d^{-3}q^{-3\ell'+3}).$$

**Step 4. Checking the kernel concentration property at level $\{(n(q), \mathsf{m}(q))\}_{q \geq 1}$.**

Let us check the kernel concentration property at level $(n, \mathsf{m})$ with the sequence of integers $\{u(q)\}_{q \geq 1}$ defined in the previous step (Assumption 4 in [38]):

(a) *(Hypercontractivity of finite eigenspaces)* The subspace spanned by the top eigenvectors $\{\psi_{q,j}\}_{j\in[u]}$ is contained in the subspace of polynomials of degree less or equal to $\mathsf{s}'-1$ on the hypercube. The hypercontractivity of this subspace is a consequence of a classical result due to Beckner, Bonami and Gross (see Lemma 4 in Section D).

(b) *(Properly decaying eigenvalues.)* From step 3 and recalling that $\mathsf{s}' \geq 1/\delta + 2\mathsf{s} + 3$ where $\delta > 0$ verifies $q \geq d^{\delta}$, we have

$$\frac{\mathrm{Tr}(\mathbb{H}_{d,>u})^2}{\mathrm{Tr}(\mathbb{H}_{d,>u}^2)} = \Omega_q(1) \cdot dq^{\mathsf{s}'-1} = \Omega_q(1) \cdot d^2 q^{2\mathsf{s}+1} \geq n^{2+\delta'},$$

for $\delta' > 0$ sufficiently small. Similarly,

$$\frac{\mathrm{Tr}(\mathbb{H}_{d,>u}^2)^2}{\mathrm{Tr}(\mathbb{H}_{d,>u}^4)} = \Omega_q(1) \cdot dq^{\mathsf{s}'-3} = \Omega_q(1) \cdot d^2 q^{2\mathsf{s}} \geq n^{2+\delta'},$$

for $\delta' > 0$ chosen sufficiently small.

(c) *(Concentration of the diagonal elements of the kernel)* From Eqs. (66) and (67), the diagonal elements of the kernel are constant and the assumption is automatically verified.

**Step 5. Checking the eigenvalue condition at level $\{(n(q), \mathsf{m}(q))\}_{q\geq 1}$.**

Let us now check the eigenvalue condition at level $\{(n(q), \mathsf{m}(q))\}_{q\geq 1}$ which corresponds to Assumption 5 in [38]):

(a) First notice that

$$\sum_{S\in\mathcal{E}_{\mathsf{s}+1}} r(S)^2 = d \sum_{h=\mathsf{s}+1}^{q} (q+1-h)^2 \binom{h-1}{\mathsf{s}-1} \geq d \sum_{h=\mathsf{s}+1}^{\lfloor q/2 \rfloor} (q+1-h)^2 \binom{h-1}{\mathsf{s}-1}$$

$$\geq \frac{dq^2}{4} \sum_{h=\mathsf{s}+1}^{\lfloor q/2 \rfloor} \binom{h-1}{\mathsf{s}-1} = \frac{dq^2}{4} \binom{\lfloor q/2 \rfloor}{\mathsf{s}} = \Omega_q(1) \cdot dq^{2+\mathsf{s}}. \tag{69}$$

Hence

$$\frac{\mathrm{Tr}(\mathbb{H}_{d,>\mathsf{m}}^2)}{\lambda_{d,\mathsf{m}+1}^2} \geq \frac{\sum_{S\in\mathcal{E}_{\mathsf{s}+1}} \xi_{d,\mathsf{s}+1}^2 r(S)^2}{q^2 \xi_{d,\mathsf{s}+1}^2} = \Omega_q(1) \cdot dq^{\mathsf{s}} \geq n^{1+\delta},$$

for $\delta > 0$ sufficiently small. Similarly,

$$\frac{\mathrm{Tr}(\mathbb{H}_{d,>\mathsf{m}})}{\lambda_{d,\mathsf{m}+1}} = \Omega_q(1) \cdot \frac{d}{q \xi_{d,\mathsf{s}+1}} = \Omega_d(1) \cdot dq^{\mathsf{s}} \geq n^{1+\delta}.$$

(b) This is a direct consequence of Eq. (65).

We can therefore apply Theorem 4 in [38], which concludes the proof. $\qquad\square$

## C.5 Proof of Theorem 8

Consider $q^{\mathsf{s}-1+\delta} \leq n \leq q^{\mathsf{s}-\delta}$ for some $\delta > 0$ and a sequence of inner-product kernels $\{h_q\}_{q\geq 1}$ that satisfies Assumptions 1 and 2 at level $\mathsf{s}$. We consider the one-layer convolutional kernel with global average pooling

$$H_{\mathsf{GP}}^{\mathsf{CK},d}(\boldsymbol{x}, \boldsymbol{y}) = \frac{1}{d} \sum_{k,k'=1}^{d} h_q\big(\langle \boldsymbol{x}_{(k)}, \boldsymbol{y}_{(k')}\rangle / q\big).$$

Again, the proof of Theorem 8 will amount to checking that the conditions of Theorem 4 in [38] hold.

For the sake of simplicity, we will further assume that $\xi_{q,\mathsf{s}} > q\xi_{q,\mathsf{s}+1}$, which simplifies some of the computation. This condition can be removed as in Theorem 7, by considering the set $\mathcal{C}_{\mathsf{s},<\tilde\alpha} = \{S \in \mathcal{C}_{\mathsf{s}} : \gamma(S) < \tilde\alpha\}$ and showing that the extra terms corresponding to these eigenfunctions are negligible.

*Proof of Theorem 8.* **Step 1. Diagonalization of the kernel and choosing** $\mathsf{m} = \mathsf{m}(q)$.

From Proposition 2 with $\omega = d$, we have the following diagonalization of $H_{d,q}^d$:

$$H_d(\boldsymbol{x}, \boldsymbol{y}) := H_{\mathsf{GP}}^{\mathsf{CK},d}(\boldsymbol{x}, \boldsymbol{y}) = \sum_{\ell=0}^q \sum_{S \in \mathcal{C}_\ell} \xi_{q,\ell} r(S) \cdot \psi_S(\boldsymbol{x}) \psi_S(\boldsymbol{y}),$$

where we recall $\psi_S(\boldsymbol{x}) = \frac{1}{\sqrt{d}} \sum_{k \in [d]} Y_{k+S}(\boldsymbol{x})$ and that $\mathcal{C}_\ell$ is the quotient space of $\mathcal{E}_\ell$ with the translation equivalence relation. It is easy to check that $|\mathcal{C}_\ell| = \binom{q-1}{\ell-1}$.

From Assumption 1, we get the same bounds on the Gegenbauer coefficients $\xi_{q,\ell}$ as Eq. (64) in the proof of Theorem 7. Denote $\{\lambda_{q,j}\}_{j \geq 1}$ the eigenvalues $\{\xi_{q,\ell} r(S)\}_{\ell=0,\dots,q;S \in \mathcal{E}_\ell}$ in nonincreasing order, and $\{\psi_{q,j}\}_{j \geq 1}$ the reordered eigenfunctions. Set $\mathsf{m}$ to be the number of eigenvalues such that $\lambda_{q,j} > q\xi_{q,\mathsf{s}+1}$ (recall $q\xi_{q,\mathsf{s}+1} = \Theta_d(q^{-\mathsf{s}})$). From the bounds (64) and our simplifying assumption that $\xi_{q,\mathsf{s}} > q\xi_{q,\mathsf{s}+1}$, we have $\{\lambda_{d,j}\}_{j \in [\mathsf{m}]}$ that contains exactly the eigenvalues associated to homogeneous polynomials of degree less or equal to $\mathsf{s}$.

Note that we have

$$\mathsf{m} = \sum_{\ell=0}^{\mathsf{s}} |\mathcal{C}_\ell| = O_q(q^{\mathsf{s}-1}) = O_q(q^{-\delta} n). \tag{70}$$

**Step 2. Diagonal elements of the truncated kernel.**

Define the truncated kernel $H_{d,>\mathsf{m}}$ to be

$$H_{d,>\mathsf{m}}(\boldsymbol{x}, \boldsymbol{y}) = \sum_{j \geq \mathsf{m}+1} \lambda_{d,j} \psi_{d,j}(\boldsymbol{x}) \psi_{d,j}(\boldsymbol{y}) = \sum_{\ell=\mathsf{s}+1}^q \sum_{S \in \mathcal{C}_\ell} \xi_{q,\ell} r(S) \cdot \psi_S(\boldsymbol{x}) \psi_S(\boldsymbol{y}).$$

The diagonal elements of the truncated kernel are given by: for any $\boldsymbol{x} \in \mathcal{Q}^d$,

$$H_{d,>\mathsf{m}}(\boldsymbol{x}, \boldsymbol{x}) = \sum_{\ell=\mathsf{s}+1}^q \xi_{q,\ell} B(\mathcal{Q}^q; \ell) \Upsilon_\ell^{(q)}(\boldsymbol{x}),$$

where

$$\Upsilon_\ell^{(q)}(\boldsymbol{x}) = \frac{1}{B(\mathcal{Q}^q; \ell)} \sum_{S \in \mathcal{C}_\ell} r(S) \psi_S(\boldsymbol{x})^2.$$

Notice that we have now

$$\sum_{S \in \mathcal{C}_\ell} r(S) = \sum_{h=\ell}^q (q+1-h) \binom{h-2}{\ell-2} = \binom{q}{\ell} = B(\mathcal{Q}^q; \ell).$$

Therefore $\mathbb{E}_{\boldsymbol{x}}[\Upsilon_\ell^{(q)}(\boldsymbol{x})] = 1$ and

$$\mathrm{Tr}(\mathbb{H}_{d,>\mathsf{m}}) = \mathbb{E}_{\boldsymbol{x}}[H_{d,>\mathsf{m}}(\boldsymbol{x}, \boldsymbol{x})] = \sum_{\ell=\mathsf{s}+1}^q \xi_{q,\ell} B(\mathcal{Q}^q; \ell) = h_{q,>\mathsf{s}}(1).$$

From Proposition 7 with $\ell = \mathsf{s}$, we have

$$\sup_{i \in [n]} \left| H_{d,>\mathsf{m}}(\boldsymbol{x}_i, \boldsymbol{x}_i) - \mathbb{E}_{\boldsymbol{x}}[H_{d,>\mathsf{m}}(\boldsymbol{x}, \boldsymbol{x})] \right| = \mathrm{Tr}(\mathbb{H}_{d,>\mathsf{m}}) \cdot o_{d,\mathbb{P}}(1),$$
$$\sup_{i \in [n]} \left| \mathbb{E}_{\boldsymbol{x}'}[H_{d,>\mathsf{m}}(\boldsymbol{x}_i, \boldsymbol{x}')^2] - \mathbb{E}_{\boldsymbol{x},\boldsymbol{x}'}[H_{d,>\mathsf{m}}(\boldsymbol{x}, \boldsymbol{x}')^2] \right| = \mathrm{Tr}(\mathbb{H}_{d,>\mathsf{m}}^2) \cdot o_{d,\mathbb{P}}(1). \tag{71}$$

**Step 3. Choosing the sequence** $u = u(d)$.

Let $\mathsf{s}'$ be chosen as in Assumption 1. Similarly to step 3 in the proof of Theorem 7, take $u = u(d)$ to be the number of eigenvalues such that $\lambda_{q,j} > q\xi_{q,\ell'}$. We get

$$\mathrm{Tr}(\mathbb{H}_{d,>u}) = \Theta_q(1),$$
$$\mathrm{Tr}(\mathbb{H}_{d,>u}^2) = O_q(q^{-\mathsf{s}'+1}),$$
$$\mathrm{Tr}(\mathbb{H}_{d,>u}^2) = \Omega_q(q^{-\ell'}),$$
$$\mathrm{Tr}(\mathbb{H}_{d,>u}^4) = O_q(q^{-3\mathsf{s}'+3}).$$

**Step 4. Checking the kernel concentration property at level** $\{(n(q), \mathsf{m}(q))\}_{q \geq 1}$**.**

The kernel concentration property at level $(n, \mathsf{m})$ hold with the sequence $\{u(q)\}_{q \geq 1}$ as defined in step 3. The hypercontractivity of finite eigenspaces and the properly decaying eigenvalues are obtained as in step 4 of the proof of Theorem 7, while the concentration of the diagonal elements of the kernel is given by Eq. (71).

**Step 5. Checking the eigenvalue condition at level** $\{(n(q), \mathsf{m}(q))\}_{q \geq 1}$**.**

This is obtained similarly as in step 5 of the proof of Theorem 7.

$\square$

### C.6 Auxiliary results

**Proposition 7.** *Let* $\mathsf{s} \geq 1$ *be a fixed integer. Assume that the sequence of inner-product kernels* $\{h_q\}_{q \geq 1}$ *satisfies Assumptions 1 and 2 at level* $\mathsf{s}$*. Define* $H_d^{>\mathsf{s}} : \mathcal{Q}^d \times \mathcal{Q}^d \to \mathbb{R}$ *as the convolutional kernel with global average pooling*

$$H_d^{>\mathsf{s}}(\boldsymbol{x}, \boldsymbol{y}) = \frac{1}{d} \sum_{k,k' \in [d]} h_{q,>\mathsf{s}}(\langle \boldsymbol{x}_{(k)}, \boldsymbol{y}_{(k')} \rangle / q),$$

*where* $h_{q,>\mathsf{s}}$ *is the inner-product kernel where the* $\mathsf{s} + 1$ *first Gegenbauer coefficients are set to* $0$*.*

*Then for* $n = O_q(q^p)$ *for some fixed* $p$*, letting* $(\boldsymbol{x}_i)_{i \in [n]} \sim \mathrm{Unif}(\mathcal{Q}^d)$*, we have*

$$\sup_{i \in [n]} \left| H_d^{>\mathsf{s}}(\boldsymbol{x}_i, \boldsymbol{x}_i) - \mathbb{E}_{\boldsymbol{x}}[H_d^{>\mathsf{s}}(\boldsymbol{x}, \boldsymbol{x})] \right| = \mathbb{E}_{\boldsymbol{x}}[H_d^{>\mathsf{s}}(\boldsymbol{x}, \boldsymbol{x})] \cdot o_{d,\mathbb{P}}(1), \tag{72}$$

$$\sup_{i \in [n]} \left| \mathbb{E}_{\boldsymbol{x}'}[H_d^{>\mathsf{s}}(\boldsymbol{x}_i, \boldsymbol{x}')^2] - \mathbb{E}_{\boldsymbol{x},\boldsymbol{x}'}[H_d^{>\mathsf{s}}(\boldsymbol{x}, \boldsymbol{x}')^2] \right| = \mathbb{E}_{\boldsymbol{x},\boldsymbol{x}'}[H_d^{>\mathsf{s}}(\boldsymbol{x}, \boldsymbol{x}')^2] \cdot o_{d,\mathbb{P}}(1). \tag{73}$$

*Proof of Proposition 7.* **Step 1. Bounding** $\sup_{i \in [n]} \left| H_d^{>\mathsf{s}}(\boldsymbol{x}_i, \boldsymbol{x}_i) - \mathbb{E}_{\boldsymbol{x}}[H_d^{>\mathsf{s}}(\boldsymbol{x}, \boldsymbol{x})] \right|$**.**

Recall that we defined

$$\Upsilon_\ell^{(q)}(\boldsymbol{x}) = \frac{1}{B(\mathcal{Q}^q; \ell)} \sum_{S \in \mathcal{C}_\ell} r(S) \psi_S(\boldsymbol{x})^2.$$

Following the same proof as Proposition 8 in [39], notice that for the integer $v$ in Assumption 2, by Lemma 2 stated below, we have

$$\sup_{i \in [n]} \left| H_d^{>\mathsf{s}}(\boldsymbol{x}_i, \boldsymbol{x}_i) - \mathbb{E}_{\boldsymbol{x}}[H_d^{>\mathsf{s}}(\boldsymbol{x}, \boldsymbol{x})] \right|$$

$$\leq \sup_{i \in [n]} \left| H_d^{>v}(\boldsymbol{x}_i, \boldsymbol{x}_i) - \mathbb{E}_{\boldsymbol{x}}[H_d^{>v}(\boldsymbol{x}, \boldsymbol{x})] \right| + \sum_{\ell=\mathsf{s}+1}^{v} \xi_{q,\ell} B(\mathcal{Q}^q; \ell) \cdot \max_{i \in [n]} \left| \Upsilon_\ell^{(d)}(\boldsymbol{x}_i) - \mathbb{E}_{\boldsymbol{x}}[\Upsilon_\ell^{(d)}(\boldsymbol{x})] \right|$$

$$= \sup_{i \in [n]} \left| H_d^{>v}(\boldsymbol{x}_i, \boldsymbol{x}_i) - \mathbb{E}_{\boldsymbol{x}}[H_d^{>v}(\boldsymbol{x}, \boldsymbol{x})] \right| + \left( \sum_{\ell=\mathsf{s}+1}^{v} \xi_{q,\ell} B(\mathcal{Q}^q; \ell) \right) \cdot o_{d,\mathbb{P}}(1).$$

By Assumption 2, there exists $C > 0$ such that for any $\gamma \in [-1, 1]$,

$$\left| h_{q,>v}(\gamma) - \sum_{r=0}^{v} \frac{1}{r!} h_{q,>v}^{(r)}(0) \gamma^r \right| \leq C \cdot |\gamma|^{v+1}, \tag{74}$$

and $|h_{q,>v}^{(r)}(0)| \leq Cq^{-(v+1-r)/2}$ for $r \leq v$. Moreover, by Hanson-Wright inequality as in Lemma 3, using $n = O_q(q^p)$ (at most polynomial in $q$) and a union bound, we have for any $\eta > 0$,

$$\sup_{1 \leq r \leq v+1} \sup_{k \neq l} \sup_{i \in [n]} \left| \langle (\boldsymbol{x}_i)_{(k)}, (\boldsymbol{x}_i)_{(l)} \rangle^r \right| \cdot q^{-k/2-\eta} = o_{q,\mathbb{P}}(1),$$

$$\sup_{1 \leq r \leq v+1} \sup_{k \neq l} \mathbb{E}\left[ \left| \langle \boldsymbol{x}_{(k)}, \boldsymbol{x}_{(l)} \rangle^r \right| \right] \cdot q^{-k/2-\eta} = o_{q,\mathbb{P}}(1).$$

Therefore, injecting these bounds in Eq. (74), we get

$$\sup_{k \neq l} \sup_{i \in [n]} \left| h_{q,>v}(\langle (\boldsymbol{x}_i)_{(k)}, (\boldsymbol{x}_i)_{(l)} \rangle / q) \right| = O_{q,\mathbb{P}}(q^{-(v+1)/2+\eta}),$$

$$\sup_{k \neq l} \mathbb{E}\left[ \left| h_{q,>v}(\langle \boldsymbol{x}_{(k)}, \boldsymbol{x}_{(l)} \rangle / q) \right| \right] = O_{q,\mathbb{P}}(q^{-(v+1)/2+\eta}).$$

Hence, we deduce that

$$\sup_{i \in [n]} \left| H_d^{>v}(\boldsymbol{x}_i, \boldsymbol{x}_i) - \mathbb{E}_{\boldsymbol{x}}[H_d^{>v}(\boldsymbol{x}, \boldsymbol{x})] \right|$$

$$\leq \frac{1}{d} \sum_{k \neq l \in [d]} \sup_{i \in [n]} \left| h_{q,>v}(\langle (\boldsymbol{x}_i)_{(k)}, (\boldsymbol{x}_i)_{(l)} \rangle / q) - \mathbb{E}_{\boldsymbol{x}}[h_{q,>v}(\langle \boldsymbol{x}_{(k)}, \boldsymbol{x}_{(l)} \rangle / q)] \right|$$

$$\leq d \sup_{k \neq l} \left\{ \sup_{i \in [n]} \left| h_{q,>v}(\langle (\boldsymbol{x}_i)_{(k)}, (\boldsymbol{x}_i)_{(l)} \rangle / q) \right| + \mathbb{E}\left[ \left| h_{q,>v}(\langle \boldsymbol{x}_{(k)}, \boldsymbol{x}_{(l)} \rangle / q) \right| \right] \right\}$$

$$= O_{q,\mathbb{P}}(dq^{-(v+1)/2+\eta}) = o_{d,\mathbb{P}}(1).$$

Furthermore, recall that by Assumption 1, we have $\mathbb{E}[\mathbb{H}_d^{>\ell}(\boldsymbol{x}, \boldsymbol{x})] \geq \xi_{q,\mathsf{s}'} B(\mathscr{Q}^q; \mathsf{s}') = \Omega_q(1)$. We get

$$\sup_{i \in [n]} \left| H_d^{>v}(\boldsymbol{x}_i, \boldsymbol{x}_i) - \mathbb{E}_{\boldsymbol{x}}[H_d^{>v}(\boldsymbol{x}, \boldsymbol{x})] \right| = \mathbb{E}[\mathbb{H}_d^{>\ell}(\boldsymbol{x}, \boldsymbol{x})] \cdot o_{q,\mathbb{P}}(1),$$

which concludes the proof of the first bound.

**Step 2. Bounding** $\sup_{i \in [n]} \left| \mathbb{E}_{\boldsymbol{x}'}[H_d^{>\mathsf{s}}(\boldsymbol{x}_i, \boldsymbol{x}')^2] - \mathbb{E}_{\boldsymbol{x}, \boldsymbol{x}'}[H_d^{>\mathsf{s}}(\boldsymbol{x}, \boldsymbol{x}')^2] \right|$.

Notice that we can write,

$$\mathbb{E}_{\boldsymbol{x}'}[H_d^{>\mathsf{s}}(\boldsymbol{x}, \boldsymbol{x}')^2] = \sum_{\ell=\mathsf{s}+1}^{q} \xi_{q,\ell}^2 R_\ell \cdot \Xi_\ell^{(d)}(\boldsymbol{x}),$$

where we denoted $R_\ell = \sum_{S \in \mathcal{C}_\ell} r(S)^2$ and

$$\Xi_\ell^{(d)}(\boldsymbol{x}) = \frac{1}{R_\ell} \sum_{S \in \mathcal{C}_\ell} r(S)^2 \psi_S(\boldsymbol{x})^2.$$

Then, by Lemma 2, we get for any $u \geq \mathsf{s}$,

$$\sup_{i \in [n]} \left| \mathbb{E}_{\boldsymbol{x}'}[H_d^{>\mathsf{s}}(\boldsymbol{x}_i, \boldsymbol{x}')^2] - \mathbb{E}_{\boldsymbol{x}, \boldsymbol{x}'}[H_d^{>\mathsf{s}}(\boldsymbol{x}, \boldsymbol{x}')^2] \right|$$

$$\leq \sup_{i \in [n]} \left| \mathbb{E}_{\boldsymbol{x}'}[H_d^{>u}(\boldsymbol{x}_i, \boldsymbol{x}')^2] - \mathbb{E}_{\boldsymbol{x}, \boldsymbol{x}'}[H_d^{>u}(\boldsymbol{x}, \boldsymbol{x}')^2] \right| + \sum_{\ell=\mathsf{s}+1}^{u} \xi_{q,\ell}^2 R_\ell \cdot \max_{i \in [n]} \left| \Xi_\ell^{(d)}(\boldsymbol{x}_i) - \mathbb{E}_{\boldsymbol{x}}[\Xi_\ell^{(d)}(\boldsymbol{x})] \right|$$

$$= \sup_{i \in [n]} \left| \mathbb{E}_{\boldsymbol{x}'}[H_d^{>u}(\boldsymbol{x}_i, \boldsymbol{x}')^2] - \mathbb{E}_{\boldsymbol{x}, \boldsymbol{x}'}[H_d^{>u}(\boldsymbol{x}, \boldsymbol{x}')^2] \right| + \left( \sum_{\ell=\mathsf{s}+1}^{u} \xi_{q,\ell}^2 R_\ell \right) \cdot o_{d,\mathbb{P}}(1).$$

We conclude following the same argument as in the proof of Proposition 9 in [39]. $\qquad\square$

**Lemma 2.** *Let $\ell \geq 2$ be an integer. Define $\Upsilon_\ell^{(d)} : \mathscr{Q}^d \to \mathbb{R}$ and $\Xi_\ell^{(d)} : \mathscr{Q}^d \to \mathbb{R}$ to be*

$$\Upsilon_\ell^{(d)}(\boldsymbol{x}) = \frac{1}{B(\mathscr{Q}^q; \ell)} \sum_{S \in \mathcal{C}_\ell} r(S) \psi_S(\boldsymbol{x})^2, \tag{75}$$

$$\Xi_\ell^{(d)}(\boldsymbol{x}) = \frac{1}{R_\ell} \sum_{S \in \mathcal{C}_\ell} r(S)^2 \psi_S(\boldsymbol{x})^2, \tag{76}$$

*where $R_\ell = \sum_{S \in \mathcal{C}_\ell} r(S)^2$.*

*Let $n \leq q^p$ for some fixed $p$. Then, for $(\boldsymbol{x}_i)_{i \in [n]} \overset{i.i.d.}{\sim} \mathrm{Unif}(\mathscr{Q}^d)$, we have*

$$\max_{i \in [n]} \left| \Upsilon_\ell^{(d)}(\boldsymbol{x}_i) - \mathbb{E}_{\boldsymbol{x}}[\Upsilon_\ell^{(d)}(\boldsymbol{x})] \right| = o_{d,\mathbb{P}}(1), \tag{77}$$

$$\max_{i \in [n]} \left| \Xi_\ell^{(d)}(\boldsymbol{x}_i) - \mathbb{E}_{\boldsymbol{x}}[\Xi_\ell^{(d)}(\boldsymbol{x})] \right| = o_{d,\mathbb{P}}(1), \tag{78}$$

*where $\mathbb{E}_{\boldsymbol{\theta}}[\Upsilon_\ell^{(d)}(\boldsymbol{\theta})] = \mathbb{E}_{\boldsymbol{x}}[\Xi_\ell^{(d)}(\boldsymbol{x})] = 1$.*

*Proof of Lemma 2.* **Step 1. Bounding $\max_{i \in [n]} \left| \Upsilon_\ell^{(d)}(\boldsymbol{x}_i) - \mathbb{E}_{\boldsymbol{x}}[\Upsilon_\ell^{(d)}(\boldsymbol{x})] \right|$.**

Define $F_\ell : \mathscr{Q}^d \to \mathbb{R}$ to be

$$F_\ell(\boldsymbol{x}) = \Upsilon_\ell^{(d)}(\boldsymbol{x}) - \mathbb{E}_{\boldsymbol{x}}[\Upsilon_\ell^{(d)}(\boldsymbol{x})] = \frac{1}{dB(\mathscr{Q}^q; \ell)} \sum_{S \in \mathcal{C}_\ell} r(S) \sum_{i \neq j \in [d]} Y_{i+S}(\boldsymbol{x}) Y_{j+S}(\boldsymbol{x}). \tag{79}$$

Notice that $F_\ell(\boldsymbol{x})$ is a degree $2\ell$ polynomial and therefore satisfies the hypercontractivity property. For any $m \geq 1$, there exists $C > 0$ such that

$$\mathbb{E}_{\boldsymbol{x}}[F_\ell(\boldsymbol{x})^{2m}]^{1/(2m)} \leq C \cdot \mathbb{E}_{\boldsymbol{x}}[F_\ell(\boldsymbol{x})^2]^{1/2}. \tag{80}$$

Let us bound the right hand side. We have

$$\mathbb{E}[F_\ell(\boldsymbol{x})^2] = \frac{1}{d^2 B(\mathscr{Q}^q; \ell)^2} \sum_{S,S' \in \mathcal{C}_\ell} r(S) r(S') \sum_{i,j,i',j' \in [d]} \omega(B_1, B_2, B_3, B_4),$$

where $B_1 = i + S$, $B_2 = j + S$, $B_3 = i' + S'$ and $B_4 = j' + S'$, and we denoted

$$\omega(B_1, B_2, B_3, B_4) = \mathbb{E}_{\boldsymbol{x}}\left[ Y_{B_1}(\boldsymbol{x}) Y_{B_2}(\boldsymbol{x}) Y_{B_3}(\boldsymbol{x}) Y_{B_4}(\boldsymbol{x}) \right] \mathbb{1}_{B_1 \neq B_2} \mathbb{1}_{B_3 \neq B_4}.$$

Notice that $\omega(B_1, B_2, B_3, B_4) = 1$ if $B_1 \Delta B_2 = B_3 \Delta B_4$ (the symmetric difference) and $0$ otherwise. In other words, every elements in $B_1 \cup B_2 \cup B_3 \cup B_4$ appears exactly in 2 or 4 of these sets.

Let us fix $i \in [d]$ and $S \in \mathcal{C}_\ell$, and bound

$$\sum_{S' \in \mathcal{C}_{q,\ell}} r(S') \sum_{j,i' \neq j' \in [d]} \omega(B_1, B_2, B_3, B_4). \tag{81}$$

Denote $|B_1 \Delta B_2| = 2k$ with $1 \leq k \leq \ell$. In order for $\omega(B_1, B_2, B_3, B_4) = 1$, $B_3$ must contain exactly $k$ points in $B_1 \Delta B_2$ while $B_4$ must contain the remaining $k$ points.

- **Case $k < \ell$.** There are at most $\ell^2$ ways of choosing $j$ such that $B_1 \cap B_2 \neq \emptyset$. Fixing $j$ (i.e., $B_1$ and $B_2$) and $S'$, then there are $2k\ell$ ways of choosing $i'$ and $2k\ell$ ways of choosing $j'$ such that $B_3 \cap (B_1 \Delta B_2) \neq \emptyset$ and $B_4 \cap (B_1 \Delta B_2) \neq \emptyset$. Hence the contribution of these terms in Eq. (81) is upper bounded by

$$\sum_{S' \in \mathcal{C}_\ell} r(S') \sum_{k=1}^{\ell-1} \ell^2 \cdot (2k\ell)^2 \leq 4\ell^7 \sum_{S' \in \mathcal{C}_\ell} r(S') = 4\ell^7 B(\mathscr{Q}^q; \ell). \tag{82}$$

- **Case $k = \ell$.** There are at most $d$ ways of choosing $j$. Furthermore, for $j$ fixed, there are at most $\binom{2\ell}{\ell}$ ways of choosing $B_3$ and $B_4$ such that $B_3 \cup B_4 = B_1 \cup B_2$ (note that $B_1 \cap B_2 = \emptyset$ and therefore $B_3 \cap B_4 = \emptyset$). Hence the contribution of these terms in Eq. (81) is upper bounded by

$$\sum_{S' \in \mathcal{C}_\ell, i', j' \in [d]} r(S') \cdot d \cdot \mathbb{1}_{B_3 \cup B_4 = B_1 \cup B_2} \leq dq \binom{2\ell}{\ell}, \tag{83}$$

where we used that $r(S') \leq q$.

Combining Eqs. (82) and (83) and using there are $dB(\mathscr{Q}^q; \ell)$ choices for $i$ and $S_1$, we get

$$\mathbb{E}[F_\ell(\boldsymbol{x})^2] \leq \frac{1}{d^2 B(\mathscr{Q}^q; \ell)^2} \sum_{i \in [d], S \in \mathcal{C}_\ell} r(S) \Big[ 4\ell^7 B(\mathscr{Q}^q; \ell) + dq \binom{2\ell}{\ell} \Big]$$

$$= O_q(1) \cdot [d^{-1} + q B(\mathscr{Q}^q; \ell)^{-1}] = O_q(q^{-1}),$$

where we used that $\ell \geq 2$ and $B(\mathscr{Q}^q; \ell) = \Omega_q(q^\ell)$.

Using Eq. (80), we deduce

$$\mathbb{E}\Big[ \max_{i \in [n]} |F_\ell(\boldsymbol{x}_i)| \Big] \leq \mathbb{E}\Big[ \max_{i \in [n]} F_\ell(\boldsymbol{x}_i)^{2m} \Big]^{1/(2m)} \leq n^{1/(2m)} \mathbb{E}\Big[ F_\ell(\boldsymbol{x}_i)^{2m} \Big]^{1/(2m)}$$

$$\leq C n^{1/(2m)} \mathbb{E}[F_\ell(\boldsymbol{x})^2]^{1/2} = n^{1/m} \cdot O_q(q^{-1/2}).$$

Using Markov's inequality and taking $m$ sufficiently small yield Eq. (77).

**Step 2. Bounding** $\max_{i \in [n]} \Big| \Xi_\ell^{(d)}(\boldsymbol{x}_i) - \mathbb{E}_{\boldsymbol{x}}[\Xi_\ell^{(d)}(\boldsymbol{x})] \Big|$.

The second bound (78) is obtained very similarly. Define $G_\ell : \mathscr{Q}^d \to \mathbb{R}$ to be

$$G_\ell(\boldsymbol{x}) = \Xi_\ell^{(d)}(\boldsymbol{x}) - \mathbb{E}_{\boldsymbol{x}}[\Xi_\ell^{(d)}(\boldsymbol{x})] = \frac{1}{d R_\ell} \sum_{S \in \mathcal{C}_\ell} r(S)^2 \sum_{i \neq j \in [d]} Y_{i+S}(\boldsymbol{x}) Y_{j+S}(\boldsymbol{x}). \tag{84}$$

Then, we have

$$\mathbb{E}[G_\ell(\boldsymbol{x})^2] = \frac{1}{d^2 R_\ell^2} \sum_{S, S' \in \mathcal{C}_\ell} r(S)^2 r(S')^2 \sum_{i, i', j, j' \in [d]} \omega(B_1, B_2, B_3, B_4).$$

Further notice that following the same computation as in Eq. (69), we get

$$R_\ell = \sum_{S \in \mathcal{C}_\ell} r(S)^2 = \sum_{h=\ell}^{q} (q + 1 - h)^2 \binom{h-2}{\ell-2} = \Omega_q(1) \cdot q^{1+\ell}.$$

Hence, the same computation as for $F_\ell$ in step 1 yields

$$\mathbb{E}[G_\ell(\boldsymbol{x})^2] \leq \frac{1}{d^2 R_\ell^2} \sum_{i \in [d], S \in \mathcal{C}_\ell} r(S)^2 \Big[ 4\ell^7 R_\ell + dq^2 \binom{2\ell}{\ell} \Big]$$

$$= O_q(1) \cdot [d^{-1} + q^2 R_\ell^{-1}] = O_q(q^{-1}),$$

where we used that $\ell \geq 2$. We deduce Eq. (78) similarly to step 1. $\qquad \square$

**Lemma 3** (Hanson-Wright inequality)**.** *There exists a universal constant $c > 0$, such that for any $t > 0$ and $q^{1/\delta} \geq d \geq q \in \mathbb{N}$ for some $\delta > 0$, when $\boldsymbol{x} \in \mathrm{Unif}(\mathscr{Q}^d)$, we have*

$$\mathbb{P}\left( \sup_{k \neq l \in [d]} |\langle \boldsymbol{x}_{(k)}, \boldsymbol{x}_{(l)} \rangle| / q > t \right) \leq 2 q^{2/\delta} \exp\{-cq \cdot \min(t^2, t)\},$$

*where we recall that $\boldsymbol{x}_{(k)} = (x_k, \ldots, x_{k+q-1})$.*

*Proof of Lemma 3.* For any $k \neq l$, denote $\boldsymbol{A} = (a_{ij})_{i,j \in [d]}$ the matrix with $a_{(k+i),(l+i)} = 1$ for $i = 0, \ldots, q - 1$ and $a_{ij} = 0$ otherwise, such that $\langle \boldsymbol{x}, \boldsymbol{A}\boldsymbol{x} \rangle = \langle \boldsymbol{x}_{(k)}, \boldsymbol{x}_{(l)} \rangle$. Note that we have $\|\boldsymbol{A}\|_F = \sqrt{q}$, $\|\boldsymbol{A}\|_{\mathrm{op}} \leq 1$ and $\mathbb{E}[\langle \boldsymbol{x}, \boldsymbol{A}\boldsymbol{x} \rangle] = 0$. By Hanson-Wright inequality of vectors with independent sub-Gaussian entries (for example, see Theorem 1.1 in [42]), we have

$$\mathbb{P}\left( |\langle \boldsymbol{x}, \boldsymbol{A}\boldsymbol{x} \rangle| / q > t \right) \leq 2 \exp\{-cq \cdot \min(t^2, t)\}.$$

Taking the union bound over $k \neq l$ concludes the proof. $\qquad \square$

# D Technical background of function spaces on the hypercube

Fourier analysis on the hypercube is a well studied subject [41]. The purpose of this section is to introduce some notations and objects that are useful in the statement and proofs in the main text.

### D.1 Fourier basis

Denote $\mathscr{Q}^d = \{-1, +1\}^d$ the hypercube in $d$ dimension, and $\tau_d$ to the uniform probability measure on $\mathscr{Q}^d$. All the functions will be assumed to be elements of $L^2(\mathscr{Q}^d, \tau_d)$ (which contains all the bounded functions $f : \mathscr{Q}^d \to \mathbb{R}$), with scalar product and norm denoted as $\langle \cdot, \cdot \rangle_{L^2}$ and $\| \cdot \|_{L^2}$:

$$\langle f, g \rangle_{L^2} \equiv \int_{\mathscr{Q}^d} f(\boldsymbol{x}) g(\boldsymbol{x}) \tau_d(\mathrm{d}\boldsymbol{x}) = \frac{1}{2^n} \sum_{\boldsymbol{x} \in \mathscr{Q}^d} f(\boldsymbol{x}) g(\boldsymbol{x}).$$

Notice that $L^2(\mathscr{Q}^d, \tau_d)$ is a $2^n$ dimensional linear space. By analogy with the spherical case we decompose $L^2(\mathscr{Q}^d, \tau_d)$ as a direct sum of $d+1$ linear spaces obtained from polynomials of degree $\ell = 0, \ldots, d$

$$L^2(\mathscr{Q}^d, \tau_d) = \bigoplus_{\ell=0}^{d} V_{d,\ell}.$$

For each $\ell \in \{0, \ldots, d\}$, consider the Fourier basis $\{Y_{\ell,S}^{(d)}\}_{S \subseteq [d], |S|=\ell}$ of degree $\ell$, where for a set $S \subseteq [d]$, the basis is given by

$$Y_{\ell,S}^{(d)}(\boldsymbol{x}) \equiv x^S \equiv \prod_{i \in S} x_i.$$

It is easy to verify that (notice that $x_i^k = x_i$ if $k$ is odd and $x_i^k = 1$ if $k$ is even)

$$\langle Y_{\ell,S}^{(d)}, Y_{k,S'}^{(d)} \rangle_{L^2} = \mathbb{E}[x^S \times x^{S'}] = \delta_{\ell,k} \delta_{S,S'}.$$

Hence $\{Y_{\ell,S}^{(d)}\}_{S \subseteq [d], |S|=\ell}$ form an orthonormal basis of $V_{d,\ell}$ and

$$\dim(V_{d,\ell}) = B(\mathscr{Q}^d; \ell) = \binom{d}{\ell}.$$

We will omit the superscript $(d)$ in $Y_{\ell,S}^{(d)}$ when clear from the context and write $Y_S := Y_{\ell,S}^{(d)}$.

We denote by $\mathsf{P}_\ell$ the orthogonal projections to $V_{d,\ell}$ in $L^2(\mathscr{Q}^d)$. This can be written in terms of the Fourier basis as

$$\mathsf{P}_\ell f(\boldsymbol{x}) \equiv \sum_{S \subseteq [d], |S|=\ell} \langle f, Y_S \rangle_{L^2} Y_S(\boldsymbol{x}). \tag{85}$$

We also define $\mathsf{P}_{\leq \ell} \equiv \sum_{k=0}^{\ell} \mathsf{P}_k$, $\mathsf{P}_{> \ell} \equiv \mathbf{I} - \mathsf{P}_{\leq \ell} = \sum_{k=\ell+1}^{\infty} \mathsf{P}_k$, and $\mathsf{P}_{< \ell} \equiv \mathsf{P}_{\leq \ell-1}$, $\mathsf{P}_{\geq \ell} \equiv \mathsf{P}_{> \ell-1}$.

### D.2 Hypercubic Gegenbauer

We consider the following family of polynomials $\{Q_\ell^{(d)}\}_{\ell=0,\ldots,d}$ that we will call *hypercubic Gegenbauer*, or G*egenbauer on the $d$-dimensional hypercube*, defined as

$$Q_\ell^{(d)}(\langle \boldsymbol{x}, \boldsymbol{y} \rangle) = \frac{1}{B(\mathscr{Q}^d; \ell)} \sum_{S \subseteq [d], |S|=\ell} Y_{\ell,S}^{(d)}(\boldsymbol{x}) Y_{\ell,S}^{(d)}(\boldsymbol{y}). \tag{86}$$

Notice that the right hand side only depends on $\langle \boldsymbol{x}, \boldsymbol{y} \rangle$ and therefore these polynomials are well defined. In particular,

$$\langle Q_\ell^{(d)}(\langle \mathbf{1}, \cdot \rangle), Q_k^{(d)}(\langle \mathbf{1}, \cdot \rangle) \rangle_{L^2} = \frac{1}{B(\mathscr{Q}^d; k)} \delta_{\ell k}.$$

Hence $\{Q_\ell^{(d)}\}_{\ell=0,\ldots,d}$ form an orthogonal basis of $L^2(\{-d, -d+2, \ldots, d-2, d\}, \tilde{\tau}_d^1)$ where $\tilde{\tau}_d^1$ is the distribution of $\langle \mathbf{1}, \boldsymbol{x} \rangle$ when $\boldsymbol{x} \sim \tau_d$, i.e., $\tilde{\tau}_d^1 \sim 2\mathrm{Bin}(d, 1/2) - d/2$.

It is easy to check more generally that

$$\langle Q_\ell^{(d)}(\langle \boldsymbol{x}, \cdot \rangle), Q_k^{(d)}(\langle \boldsymbol{y}, \cdot \rangle) \rangle_{L^2} = \frac{1}{B(\mathscr{Q}^d; k)} Q_k(\langle \boldsymbol{x}, \boldsymbol{y} \rangle) \delta_{\ell k}.$$

Furthermore, Eq. (86) imply that —up to a constant— $Q_k^{(d)}(\langle \boldsymbol{x}, \boldsymbol{y} \rangle)$ is a representation of the projector onto the subspace of degree-$k$ polynomials

$$(\mathsf{P}_k f)(\boldsymbol{x}) = B(\mathscr{Q}^d; k) \int_{\mathscr{Q}^d} Q_k^{(d)}(\langle \boldsymbol{x}, \boldsymbol{y} \rangle) \, f(\boldsymbol{y}) \, \tau_d(\mathrm{d}\boldsymbol{y}) \,. \tag{87}$$

For a function $\sigma(\cdot/\sqrt{d}) \in L^2(\{-d, -d+2, \ldots, d-2, d\}, \tilde{\tau}_d^1)$, denote its hypercubic Gegenbauer coefficients $\xi_{d,k}(\sigma)$ to be

$$\xi_{d,k}(\sigma) = \int_{\{-d,-d+2,\ldots,d-2,d\}} \sigma(x/\sqrt{d}) Q_k^{(d)}(x) \tilde{\tau}_d^1(\mathrm{d}x) \,. \tag{88}$$

To any inner-product kernel $H_d(\boldsymbol{x}_1, \boldsymbol{x}_2) = h_d(\langle \boldsymbol{x}_1, \boldsymbol{x}_2 \rangle/d)$, with $h_d(\cdot/\sqrt{d}) \in L^2(\{-d, -d+2, \ldots, d-2, d\}, \tilde{\tau}_d^1)$, we can associate a self adjoint operator $\mathscr{H}_d : L^2(\mathscr{Q}^d) \to L^2(\mathscr{Q}^d)$ via

$$\mathscr{H}_d f(\boldsymbol{x}) \equiv \int_{\mathscr{Q}_d} h_d(\langle \boldsymbol{x}, \boldsymbol{x}_1 \rangle/d) \, f(\boldsymbol{x}_1) \, \tau_d(\mathrm{d}\boldsymbol{x}_1) \,. \tag{89}$$

By permutation invariance, the space $V_k$ of homogeneous polynomials of degree $k$ is an eigenspace of $\mathscr{H}_d$, and we will denote the corresponding eigenvalue by $\xi_{d,k}(h_d)$. In other words $\mathscr{H}_d f(\boldsymbol{x}) \equiv \sum_{k=0}^{q} \xi_{d,k}(h_d) \mathsf{P}_k f$. The eigenvalues can be computed via

$$\xi_{d,k}(h_d) = \int_{\{-d,-d+2,\ldots,d-2,d\}} h_d(x/d) Q_k^{(d)}(x) \tilde{\tau}_d^1(\mathrm{d}x) \,. \tag{90}$$

### D.3 Hermite polynomials

The Hermite polynomials $\{\mathrm{He}_k\}_{k \geq 0}$ form an orthogonal basis of $L^2(\mathbb{R}, \gamma)$, where $\gamma(\mathrm{d}x) = e^{-x^2/2}\mathrm{d}x/\sqrt{2\pi}$ is the standard Gaussian measure, and $\mathrm{He}_k$ has degree $k$. We will follow the classical normalization (here and below, expectation is with respect to $G \sim \mathsf{N}(0,1)$):

$$\mathbb{E}\{\mathrm{He}_j(G) \, \mathrm{He}_k(G)\} = k! \, \delta_{jk} \,. \tag{91}$$

As a consequence, for any function $g \in L^2(\mathbb{R}, \gamma)$, we have the decomposition

$$g(x) = \sum_{k=0}^{\infty} \frac{\mu_k(g)}{k!} \, \mathrm{He}_k(x) \,, \qquad \mu_k(g) \equiv \mathbb{E}\{g(G) \, \mathrm{He}_k(G)\} \,. \tag{92}$$

The Hermite polynomials can be obtained as high-dimensional limits of the Gegenbauer polynomials introduced in the previous section. Indeed, the Gegenbauer polynomials (up to a $\sqrt{d}$ scaling in domain) are constructed by Gram-Schmidt orthogonalization of the monomials $\{x^k\}_{k \geq 0}$ with respect to the measure $\tilde{\tau}_d^1$, while Hermite polynomial are obtained by Gram-Schmidt orthogonalization with respect to $\gamma$. Since $\tilde{\tau}_d^1 \Rightarrow \gamma$ (here $\Rightarrow$ denotes weak convergence), it is immediate to show that, for any fixed integer $k$,

$$\lim_{d \to \infty} \mathrm{Coeff}\{Q_k^{(d)}(\sqrt{d}x) \, B(\mathscr{Q}^d; k)^{1/2}\} = \mathrm{Coeff}\left\{\frac{1}{(k!)^{1/2}} \, \mathrm{He}_k(x)\right\} \,. \tag{93}$$

Here and below, for $P$ a polynomial, $\mathrm{Coeff}\{P(x)\}$ is the vector of the coefficients of $P$. As a consequence, for any fixed integer $k$, we have

$$\mu_k(\sigma) = \lim_{d \to \infty} \xi_{d,k}(\sigma)(B(\mathscr{Q}^d; k)k!)^{1/2}, \tag{94}$$

where $\mu_k(\sigma)$ and $\xi_{d,k}(\sigma)$ are given in Eq. (92) and (88).

### D.4 Hypercontractivity of uniform distributions on the hypercube

By Holder's inequality, we have $\|f\|_{L^p} \leq \|f\|_{L^q}$ for any $f$ and any $p \leq q$. The reverse inequality does not hold in general, even up to a constant. However, for some measures, the reverse inequality will hold for some sufficiently nice functions. These measures satisfy the celebrated hypercontractivity properties [4, 5, 8, 25].

**Lemma 4** (Hypercube hypercontractivity [4]). *For any $\ell = \{0, \ldots, d\}$ and $f_d \in L^2(\mathscr{Q}^d)$ to be a degree $\ell$ polynomial, then for any integer $q \geq 2$, we have*

$$\|f_d\|_{L^q(\mathscr{Q}^d)}^2 \leq (q-1)^{\ell} \cdot \|f_d\|_{L^2(\mathscr{Q}^d)}^2 \,.$$