# OpenReview forum: "Learning with convolution and pooling operations in kernel methods"
_NeurIPS.cc/2022/Conference — NeurIPS 2022 Accept_

### Official Review · Reviewer_ykM8 · 2022-07-08

**Rating:** 7
**Confidence:** 4
**Soundness:** 3 good
**Presentation:** 3 good
**Contribution:** 3 good

**Summary:**

This paper analyzes the approximation and generalization performance of CNNs using kernel methods, with a focus on their tradeoff. By studying several convolution, pooling and down sampling operations, theoretical results on the sample complexity to achieve a certain approximation accuracy are presented, based on ideal assumptions on the input data distribution. The paper is well-written and the numerical results support theoretical results in some toy examples.

**Questions:**

Below I have a few suggestions to improve the quality of the paper.

Could you add a picture to introduction for a reader to understand what the CNN model you are considering? It is quite abstract and non-standard to write this model without the notation of the convolution.

Are you analyzing the NTK kernel of CNNs (CNTK)? Maybe this could be made more clear in the abstract or introduction.

In Section 1.2, it is unclear what is the difference between the fixed dimension regime and the high-dimension regime. I think the citation 47, Chap 13 is about high-dimension regime, isn’t it? In the latter case, do you mean that the dimension goes to infinity.

What is the g in assumption A2? Should it be f_* instead of f in Theorem 1, line 229? When you mean q,d large in eq. 6 Theorem 2, would you write o_{q}(1) or o_{q,d}(1)? Shouldn’t the statement of Theorem 2 be probabilistic?


**Ethics Review Area:**

["I don’t know"]

**Strengths And Weaknesses:**

This paper addresses theoretical problem using quite technical tools. The limitation is that it is not clear how to apply these tools to analyze more general CNNs, as indicated in the conclusion of the paper.

---

> ### Author Response · Authors · 2022-08-01
> **Response to Reviewer ykM8**
>
> We thank the reviewer for his comments and suggestions.
>
> The kernel (CK-AP-DS) can be viewed as the CNTK of the one-layer CNN (CNN-AP-DS). However it is not necessary, one can introduce such a kernel directly without mentioning CNTK (as it was done in [35,36,45]). We will make it more clear in the abstract and introduction.
>
> Indeed, Theorem 13.17 [47] and similar results with source/capacity conditions do not refer explicitly to the dimension $d$ (they can be applied to cases where $n,d$ are both large). However these results are proved using an approximation/generalization decomposition where both terms are balanced by tuning the regularization parameter, and they provide upper bounds that are only tight in a minmax sense (up to a multiplicative constant, which might depend on $d$, depending on the theorem). In contrast, the results proved in [22] are exact up to a vanishing additive constant, and hold for any given function and (not too large) regularization parameter (which does not have to be tuned, and can be set to 0).
>
> In assumption (A2), $g$ is a function on the $q$-dimensional hypercube. We assume that $q,d$ are both large and polynomially related and $o_q = o_d = o_{d,q}$. Indeed, the statements in Theorems 2,3 and 5 hold with high probability (the test error minus the prediction converges in probability to zero, and we should use the small-o in probability notation). We will clarify and correct these points in the final version of this paper.

---

### Official Review · Reviewer_6nL3 · 2022-07-13

**Rating:** 5
**Confidence:** 3
**Soundness:** 3 good
**Presentation:** 3 good
**Contribution:** 3 good

**Summary:**

Under the assumption that inputs are from a distribution over the discrete hypercube in d dimensions, the work provides an explanation of why one-layer convolutional kernels outperform fully connected cases for fitting locally translation-invariant functions by showing that the sample complexity required by convolutional kernels is far less than the fully connected case (the degree of efficiency brought by the convolutional kernel is depending on the kernel size). Moreover, the work shows that by adding average pooling, this efficiency could be further improved. The work also shows that the average pooling/downsampling operations facilitate the learning of low-frequency functions.

**Questions:**

1. Since the paper is majorly focused on image classification tasks, is it possible to add some numerical experiments on more realistic data (e.g., CIFAR-10)?

2. (A minor comment on writing) Some abbreviations used in the paper appear before the whole phrase, which makes reading the paper at the beginning a bit hard, for example, the term RKHS is used in the Abstract, Pages 1 and 2, but the full phrase of it first appears in page 3.

**Strengths And Weaknesses:**

Strengths:
The overall presentation of the work is clear. The technical contributions of the work and relationships with prior arts are discussed in detail. The numerical experiment provided in the paper is well-aligned with the theoretical results. Overall, the work provides a nice and intuitive way to understand the benefits of each individual component in modern neural networks.

Weaknesses:
I feel the paper is lacking some practical implications. The data assumption made in the paper is very specific and not realistic. Since the work is dedicated to understanding general components in practical networks, results only on synthetic data and self-defined functions may not suffice.

---

> ### Author Response · Authors · 2022-08-01
> **Response to Reviewer 6nL3**
>
> We thank the reviewer for his comments and suggestions.
>
> About the assumption on the distribution, see our response to reviewer u1hF.
>
> Note that there exists plenty of work exploring the performance of convolutional kernels on image classification tasks (e.g., [6,35,36] on CIFAR-10). In this work, we did not include numerical experiments on real data, as our primary focus is theoretical. We included Figure 1 to show that the predictions (Table 1) derived in the high-dimensional framework (requiring large $d, q$) remain valid in moderate/low dimension ($d = 30, q = 10$). We believe such numerical simulations have justified the theoretical predictions in our idealized setting, which makes the paper self-contained.
>
> Using the insights of this paper, it would be very interesting to do a simulation study on real data set to explore locality, invariance and interaction between the patches. For example, plotting the test error versus the patch size, the pooling window size etc. However, simulating convolutional kernels is computational costly and such a study would deserve a separate paper. We hope to share some progress on that topic in the future.

---

### Official Review · Reviewer_rgYi · 2022-07-24

**Rating:** 7
**Confidence:** 4
**Soundness:** 4 excellent
**Presentation:** 3 good
**Contribution:** 4 excellent

**Summary:**

The paper studies statistical and approximation properties of shallow convolutional architectures with varying patch sizes and pooling window sizes, using corresponding kernels. The authors provide both statistical rates in finite dimension, as well as exact asymptotics of the excess risk in high dimensional regimes. The theory is complemented with experiments that illustrate the different scalings.

**Questions:**

* Could you give more details on the difficulties overlapping local pooling? What do you mean by "mixing of eigenvalues," and what is causing this?

* Given that the main goal of the paper is to derive exact and rigorous asymptotics, why did you also include finite-dimensional bounds? I find this to make the paper unnecessarily longer and less focused. Parts of the appendix also seem somewhat superfluous, e.g. section B.1.

Minor points/typos:
- L140-142: do you mean that the uniform distribution on the torus $[0,1]^d$, as considered in [22], is not amenable to a high-dimensional analysis?
- L179: scaler -> scalar
- In Theorem 3, you are using the notation $\ll$ in contrast to Theorems 2 and 5 which use~$0 < \nu < 1$. It would be good to use a consistent notation in each of these results.

**Strengths And Weaknesses:**

Understanding the role of neural network architectures for learning specific classes of functions is an important problem, which makes this paper timely and relevant for the community.

This work is part of a series of recent papers (many of them concurrent to the present paper) studying statistical properties of convolutional kernels. The main strength of the present paper is that it provides exact asymptotic expressions of the risk in certain high-dimensional scalings, using the powerful machinery in [38], which ensures that the derived quantities are tight. I believe this by itself makes it worthy of publication.

In addition, it considers architectures with overlapping patches (like [22] but unlike [6,43,48]), with dot-product kernels (unlike [22] which considers less realistic translation-invariant kernels), and localized pooling (like [6] but unlike [22,48]). This makes the setting richer in a sense compared to these other works, although the data distribution considered (on the discrete hypercube) is less realistic. The comparison to these works could be made more precise in the manuscript. The recent work of [Geifman et al.](https://arxiv.org/abs/2203.09255) should also be cited.

---

> ### Author Response · Authors · 2022-08-01
> **Response to Reviewer rgYi**
>
> We thank the reviewer for his comments and suggestions.
>
> In the eigen-decomposition of convolutional kernels with overlapping local pooling (Equation 9), the eigenvalues of degree-$k$ parity functions of frequency $j$ (as eigen-functions) are proportional to $\xi_k \kappa_j$, where $\xi_k \sim q^{-k}$ and $\kappa_j$ scales from $1/d$ to $1$ as $j$ varies from $1$ to $d$. Since $d \gg q$, some higher degree polynomials  will have eigenvalues that are bigger than some lower degree polynomials. This is what we meant by "mixing of eigenvalues". Such "mixing of eigenvalues" phenomenon will make it less straightforward to check the assumptions to apply the general framework in [38]. However, we believe that this is purely a technical problem: the shrinkage prediction (Equation 41) should still be correct in that case. We believe that one can adapt the proof of [38] to prove the high dimensional asymptotic result for overlapping local pooling, but the proof will be long and tedious.
>
> We included source-capacity condition bounds (finite-dimensional bounds) as those are more standard and are the usual approach for bounding the test error of kernel methods. They are much easier to derive than the asymptotic predictions and we found them useful to motivate the high-dimensional approach. In a final version of the paper, we will less emphasize the finite-dimensional result and shorten the related appendices.
>
>
> The torus case $[0,1]^d$ of [22] might fall under the assumptions of [38]: we only need to show hypercontractivity of the low-frequency functions on the torus.

---

### Official Review · Reviewer_u1hF · 2022-07-24

**Rating:** 4
**Confidence:** 4
**Soundness:** 2 fair
**Presentation:** 3 good
**Contribution:** 2 fair

**Summary:**

This is a paper that studies the theoretical properties of the convolution layer when used with the other common layers in the neural networks.


**Questions:**

Can the authors provide a clear intuition of this paper?

**Ethics Review Area:**

["I don’t know"]

**Limitations:**

The assumption of this theory is irrelevant to the nature of learning. If the data is uniform random then no learning should be done.

**Strengths And Weaknesses:**

Strengths:
The authors are motivated to develop theories.

Weakness:

There is no clear intuition behind this paper.  Moreover, I do not find the theories to be of practical use -- the basic assumption of this paper is on something purely theoretical. And this assumption strongly violates the properties of real world data. The numbers reported in the paper are not convincing.

"This paper, we consider the stylized setting of covariates (image pixels) uniformly distributed on the hypercube, and characterize exactly the RKHS of kernels composed of single layers of convolution, pooling, and downsampling operations."

Note that the paper puts its foundation on something wrong. Natural images statistics strongly violates the condition. Figuring out and utilizing the correlation in the real world data is a critical aspect of learning. The following paper provides a discussion for a one layer convolution operation:  Ye, Chengxi, et al. "Network deconvolution." arXiv preprint arXiv:1905.11926 (2019). I wonder if the findings of this paper shares any connections with it?

1. Do not use the RKHS acronym in the abstract.
2. L31 nonlinear convolution?

---

> ### Author Response · Authors · 2022-08-01
> **Response to Reviewer u1hF**
>
> We thank the reviewer for his comments and suggestions.
>
> **Intuition:**
>
> The goal of this paper is to provide a precise picture of the interplay between model choice, data distribution and generalization, for convolutional architecture. The basic premise of architecture design is that models that are adapted to a given task (data distribution) are more sampled efficient ('inductive bias'). This is a widely accepted explanation for the success of convolutional architectures. However, theoretical results that make this picture rigorous and precise are lacking. How does pooling bias learning towards translation-invariant functions? How do convolution and pooling operations interact? How does each of these operations impact the sample complexity for learning a given function? Existing studies only analyze the approximation error without generalization, consider generalization under abstract conditions or without tractable algorithm, or make simplifications on the architecture.
>
> In this paper, we decided to get a clear, unequivocal and rigorous picture in the case of one-layer convolutional kernels. (Note that Kernel methods are interesting as it removes the computational issue of training CNNs--difficult to study--: the approximation and generalization errors only depend on the eigendecomposition of the kernel and the target function.) To achieve this, we made two choices: 1) we consider an input distribution that is simple enough so that we can fully and explicitly characterize the eigendecomposition of these kernels (without further simplifications); 2) we consider the generalization error in high dimension ($n,d,q$ large), as it provides precise predictions that hold for a given function (instead of minmax and without the usual abstract 'source condition'). This allows us to provide an explicit and rigorous answer to the following question: given a target function and a number of samples $n$, what is the impact of the architecture on the test error? (see for example Table 1 for the inverse question: given a degree-$\ell$ invariant polynomial, what is the impact of the architecture on $n$ to achieve small test error). We believe that such a simple, explicit and rigorous case study adds conceptual value to the community.
>
>
> **Practical motivation:**
>
>  Besides this theoretical motivation, there is a practical motivation. The case of convolutional kernels is particularly intriguing as they currently achieve state-of-the-art performance for kernel methods on CIFAR-10, vastly outperforming previously handcrafted fixed-feature methods. For example, one layer convolutional kernel (as studied in our paper) achieves $81$% accuracy on CIFAR-10 [6], against $79.6$% for the best former unsupervised feature-extraction method [14]. Precisely understanding the inductive bias of these methods, coupled with simulation studies, would improve our understanding of Kernel/feature engineering as well as of the properties of the target function. For example, adding an extra layer with degree-2 polynomial activation to the one-layer CK increases the test accuracy from $81$% and $87$% on CIFAR-10 [6]: this increase is due to allowing pairwise interactions between the patches.

---

> > ### Author Response · Authors · 2022-08-01
> > **Response to Reviewer u1hF (2)**
> >
> > **Assumption data uniform on the hypercube:**
> >
> > We could extend our results to more general data distribution at the price of either less precise results [6,22] or by simplifying the architecture such as non-overlapping patches [48]. In this paper, we chose instead to get precise and unequivocal results about the generalization and approximation errors. Note that we believe that our results can directly extend to data uniformly distributed on the continuous hypercube $[0,1]^d$ (see [22]): we would only need to show a technical lemma on hypercontractivity of low-frequency Fourier functions [38] and our predictions would look essentially the same.
> >
> > We further remark that such idealized assumption on the image distribution $p(x)$ (where $x$ is the pixel representation of the image) have been used in a recent line of theoretical papers: uniform on the sphere [7,23,39,43,48] or uniform on the continuous cube [22]. This is a compromise in modelling: this idealized 'mathematical model' allows to get precise theoretical result, and to focus instead on the conditional distribution $p(y|x)$ (where $y$ is the label associated to image $x$). In this paper, we encode in $p(y|x)$ some natural properties such as locality and translation invariance.
> >
> > We believe that the main difference with real world data is the isotropy assumption (as pointed out by the reviewer):  real world data are highly anisotropic [23]. However, in practice, it was noticed in [35,36,6] that patch whitening is important for the good performance of CKs: the data on each patch is whitened to have identity covariance (this is related to the deconvolution in Chengxi et al. (2019)). This leads to an increase of performance of $4-6$%. Hence, the input of these models are often preprocessed so that the pixels are uncorrelated and isotropic.
> >
> > For the above reasons, while uniform on hypercube might be very simple, we believe that this is a reasonable model for high-dimensional data that has been whitened, and is justified by the theoretical purpose of our paper. We agree that extending the theory to more realistic data distribution is important, and should be further explored.

---

### Meta-Review · Area_Chair_81cU · 2022-08-31

**Recommendation:** Accept
**Confidence:** Certain

**Metareview:**

The reviewers appreciated the contributions. Several concerns were raised, and the authors addressed them in their submitted response.

Accept.

**Award:**

No

---

### Decision · Program_Chairs · 2022-09-14

Accept